# Ectoparasites enhance survival by suppressing host exploration and limiting dispersal

Pengbo Liu[1], Dongsheng Ren[1], Guichang Li[1], Xiaoming Xu[2], Luca Presotto[3,4], Wei Liu[2], Ning Zhao[1], Dongmei Li[1], Min Chen[1], Jun Wang[1], Xiaobo Liu [1], Chunchun Zhao[1], Liang Lu [1,5] ✉ & Qiyong Liu [1,5] ✉

Parasites enhance their fitness by manipulating host dispersal. However, the strategies used by ectoparasites to influence host movement and the underlying mechanisms remain poorly understood. Here, we show that ectoparasites alter metabolic activity in specific brain regions of mice, with evidence pointing to a potential role for microglial activation in the prefrontal cortex. This activation appears to contribute to synaptic changes and altered neuronal differentiation, particularly in GABAergic neurons. Consequently, exploratory behavior decreases—an effect likely mediated through the skin–brain axis. In both indoor and field experiments with striped hamsters, ectoparasites reduce host exploration and modify their dispersal patterns. This behavioral shift ultimately restricts the host's distribution, enabling parasites to avoid environmental pressures. Our findings reveal that ectoparasites limit host dispersal to improve their own fitness, offering key insights for parasite control strategies that promote health and preserve ecological stability within the One Health framework.

Host dispersal significantly influences the ecology and evolution of parasites. Most notably, it can facilitate the spread of parasites across large spatial scales[1]. However, host dispersal can also impact parasite survival[2]. Evidence indicate that hosts employ dispersal to effectively remove parasites[3–5]. To survive, some parasites have evolved the ability to precisely and adaptively manipulate host dispersal[6], such as altering the likelihood and timing of departure[7] or even influencing the distance of transfer[8].

Although host manipulation by parasites has been documented in several hundred host-parasite associations across major phylogenetic groups, manipulation by ectoparasites appears to have been relatively understudied. Ectoparasites are organisms that live on a host's surface, feeding on its blood or skin, including fleas, ticks, and mites. For

ectoparasites, host dispersal can introduce greater stress, as ectoparasites are more vulnerable to environmental pressure[9]. Consequently, these ectoparasites may need to develop strategies to manipulate their hosts in order to enhance their own fitness. We hypothesize that ectoparasites may suppress their hosts' exploratory behavior to improve the parasites' survival, thereby reducing host dispersal and occupancy.

The flea-rodent parasitic system provides a unique opportunity to deepen our understanding of how ectoparasites manipulate their hosts' dispersal. Fleas are significant vectors of zoonotic diseases, including plague, murine typhus, and rural epidemic typhus[10]. At the same time, wild rodents—the primary reservoirs of these pathogens—are almost always infested with fleas. Historically, outbreaks of vector-borne diseases have often been linked to changes at the human-animal

[1]National Key Laboratory of Intelligent Tracking and Forecasting for Infectious Diseases, National Institute for Communicable Disease Control and Prevention, Chinese Center for Disease Control and Prevention, Beijing, China. [2]National Key Laboratory of Animal Biodiversity Conservation and Integrated Pest Management, Institute of Zoology, Chinese Academy of Sciences, Beijing, China. [3]Department of physics G. Occhialini, University of Milano-Bicocca, Milano, MI, Italy. [4]Milan Centre for Neuroscience, University of Milano-Bicocca, Milan, Italy. [5]These authors contributed equally: Liang Lu, Qiyong Liu. ✉e-mail: luliang@icdc.cn; liuqiyong@icdc.cn

interface[11]. Parasite-induced behavioral manipulation of hosts can significantly alter the likelihood of pathogen spillover from wildlife to humans. The One Health approach helps prevent and control outbreaks of vector-borne diseases by recognizing the interconnection between human, animal, and environmental health[12,13]. Investigating how ectoparasites influence host behavior is therefore crucial for understanding pathogen spillover, monitoring shifts at the human-animal interface, and advancing One Health strategies.

Here, we show that ectoparasites manipulate host dispersal to enhance their own survival using two rodent models. Mice infected with fleas exhibit reduced exploratory behavior and increased anxiety, along with metabolic and functional abnormalities in the prefrontal cortex, including microglial activation and GABAergic neuron damage. Using a flea–striped hamster (*Cricetulus barabensis*) model, we replicate these effects through controlled indoor infections and enclosed field experiments, confirming that flea bites limit host exploration. Simulations further reveal that fleas restrict host movement to influence host distribution, helping them avoid environmental heterogeneity (Fig. 1). This study expands our understanding of how parasites manipulate host behavior, identifies potential neurobiological mechanisms underlying this phenomenon, and provides insights into parasite-induced behavioral changes that inform intervention strategies within the One Health framework.

## Results

### Flea bites suppress exploratory behavior

Fleas are blood-sucking arthropods, and different species exert varying levels of selective pressure on their hosts[14]. Meanwhile, the greater the pressure exerted by fleas, the lower the survival rate[15]. To establish a stable flea-mouse system, we compared the survival rates of two flea species, *Ctenocephalides felis* (*C. felis*) and *Xenopsylla cheopis* (*X. cheopis*). After one week, all *C. felis* had died, while approximately 50% of *X. cheopis* fleas survived (Supplementary Fig. 1). Thus, we chose *X. cheopis* for our infection experiments to ensure sustained chronic stress.

After infecting for four weeks, we immediately conducted behavioral tests. We assessed dispersal behavior by measuring the time mice spent exploring the central area in the open field test (OFT) and the open arms in the elevated plus maze (EPM). Compared to the Flea− group, the Flea+ group spent less time in the central area of the open field and made fewer attempts to enter it (Fig. 2a). Moreover, in the EPM test, the Flea+ group spent less time in the open arms (Fig. 2b), indicating that flea bites increased anxiety-like behaviors/reduced exploratory behavior in the mice.

### Altered brain's expression patterns

To determine which brain regions are affected by flea bites, we mapped glucose uptake in the brain, using tail injections of 2-deoxy-2-

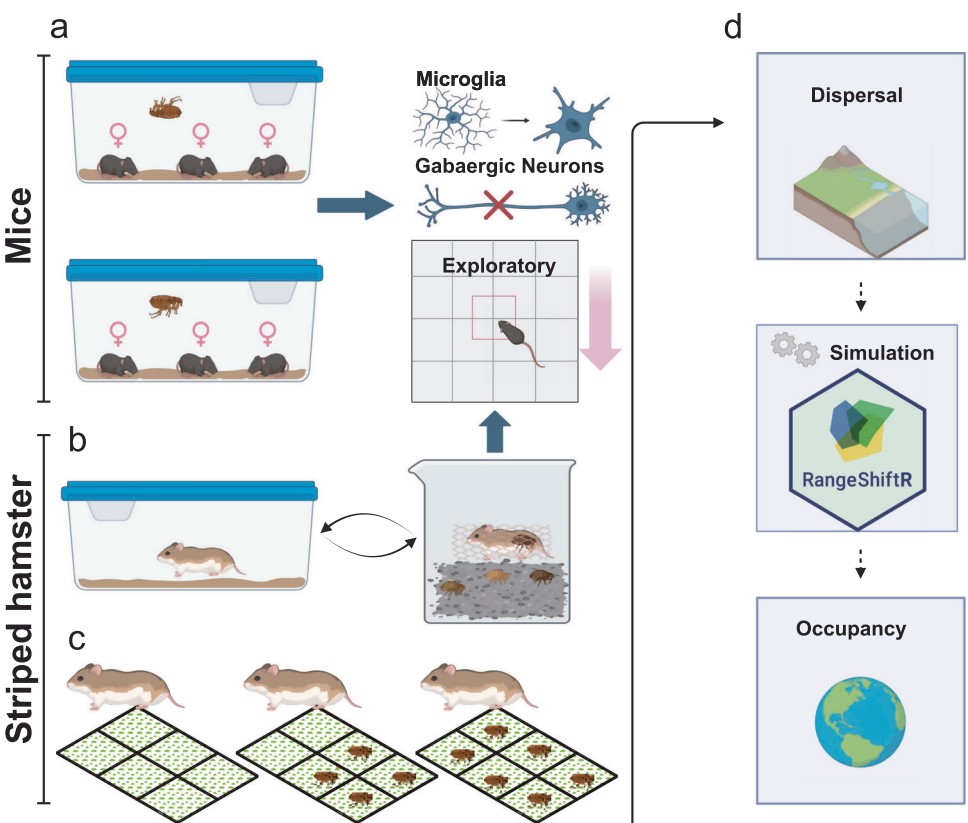

**Fig. 1 | Schematic diagrams of the experiment design in this study, created in BioRender.** Liu, P. (2025) https://BioRender.com/t7hic5v. **a** Three female mice were housed per cage and allowed to acclimate for one week. Flea infection was initiated by introducing 150 live fleas (*X. cheopis*) into each cage in the Flea+ group, while the Flea− group received the same number of heat-killed fleas as a control. Each week, remaining fleas were removed, and 150 new fleas were reintroduced to maintain infection. After four weeks of continuous infection, behavioral and molecular assays were conducted to test the hypothesis that ectoparasites reduce host exploratory behavior and to investigate the underlying neuro-molecular mechanisms. **b** Striped hamsters captured from the wild were acclimated to laboratory conditions for one month before flea infection. The fleas used for infection were derived from a wild population naturally infecting striped hamsters.

Hamsters were housed individually at night and exposed to fleas in glass tanks during the day. After four weeks of infection, behavioral experiments were performed to assess changes in exploratory behavior. **c** Six 30 m × 30 m field enclosures were constructed to conduct three rounds of flea infection experiments, each lasting four weeks. In the first round, only striped hamsters were introduced. In the second round, fleas were added to a subset of enclosures, and in the third round, fleas were introduced to all enclosures. After each round, hamsters were recaptured, and their exploratory behavior was evaluated based on capture rates. **d** Using dispersal data from the experiments, combined with demographic data and habitat suitability maps, the RangeShiftR model was employed to simulate the impact of flea-induced reductions in exploratory behavior on the distribution and range dynamics of striped hamsters.

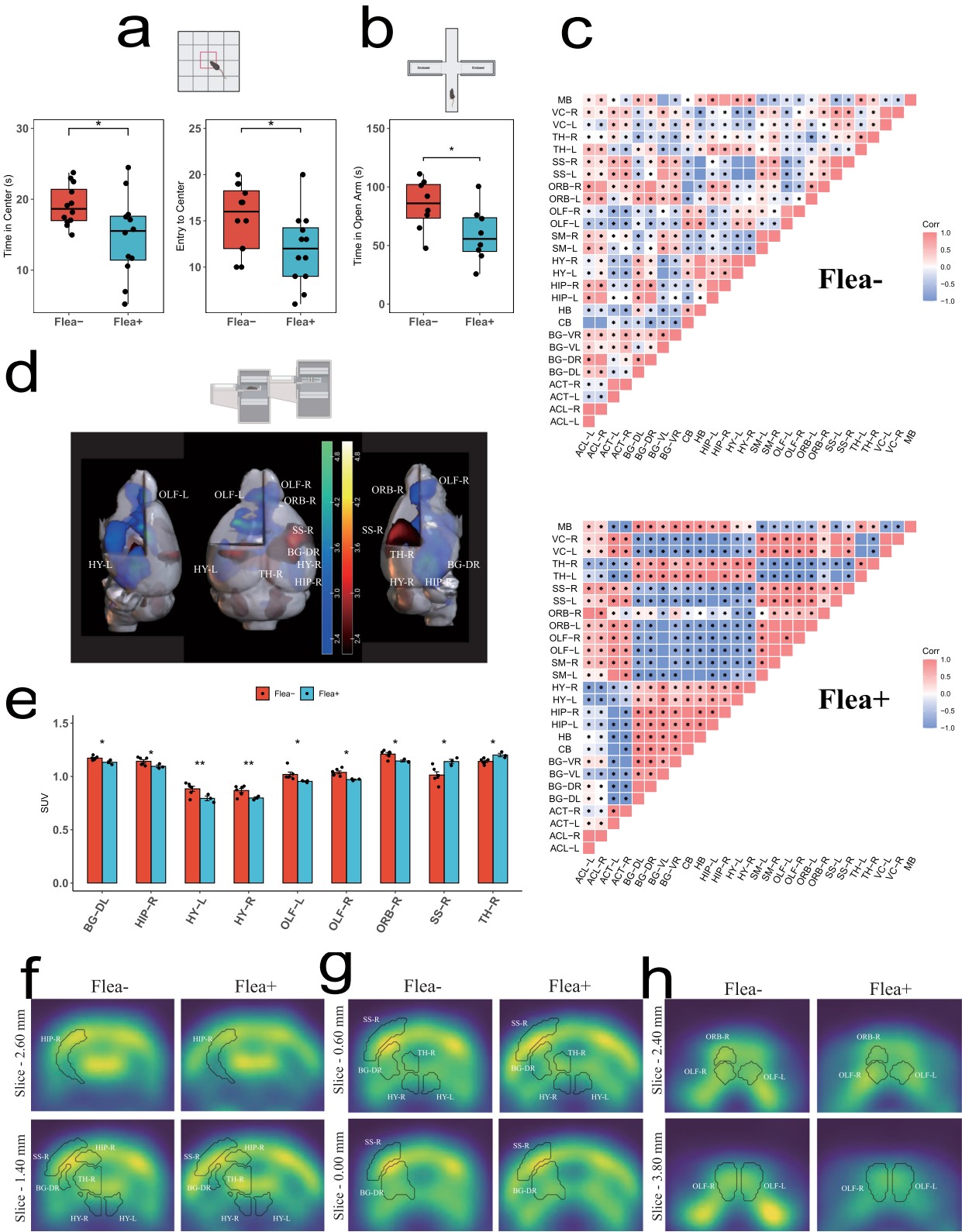

[fluorine-18] fluoro-D-glucose (18F-FDG)[16], which is rapidly absorbed by active brain regions. We then aligned the metabolic images produced by PET-CT with the Allen Brain Atlas and calculated the standard uptake value for 18F-FDG in each brain region[17]. After flea bites, there was a significant increase in the correlation of activity across different brain regions (Fig. 2c), indicating that flea bites induce consistent changes in brain activity. Flea bites caused changes in glucose uptake in Basal Ganglia Dorsal Right (BG-DR), Hippocampal Right (HIP-R),

Hypothalamus Left (HY-L), Hypothalamus Right (HY-R), Olfactory Left (OLF-L), Olfactory Right (OLF-R), Orbital Right (ORB-R), Somatosensory Right (SS-R), and Thalamus Right (TH-R) (Fig. 2d–h and Supplementary Fig. 2). Notably, many of these regions have also been linked to anxiety in previous studies[18,19]. We concluded that flea bites lead to changes in multiple brain regions, including those involved in emotion regulation[20–22]. Given the impact of flea bites on anxiety in mice, our subsequent research will focus on three key areas: the prefrontal

**Fig. 2 | Flea bites reduce mice's exploratory behavior and alter brain metabolic patterns. a** Open field test results shown as a boxplot: Flea− (red, $n = 12$) and Flea+ (blue, $n = 12$). One-sided $t$ test was used. Flea+ mice exhibited reduced time in the central area (left) and fewer entries into the central area (right) compared to Flea− mice. Thick bars indicate the interquartile range (IQR) around the median, and whiskers represent 1.5 times the interquartile range (maxima: $Q3 + 1.5 \times IQR$, minima: $Q1 - 1.5 \times IQR$). Each black dot represents an individual. Asterisks indicate the significance. **b** Elevated plus maze results shown as a boxplot: Flea− (red, $n = 8$) and Flea+ (blue, $n = 8$). One-sided $t$ test was used. Flea+ mice spent less time in the open arms compared to Flea− mice. Thick bars indicate the interquartile range (IQR) around the median, and whiskers represent 1.5 times the interquartile range (maxima: $Q3 + 1.5 \times IQR$, minima: $Q1 - 1.5 \times IQR$). Each black dot represents an individual. Asterisks indicate the significance. **c** Spearman correlation of 18F-FDG

uptake between brain regions in the Flea− group ($n = 6$) and Flea+ group ($n = 3$). Two-sided $t$ test was used. **d** 3D rendering of SPM-based voxel-level statistical analysis overlaid on an MR template image. Blue indicates hypometabolic regions in the Flea+ group, while red indicates hypermetabolic regions. Statistically significant clusters are marked in the figure. **e** Bar graph showing brain regions with significant differences in $^{18}$F-FDG uptake between the Flea− (red, $n = 6$) and Flea+ (blue, $n = 3$) groups. Two-sided $t$ test was used. Each black dot represents an individual. Asterisks indicate brain regions with significant differences. Data are presented as mean ± SEM. **f–h** Representative images of selected axial slices after spatial normalization. ARA standardized coordinates are indicated. PET brain activity is scaled to the cerebral global mean, with identical window/level settings applied to all animals. Source data are provided as a Source Data file. **p < 0.01, *p < 0.05.

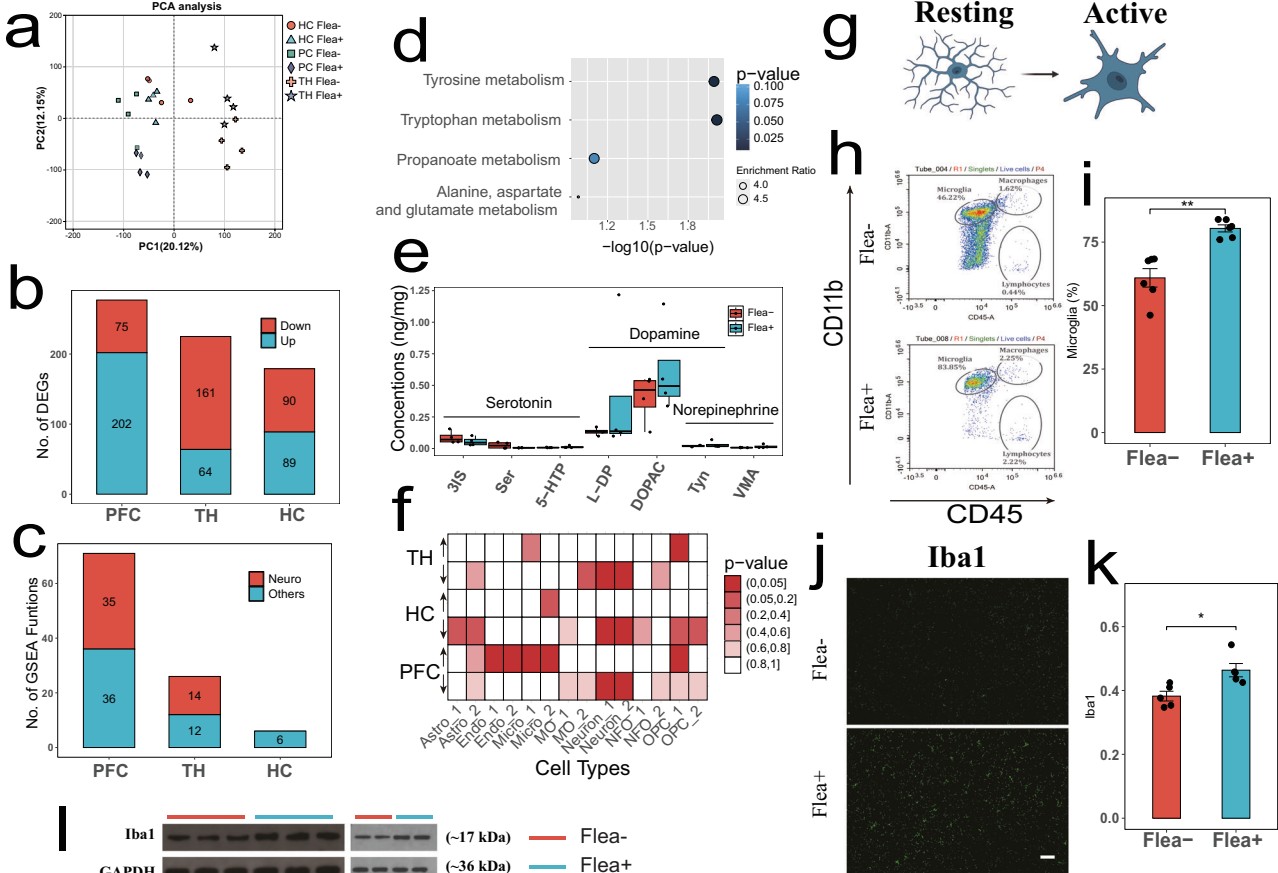

**Fig. 3 | Flea bites alter the expression patterns in the prefrontal cortex.**
**a** Principal component analysis of all transcriptome data showing tight clustering of brain regions. **b** Differential gene analysis revealed the number of differentially expressed genes in the three tested brain regions, using one-way ANOVA. **c** GSEA analysis of transcriptomic data from three brain regions showed that the PFC was enriched with the most neural function-related pathways, supporting that flea bites have the greatest impact on PFC function. **d** KEGG pathway enrichment analysis of differential neurotransmitters in the PFC. Fisher's Exact Test was used.
**e** Neurotransmitters related to synthesizing dopamine, norepinephrine, and serotonin all showed intergroup differences based on PET-CT for Flea− ($n = 6$) and Flea+ ($n = 3$) groups; Thick bars indicate the interquartile range (IQR) around the median, and whiskers represent 1.5 times the interquartile range (maxima: $Q3 + 1.5 \times IQR$, minima: $Q1 - 1.5 \times IQR$). Data are presented as mean ± SEM. **f** Cell-specific enrichment analysis based on DEGs from the transcriptomic data of three brain regions: $p$-value < 0.05 in red. Fisher's Exact Test was used. **g** Schematic of

microglial cells in resting and activated states, created in BioRender. Liu, P. (2025) https://BioRender.com/lrddnmc. **h** Representative flow cytometry plots of the PFC from the Flea− and Flea+ group. **i** Comparison of activated microglial cells between groups in the PFC using quantitative flow cytometry. Two-sided $t$ test was used. Red represents the Flea− group ($n = 6$), and blue represents the Flea+ group ($n = 6$). Each black dot represents an individual. Asterisks indicates brain regions with significant differences. Data are presented as mean ± SEM. **j** Representative images of brain sections stained with IBA1 from Flea− and Flea+ mice; Scale 50 μm. **k** Comparison of IBA1 protein fluorescence intensity in brain sections between the Flea− (red, $n = 5$) and Flea+ (blue, $n = 5$) groups of mice. Two-sided $t$ test was used. Each black dot represents an individual. Asterisks indicates brain regions with significant differences. Data are presented as mean ± SEM. **l** Western blot images of IBA1 protein expression in the PFC of mice from the Flea− and Flea+ groups. Source data are provided as a Source Data file. **p < 0.01, *p < 0.05.

cortex (PFC, representing orbital cortex), thalamus (TH), and hippocampus (HC).

To uncover the molecular mechanisms underlying flea bite-induced anxiety in mice, we performed transcriptome sequencing on

three brain regions from both groups. Cluster analysis showed significant differences in gene expression of flea bite in different brain regions (Fig. 3a), and high intergroup repeatability in brain regions (Supplementary Figs. 3a, 4a, and 5a). The total number of differentially

expressed genes (DEGs) was similar across the three brain regions (Fig. 3b), indicating that they were affected to a comparable extent. However, the PFC exhibited a higher number of upregulated DEGs compared to the HC and TH (TH: 225; PFC: 277; HC: 179, Supplementary Figs. 3b, 4b, and 5b, and Supplementary data 1).

We conducted functional enrichment analysis on the upregulated and downregulated DEGs in each brain region (Supplementary data 2). Some functions that are enriched with upregulated genes in all three regions include hemoglobin complexes, oxygen transport, and detoxification (Supplementary Figs. 3c, 4c, and 5c). This indicates a coordinated response across the regions, potentially enhancing oxygen transport and detoxification to protect against oxidative stress. In the PFC, additional upregulated functions included immune responses, metal ion regulation, erythrocyte development, and extracellular space functions. The downregulated genes in the PFC, associated with cell fate and neuronal differentiation, suggest significant alterations that may indicate long-term damage from flea bites (Supplementary Fig. 3d). In contrast, the HC and TH had downregulated genes linked to neuropeptide signaling and stress regulation (Supplementary Figs. 4d and 5d). Gene set enrichment analysis (GSEA) revealed that flea bites primarily affected PFC function, as a greater number of neurophysiological processes were suppressed in the PFC of the Flea+ group (PFC: 35, TH: 14, HC: 0; Fig. 3c). After flea bites, the PFC exhibited activated metabolic processes, energy production, immune responses, and inflammation, while functions related to synaptic plasticity, signal transduction, and neuron development were suppressed (Supplementary Figs. 3e–g, 4e–g, and 5e–g). This suppression may correlate with the decreased exploratory behavior observed in the mice[23].

We also conducted neurotransmitter metabolomics analysis on the three brain regions mentioned above. Both PCA and PLS-DA clustering analyses demonstrated high reproducibility of the experimental treatment across all brain regions (Supplementary Fig. 6a, b, d, e, g, h). The PFC exhibited 14 differential neurotransmitter metabolites (Supplementary Fig. 5c), more than 6 identified in the HC (Supplementary Fig. 6f) and 3 in the TH (Supplementary Fig. 6i). We performed KEGG pathway analysis on these differential metabolites. In the PFC, significant enrichment was observed in Tyrosine and Tryptophan metabolism pathways (Fig. 3d), which are crucial for synthesizing dopamine, norepinephrine, and serotonin. These neurotransmitters regulate mood, cognition, and pain perception, with pathway abnormalities potentially impacting neural excitability and signal transmission[24,25]. Notably, we observed a decrease in tryptophan and serotonin levels in the Flea+ group (Fig. 3e). In the HC, enrichment of the Histidine metabolism pathway (Supplementary Fig. 6j), which influences histamine production—a neurotransmitter involved in wakefulness, attention, and appetite—was noted, though its role in emotion regulation is less clear[26].

## Microglia activation

We infer that flea bites primarily impact the PFC, which may be linked to reduced exploratory behavior in mice. Cell-specific enrichment analysis revealed that neuron-specific expression decreased across all three brain regions, while microglia and endothelial cell expression increased in the PFC (Fig. 3f). Microglia, the brain's resident macrophages, are responsible for clearing damaged neurons and promoting neuronal differentiation and development[27]. Genes related to microglial activation, such as *DAP12*, *FCER1*, were upregulated in the Flea+ group (Supplementary Fig. 7). To test the hypothesis that flea bites promote microglial activation (Fig. 3h), we conducted flow cytometry, which revealed a significant increase in CD45+ and CD11b+ cells in the PFC of Flea+ mice (Fig. 3h, i). This finding was further supported by immunofluorescence (Fig. 3j, k) and Western blot (WB) analyses (Fig. 3l and Supplementary Fig. 8a), confirming that flea bites preferentially induce microglial activation in the PFC.

## Reduction in GABAergic neurons

We observed a decreased expression of the mature neuron (NeuN), alongside an apoptotic (TUNEL) increase in the PFC (Fig. 4a–c). To determine whether damage to neuronal synapse was responsible for impaired neuronal differentiation and subsequent reduction in neuronal expression, we observed a decrease in PSD95 expression in the Flea+ group, accompanied by an increase in the synaptic apoptosis marker Homer1, indicating damage to synaptic terminals (Fig. 4d–f). Furthermore, neuronal synaptic damage was accompanied by a reduction in synaptic vesicles (Fig. 4d, g). Transmission electron microscopy revealed that the synaptic terminals in the Flea+ group were more fragmented (Fig. 4h).

Given the transcriptomic evidence of downregulated GABAergic neuron differentiation, we investigated whether the damage and reduced expression of GABAergic neurons contributed to the observed effects. GABAergic neurons, responsible for releasing γ-aminobutyric acid (GABA), are the main inhibitory neurons in the central nervous system (CNS)[28]. When these neurons malfunction, mice remain in a heightened state of arousal, leading to increased anxiety-like behaviors[29,30]. The presynaptic marker GAD65/67 and the postsynaptic marker GABARγ2 both showed decreases after flea bites (Fig. 4i–l and Supplementary Fig. 8b, c). Considering the broader impact on pan-neuronal expression, we also examined the glutamatergic neuron marker VGLUT1, which similarly showed a decrease in expression (Fig. 4j, m). Furthermore, even after controlling for glutamatergic neurons, the reduction in GABAergic neurons in the Flea+ group remained significant and was independent of any decrease in the number of mature neurons (Fig. 4n), indicating that flea bites primarily cause structural damage to GABAergic neurons, potentially linked to reduced exploratory behavior.

## Systemic inflammation and PFC damage

Flea bites may manipulate the host's CNS through the skin-brain axis, which refers to the bidirectional communication between the skin and the CNS, influencing brain function and behavior[31]. When fleas feed on blood, they inject substances like anesthetics and anticoagulants into the host, which the host's immune system recognizes as antigens, initiating an immune response. The skin transcriptome showed high reproducibility between the Flea− and Flea+ groups (Supplementary Fig. 9a). We analyzed the skin transcriptome data, identifying 103 upregulated and 71 downregulated DEGs (Supplementary Fig. 9b). The upregulated DEGs were associated with numerous immune-related pathways (Fig. 5a and Supplementary Fig. 9c), including inflammation and oxidative damage, while no significant functions were enriched in the downregulated genes (Supplementary Fig. 9d). GSEA and the protein-protein interaction (PPI) network analysis highlighted that flea bites provoke an immune defense response in mice and cause vascular dilation (Supplementary Fig. 9e, f). We measured serum IgE levels, as previous research suggests that small antigens are first recognized by IgE, which activates mast cells to release histamine, initiating an immune response via IgG binding. In the Flea+ group, we observed an increase in serum IgE and skin IgG levels, accompanied by elevated expression of inflammatory cytokines that promote inflammation and oxidative stress (Fig. 5b). However, HE staining did not reveal significant skin damage (Supplementary Fig. 9g), indicating that flea bites may cause only mild inflammation in mice. This observation aligns with the finding that the downregulated gene set in the transcriptome did not show significant functional enrichment.

To explore whether flea bites can trigger systemic inflammation, we assessed serum cytokine levels and observed increased stress, inflammation, and oxidative stress in mice (Fig. 5c). Further ELISA analysis of the PFC reveals elevated levels of inflammatory and oxidative markers (Fig. 5d). Consistently, HE staining shows noticeable cellular damage in the PFC of the Flea+ group, with a small number of infiltrating mononuclear cells at the injury site and extensive nuclear

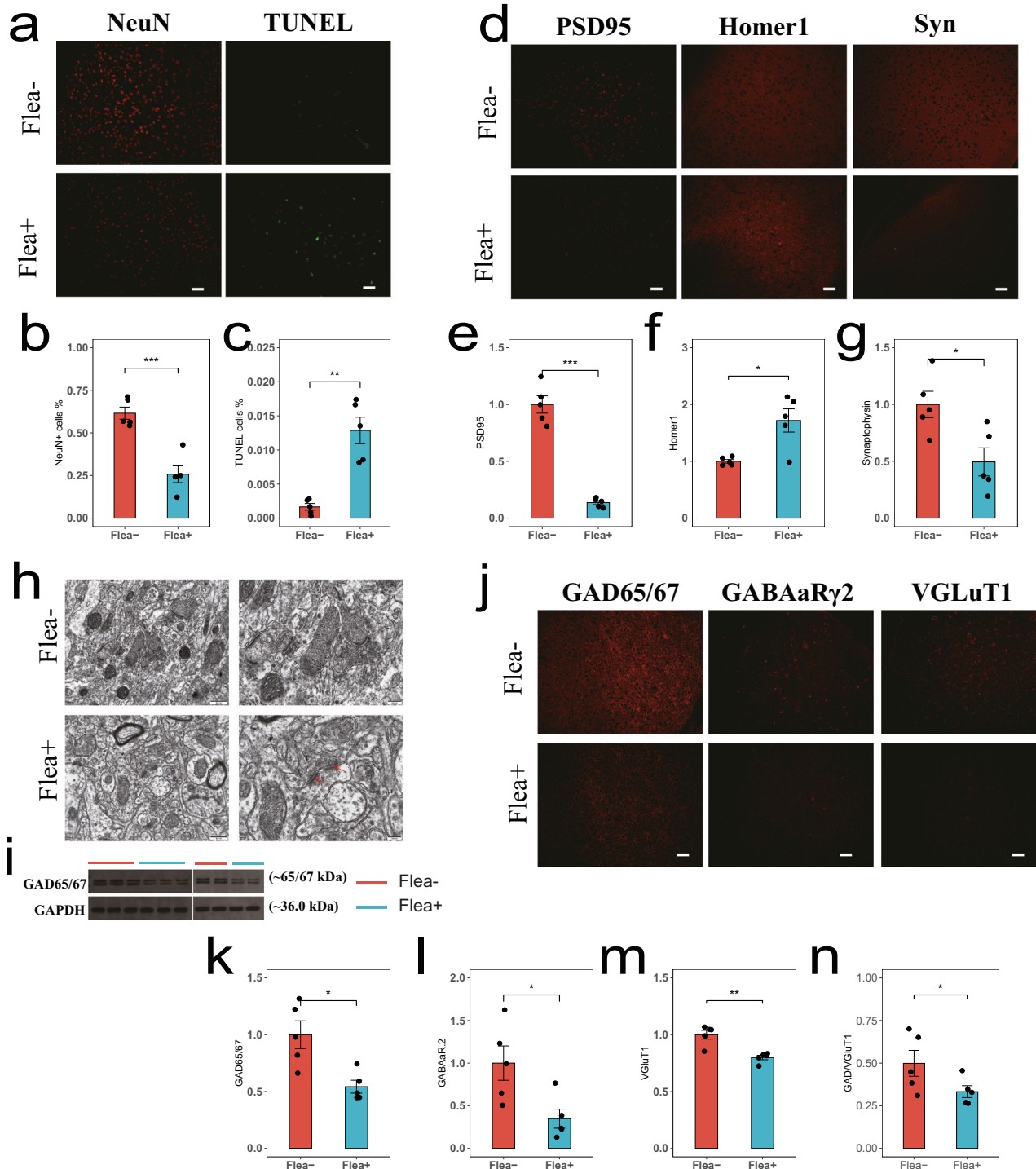

**Fig. 4 | Flea bites cause neuronal damage in the PFC and a reduction in GABAergic neurons. a** Representative images of brain sections from Flea− and Flea+ mice; NeuN (red) and TUNEL (green). Scale 50 μm. **b** Comparison of mature neurons in brain sections between the Flea− (red, $n = 5$) and Flea+ (blue, $n = 5$) groups of mice. Two-sided $t$ test was used. Asterisks indicates brain regions with significant differences. Data are presented as mean ± SEM. **c** Comparison of cell apoptosis in brain sections between the Flea− (red, $n = 5$) and Flea+ (blue, $n = 5$) groups of mice. Two-sided $t$ test was used. Asterisks indicates brain regions with significant differences. Data are presented as mean ± SEM. **d** Representative images of brain sections from Flea− and Flea+ mice; PSD95 (left), Homer1 (mid), and Syn (right). Scale 50 μm. **e–g** Comparison of synaptic structure: PSD95 (**e**), Homer1 (**f**), and Syn (**g**) between the Flea− (red, $n = 5$) and Flea+ (blue, $n = 5$) groups of mice. Two-sided $t$ test was used. Asterisks indicates brain regions with significant differences. Data are presented as mean ± SEM; Scale 50 μm.

**h** Representative transmission electron microscopy images.; Scale 500 nm (left) and 200 nm (right). Synaptic gaps between neurons are indicated by red arrows. **i** Western blot images of GAD65/67 expression in the PFC of mice from the Flea− and Flea+ groups. **j** Representative images of brain sections from Flea− and Flea+ mice; GAD65/67 (left), GABAaRγ2 (mid) and VGLuT1(right). Scale 50 μm. **k–m** Comparison of synaptic structure: GAD65/67 (**j**), GABAaRγ2 (**k**), and VGLuT1 (**l**) between the Flea− (red, $n = 5$) and Flea+ (blue, $n = 5$) groups of mice. Two-sided $t$ test was used. Each black dot represents an individual. Asterisks indicates brain regions with significant differences. Data are presented as mean ± SEM.
**n** Differences in GABAergic neuron expression levels between Flea− and Flea+ after controlling for glutamatergic neuron expression levels (Flea− $n = 5$, Flea+ $n = 5$). Two-sided $t$ test was used. Asterisks indicates brain regions with significant differences. Data are presented as mean ± SEM. Source data are provided as a Source Data file. ***$p < 0.001$, **$p < 0.01$, *$p < 0.05$.

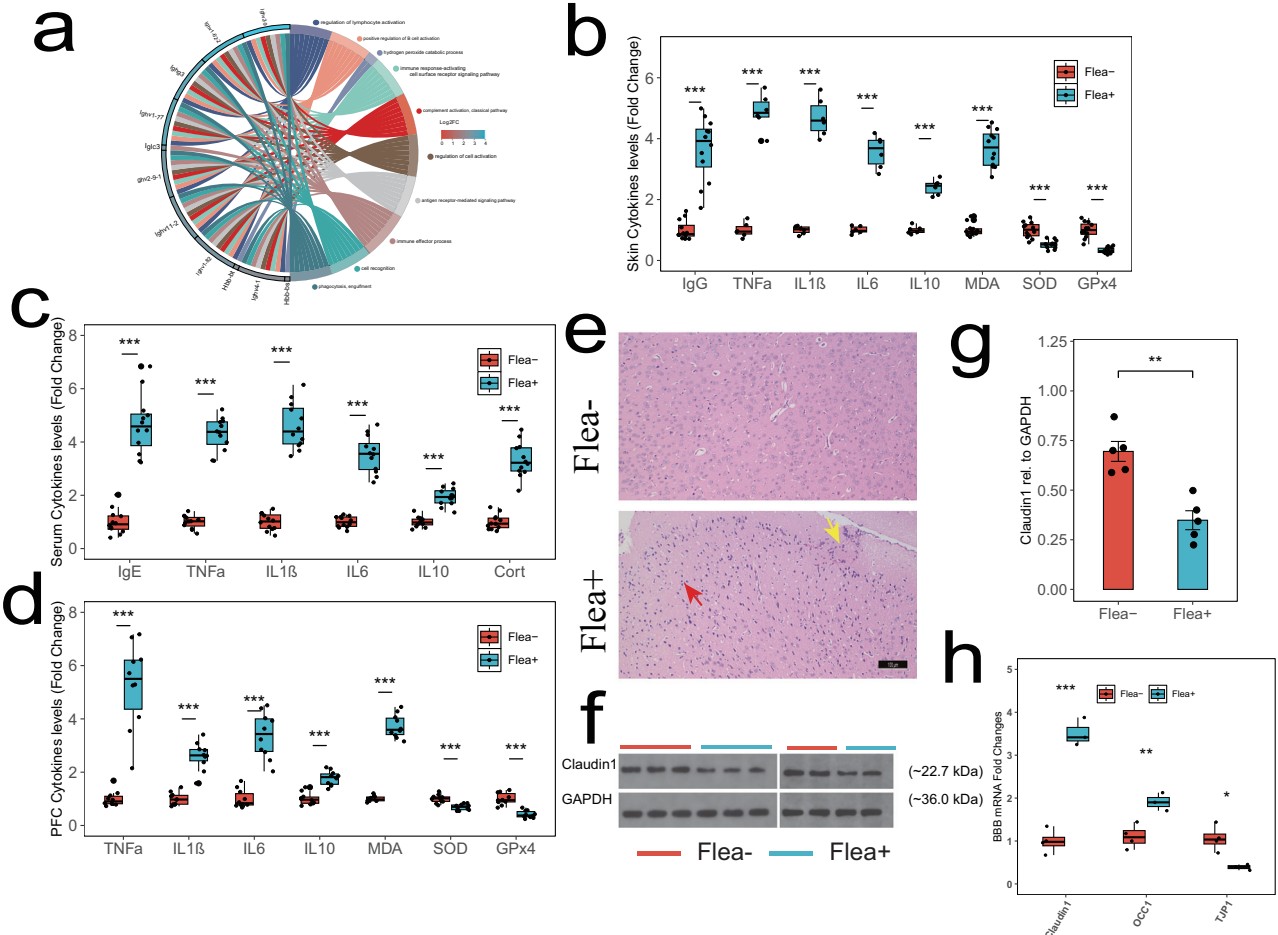

**Fig. 5 | Flea bites affect neural function through the skin-brain axis. a** GSEA result on upregulated genes from the mice skin transcriptome. Functions related to immunoglobulin and hemoglobin genes are significantly enriched. **b** Cytokine levels in mice skin presented in boxplot where red is Flea− ($n = 6/12$) and blue is Flea+ ($n = 6/12$). Two-sided $t$ test was used. Asterisks indicate the significance. All $p$-values were below 0.001. **c** Cytokine levels in mice serum presented in boxplot where red is Flea− ($n = 12$) and blue is Flea+ ($n = 12$). Two-sided $t$ test was used. Thick bars indicate the interquartile range (IQR) around the median, and whiskers represent 1.5 times the interquartile range (maxima: Q3 + 1.5 × IQR, minima: Q1 − 1.5 × IQR). Asterisks indicate the significance. All $p$-values were below 0.001. **d** Cytokine levels in mice PFC presented in boxplot where red is Flea− ($n = 10$) and blue is Flea+ ($n = 10$). Two-sided $t$ test was used. Asterisks indicate the significance. All $p$-values were below 0.001. **e** Representative HE plots of the PFC. In the Flea+ group, cortical tissue shows evident cellular damage, with minor

mononuclear cell infiltration (yellow arrow) at the injury site. Additionally, widespread nuclear pyknosis and condensation (red arrow) are observed in the parenchyma. The infected group exhibits a larger area of damage; Scale 50 μm. **f** Western blot analysis of Claudin1 markers of tight junction protein from the PFC brain region ($n = 5$ samples each group). **g** Comparison of Claudin1 in WB between the Flea− (red, $n = 5$) and Flea+ (blue, $n = 5$) groups of mice. Two-sided $t$ test was used. Each black dot represents an individual. Asterisks indicates brain regions with significant differences. Data are presented as mean ± SEM. Source data are provided as a Source Data file. **h** Validation by qPCR of select BBB genes altered between Flea− ($n = 3$) and Flea+ ($n = 3$) group. Two-sided $t$ test was used. Thick bars indicate the interquartile range (IQR) around the median, and whiskers represent 1.5 times the interquartile range (maxima: Q3 + 1.5 × IQR, minima: Q1 − 1.5 × IQR). Each black dot represents pooled sample of 3 decapitated individuals. Asterisks indicate the significance. ***$p < 0.001$, **$p < 0.01$, *$p < 0.05$.

pyknosis and condensation leading to deep staining in the parenchyma. Additionally, the Flea+ group exhibits a larger area of damage (Fig. 5e). WB analysis showed a decrease (~50%) in Claudin1 protein expression in the Flea+ group (Fig. 5f, g). Although this was not supported by the qPCR results (Fig. 5h), we still believe that systemic inflammation triggered PFC inflammation by disrupting the blood-brain barrier (BBB). In summary, flea bites trigger inflammation and oxidative stress in the skin, elevate stress levels, and are associated with systemic inflammation, which may contribute to PFC BBB alterations. This may lead to inflammation and oxidative changes in the PFC, potentially contributing to microglial activation.

### Reduced exploratory behavior in Flea+ striped hamster
We wanted to determine whether the observed reduction in exploratory behavior due to ectoparasite infestation in hosts could be

generalized to other parasite-host pairs in natural settings. To test this, we conducted a laboratory infection experiment using field-captured striped hamsters and their associated fleas (Fig. 6a and Supplementary Fig. 10a–c). After 4 weeks of infection, we assessed exploratory levels in the hamsters using an OFT. Compared to the Flea− group, female striped hamsters in the Flea+ group spent less time in the central area (Fig. 6b); however, there was no difference in the time spent in the central area between the Flea− and Flea+ groups of male striped hamsters (Supplementary Fig. 10d), similar to the findings in mice. Additionally, heightened immune responses, increased inflammation, and oxidative stress were detected in both the skin and serum of the infected hamsters based on elisa results (Fig. 6c, d).

To further evaluate whether this behavioral phenotype persisted in natural conditions, we conducted a field enclosure experiment, a larger-scale OFT (Supplementary Fig. 10e). We hypothesized that flea-

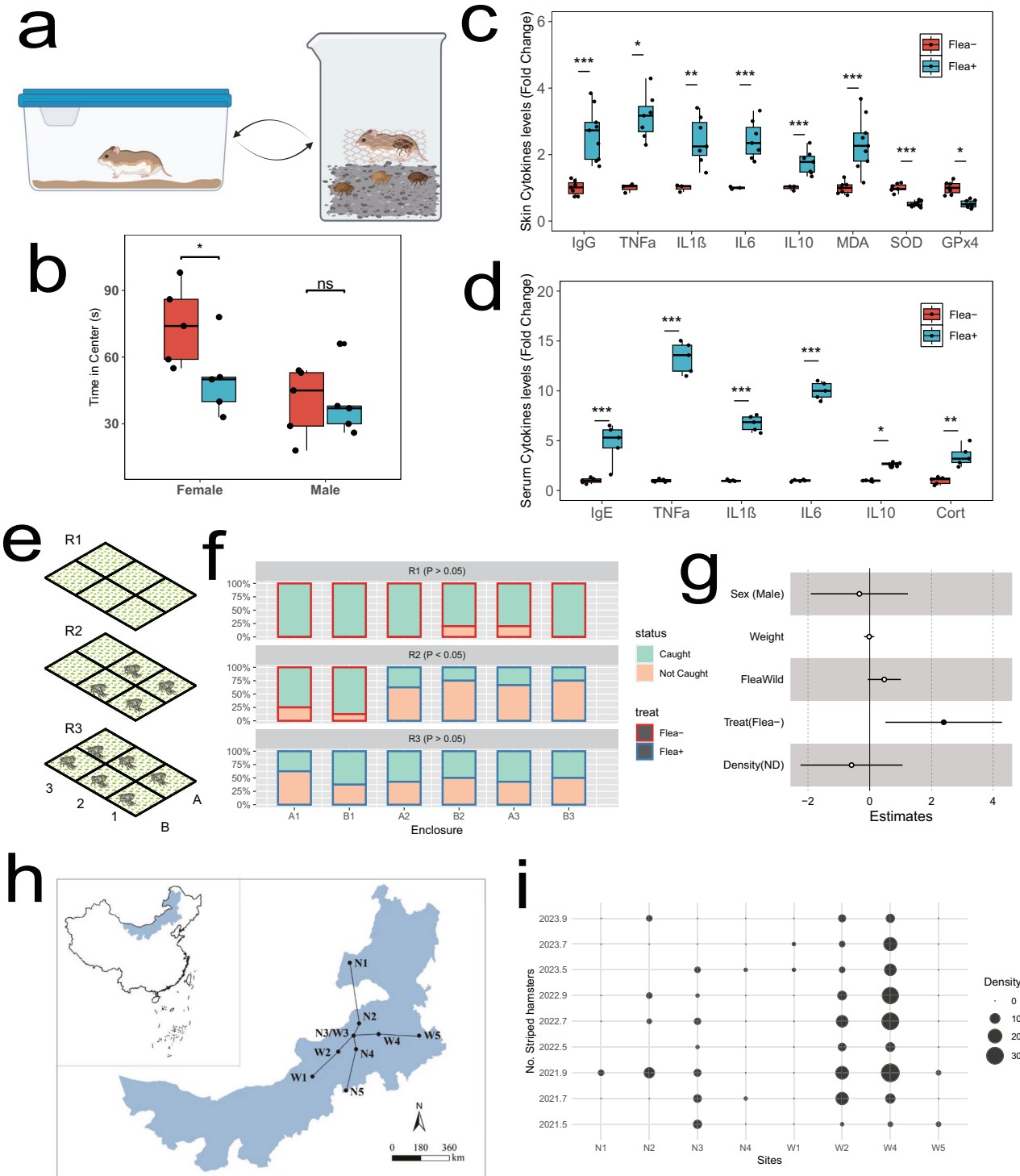

bitten striped hamsters would spend more time near the borders of the enclosure and be more likely to escape, resulting in a higher non-capture rate. On the contrary, a hamster that explores more will end up in the center of the enclosure and is more likely to be recaptured. Before starting the experiment, we removed any existing rodents and ectoparasites from the enclosures and confirmed that there were no significant differences in plant diversity between the enclosures (Supplementary Fig. 10f). We conducted three rounds of enclosure infection experiments with striped hamsters: no fleas were introduced in any enclosures-R1, fleas were introduced in some enclosures-R2, and fleas were introduced in all enclosures-R3 (Fig. 6e and Supplementary

Fig. 10g). In the first round, most hamsters remained in their respective enclosures, with few escaping. In the second round, the flea-infested hamsters exhibited a higher non-capture rate. In the third round, we observed a significant increase in non-capture rates compared to the first round, but no difference between the enclosures (Fig. 6f). Flea bites significantly increased the non-capture rate of striped hamsters. However, other factors, particularly host sex and enclosure density, did not affect the likelihood of escaping the enclosure (Fig. 6g and Supplementary Fig. 10h). Overall, the semi-natural enclosure experiment demonstrated that flea bites reduce host exploratory behavior, thereby decreasing the likelihood of the host leaving its natal habitat.

**Fig. 6 | Striped hamsters exposed to flea bites exhibited reduced exploratory behavior in both indoor infection and outdoor enclosure experiments.** **a** Schematic diagram of laboratory infection in striped hamsters, created in BioRender. Liu, P. (2025) https://BioRender.com/ps1cm2y. **b** Open field test for striped hamster, n = 5 each group, tested using a Generalized Linear Model (GLM). Thick bars indicate the interquartile range (IQR) around the median, and whiskers represent 1.5 times the interquartile range (maxima: Q3 + 1.5 × IQR, minima: Q1 − 1.5 × IQR). Each black dot represents an individual. Asterisks indicate the significance. ($P_{female}$ = 0.03, $P_{male}$ = 0.26). **c** Cytokine levels in striped hamster skin presented in boxplot where red is Flea− (n = 3–7) and blue is Flea+ (n = 7–9). Two-sided $t$ test was used. Thick bars indicate the interquartile range (IQR) around the median, and whiskers represent 1.5 times the interquartile range (maxima: Q3 + 1.5 × IQR, minima: Q1 − 1.5 × IQR). Each black dot represents an individual. Asterisks indicate the significance. **d** Cytokine levels in striped hamster serum presented in boxplot where red is Flea− (n = 5) and blue is Flea+ (n = 5). Two-sided $t$ test was used. Thick bars indicate the interquartile range (IQR) around the median, and whiskers represent 1.5 times the interquartile range (maxima: Q3 + 1.5 × IQR, minima: Q1 − 1.5 × IQR). Each black dot represents an individual. Asterisks indicate the significance. **e** Schematic diagram of the three rounds of enclosure field experiments under natural conditions. **f** Bar graph of the capture rate of striped hamsters released into the enclosures. **g** Forest plot illustrating that Flea+ (n = 72) striped hamsters had a significantly higher escape rate than Flea− (n = 62) hamsters, tested using a generalized linear mixed model (GLMM). The x-axis represents the coefficients of various predictor variables. Solid circles indicate statistically significant results, while hollow circles represent non-significant results. Data are presented as mean values ± SEM. **h** Field sampling sites for striped hamsters, selected based on temperature and precipitation gradients. Created using the open-source R packages ggplot2 (MIT License). **i** The number of striped hamsters captured during each sampling event at each site, with no capture records for striped hamsters at site N5. Source data are provided as a Source Data file. ***$p$ < 0.001, **$p$ < 0.01, *$p$ < 0.05.

## Reduced occupancy in Flea+ host

We used individual-based mechanistic models that integrate host habitat preferences, demographic indicators, and migration patterns to simulate the impact of fleas on the occupancy of striped hamsters (Supplementary Fig. 11). We constructed two types of host habitat preferences based on static and dynamic climate models, as we aim to simulate long-term changes (e.g., 100 years), which necessitates considering the impact of climate change. Numerous studies have shown that climate change significantly influences species distribution[32,33]. We determined the demographic data for striped hamsters through field sampling. Over three years and nine sampling events, the highest number of striped hamsters was captured in W4 sample site, with a single event yielding up to 34 individuals (Fig. 6h, i). For more details on the demographic data, see the Methods section.

We constructed current and future habitat suitability for striped hamsters. We used climate variables and elevation as predictors (Supplementary Fig. 12) and gathered 562 occurrence records from literature, GBIF, and field surveys (Fig. 7a and Supplementary Table 1). We excluded the SRE and ANN models due to their poor accuracy (Fig. 7b). The remaining 10 models were integrated to estimate the current and future global and Chinese distribution of suitable habitats for striped hamsters. The suitable habitats are primarily located in central Eurasia, particularly in northern China, Mongolia, and the Russian Far East, with some suitable areas in central North America and Alaska (Supplementary Fig. 13a). In China, these habitats are mainly distributed in the northeast, north, and parts of the northwest (Fig. 7c). Under future climate change scenarios, suitable habitats are predicted to be lost in central North America and parts of Central Asia (Supplementary Fig. 13b), while in China, some habitats in the Yangtze River basin will be lost, and the suitability index will increase in the northwest (Fig. 7d). To account for the impact of parasite-mediated host dispersal on population dynamics under climate change, we selected three sites for simulation, representing scenarios where the suitability index remained constant, decreased, or increased under future climate conditions (Fig. 7e–h).

Simulations across the three study sites consistently showed that flea infestation significantly reduced the density of striped hamsters. However, dynamic climate effects on density were only significantly detected at site S2 (Fig. 8a–c). In the static climate model, the host distribution range was significantly lower in the Flea+ group across all three sites, particularly in areas with low suitability (Fig. 8d–f). Based on dynamic climate model simulations, over a 100-year period, the distribution range of the Flea+ group remained consistently lower than that of the Flea− group in each decade (Fig. 9a–f).

## Discussion

Parasites employ a variety of manipulation strategies to enhance their fitness by altering host behavior[34–37]. The phenomenon of parasite-induced host manipulation was first observed in amphipods infected by larval acanthocephalan parasites, which exhibited abnormal behaviors and altered coloration, increasing their susceptibility to predation by the parasite's next host[38]. This discovery led to widespread interest and the documentation of similar phenomena across numerous host-parasite systems. While most reported cases involve endoparasites residing within the host, our study demonstrates that ectoparasites can also modulate host behavior, specifically by reducing exploratory tendencies. This behavioral alteration may limit host dispersal and protect the parasite from environmental pressures. The ability of parasites to modify host behavior has been documented across diverse taxa. For instance, some parasites influence reproductive behaviors or alter host morphology to enhance their own transmission[39,40]. A well-known example is the nematode that infects mayflies, feminizing male hosts to ensure they return to water, thereby facilitating parasite reproduction[41]. Similarly, the parasitic wasp larva manipulates its spider host to construct a protective web structure before the larva emerges, shielding it from environmental hazards[42]. Notably, while most known cases of parasite-induced behavioral change involve endoparasites, such as *Toxoplasma gondii*, which enhances host exploratory behavior to promote transmission, our study provides rare evidence of an ectoparasite exerting direct behavioral influence on a mammalian host[43–45].

Research on the molecular mechanisms of host manipulation by parasites remains limited, with *T. gondii* being the most extensively studied example. *T. gondii* infection causes intermediate hosts to exhibit bold behaviors. For example, infected rats become attracted to cat urine, a stark contrast to their natural aversion[45]. Similarly, three decades of field observations have shown that *T. gondii*-infected hyena cubs approach lions more closely than their uninfected counterparts, resulting in higher mortality rates from lion predation[46]. Upon breaching the blood-brain barrier, *T. gondii* invades neurons and forms cysts, provoking microglial activation and the release of inflammatory mediators, such as TNFα, IL-6, and IL-1β, which alter neuronal function and host behavior[47]. In contrast to the well-documented effects of *T. gondii*, our study identifies flea bite-induced microglial activation and disrupted GABAergic neuron differentiation in the PFC as potential drivers of reduced exploratory behavior. This highlights a novel mechanism by which ectoparasites can directly influence host behavior without relying on resource reallocation. Unlike previous studies focusing on endoparasites, our findings highlight how ectoparasite infection influences microglia-neuron interactions in a region-specific manner, disrupting neural circuits governing exploratory behavior[48–50]. While we observed metabolic and expression abnormalities in several brain areas, the PFC stands out as the primary region for emotion regulation. Neuronal dysfunction in the PFC may heighten neural excitability, potentially leading to anxiety-like behaviors, further linking flea bite-induced neuroinflammation to behavioral alterations. Comparable findings in microbiome research suggest

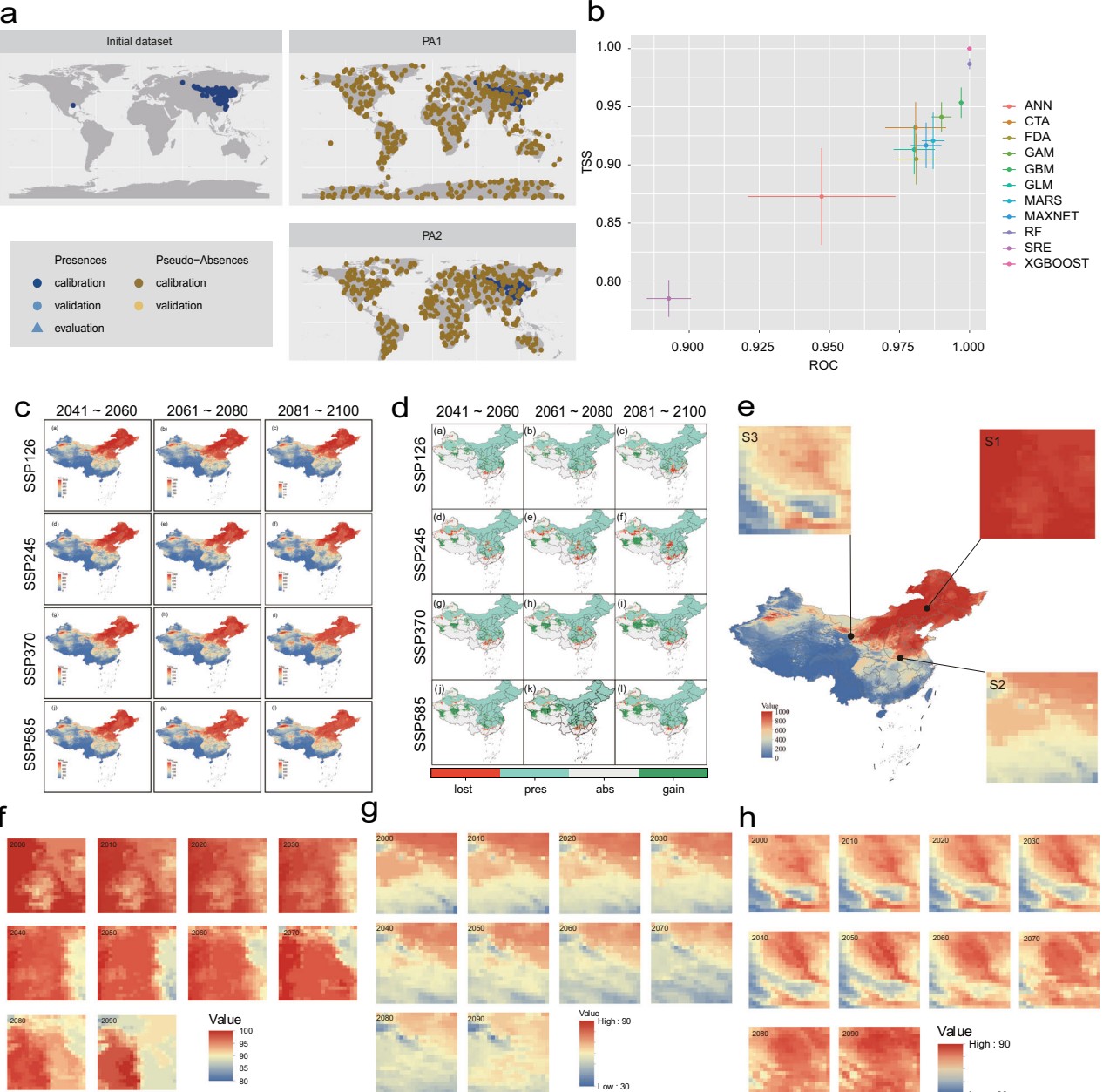

**Fig. 7 | The suitable habitat distribution of striped hamsters is based on species distribution modeling.** These figures were created using the open-source R packages bomod2 and ggplot2, both licensed under the MIT License. **a** Actual distribution points and pseudo-absence points for striped hamsters. **b** Evaluation of prediction performance for 11 models in the Biomod2 package, based on TSS and ROC metrics. **c** Habitat suitability distribution for striped hamsters in China under future climate scenarios. **d** Changes in habitat suitability for striped hamsters in China under future climate scenarios. **e** Suitable habitat distribution of striped hamsters in China and the selection of three simulated sites. **f–h** Changes in habitat suitability index for striped hamsters over 100 years at three sites used for mechanistic modeling: S1 ((**f**), high suitability with stable index), S2 ((**g**), moderate suitability with decreasing index), and S3 ((**h**), moderate suitability with increasing index).

that colonization by 4EP-producing bacteria reduces neuronal myelination, contributing to anxiety-like phenotypes[23]. This parallel underscores the broader impact of neuroimmune interactions on behavior. Although BBB disruption-induced microglial activation remains our primary hypothesis, we cannot exclude the role of the vagus nerve in transmitting peripheral inflammatory signals, particularly in response to histamine release[27]. Future investigations will aim to delineate the precise molecular pathways linking flea bite-induced peripheral inflammation to microglial activation and behavioral shifts in the PFC.

We aim to translate the impact of fleas on mouse personality into differences in host dispersal rates through enclosure experiments, a novel approach for mechanism modeling. Given the challenges in assessing dispersal impacts on population dynamics, we hope our research provides a fresh perspective for ecologists by integrating historical data with advanced modeling techniques to uncover more natural phenomena.

Understanding these behavior modifications has critical implications for controlling parasite spread, protecting human and animal health, and maintaining ecological balance, highlighting the

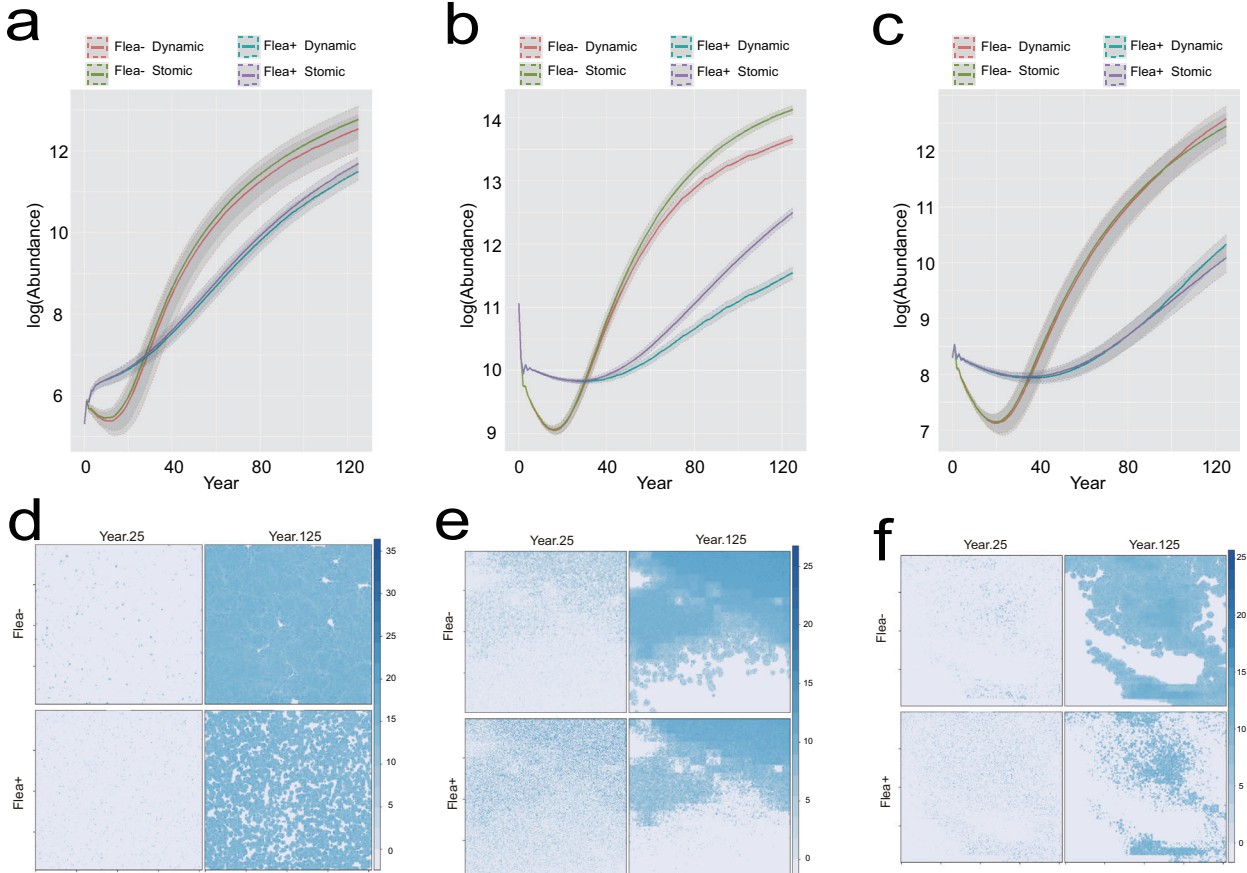

**Fig. 8 | Modeling density and range shifts in striped hamsters based on mechanistic models. a–c** The simulation results of striped hamster densities at the three sites under future climate conditions, S1 (**a**), S2 (**b**), S3 (**c**). **d–f** Fitted plots showing changes in the distribution range of striped hamsters over a 100-year period under static climate conditions, comparing Flea− and Flea+ scenarios ((**d**) for S1, (**e**) for S2, (**f**) for S3).

interconnectedness of ecological and public health systems under the One Health framework.

## Methods

### Ethical statement
All animal procedures strictly adhered to the ARRIVE guidelines and relevant regulations. Animals were used exclusively for this study and euthanized using anesthetics. Their housing, handling, and disposal complied with ethical standards and approved protocols. This study was approved by the Ethical Committee of the National Institute for Communicable Disease Control and Prevention, Chinese Center for Disease Control and Prevention (No. 2022-027).

### Survival comparisons between two flea species
Three-week-old female C57BL/6J mice, purchased from SiPeiFu biotechnology Co. Ltd. (Beijing, China), were used for flea infection experiments. We compared the survival rates of two flea species maintained on Kunming mouse blood in the laboratory: *X. cheopis*, primarily hosted by commensal or semi-commensal rodents, and *C. felis*, which primarily infests cats. Both flea species have been adapted to mice in our laboratory for over 10 years and have been tested to ensure they do not carry pathogens. Previous research demonstrated that the greater the phylogenetic distance between host species, the more stress the host experiences[14]. However, increased stress on the fleas can also trigger a stronger defensive response from the host, leading to higher parasite mortality[15]. Twenty-four mice were divided into two groups, each infected with one of the two flea species. The

mice were housed three per cage, with four cages per group, and 150 fleas were introduced into each cage. After 1 week, the survival rates of both attached and free fleas were assessed.

### Experiment design of mice infection
Three-week-old female C57BL/6J mice were housed in groups of three per plastic cage, with autoclaved corncob bedding. The mice were fed mouse breeding feed (SPF-F01-001, SPF Biotech, Beijing, China) and had free access to water. The housing conditions were maintained at $23 \pm 0.5\,°C$ with $50 \pm 5\%$ humidity, under a 14/10-h light/dark cycle. After a one-week acclimation period, the mice were infected with 150 fleas per cage (Flea+ group). The control group (Flea− group) was exposed to 150 heat-killed fleas to eliminate any visual aversion effects that could influence the results. The cages were covered with fine mesh to prevent flea escape, and the infection was refreshed weekly with new fleas. Four weeks later, behavioral tests were conducted to assess the impact of flea bites on the mice's exploratory behavior.

### Wild striped hamsters husbandry
We conducted an infection experiment with striped hamsters in Xilinhot, Inner Mongolia, China. Striped hamsters, a dominant species in the Xilingol grassland, live solitarily, making their ectoparasite load a reliable indicator of individual infection levels. The hamsters were live-trapping using Sherman traps in West Ujimqin Banner (W4, Fig. 3f). Each morning, the traps were checked, and the captured hamsters were brought back to the laboratory for weighing. Fleas were combed from their fur to establish experimental populations. Furthermore, their ear tissue samples were collected for MHC analysis. The hamsters were housed

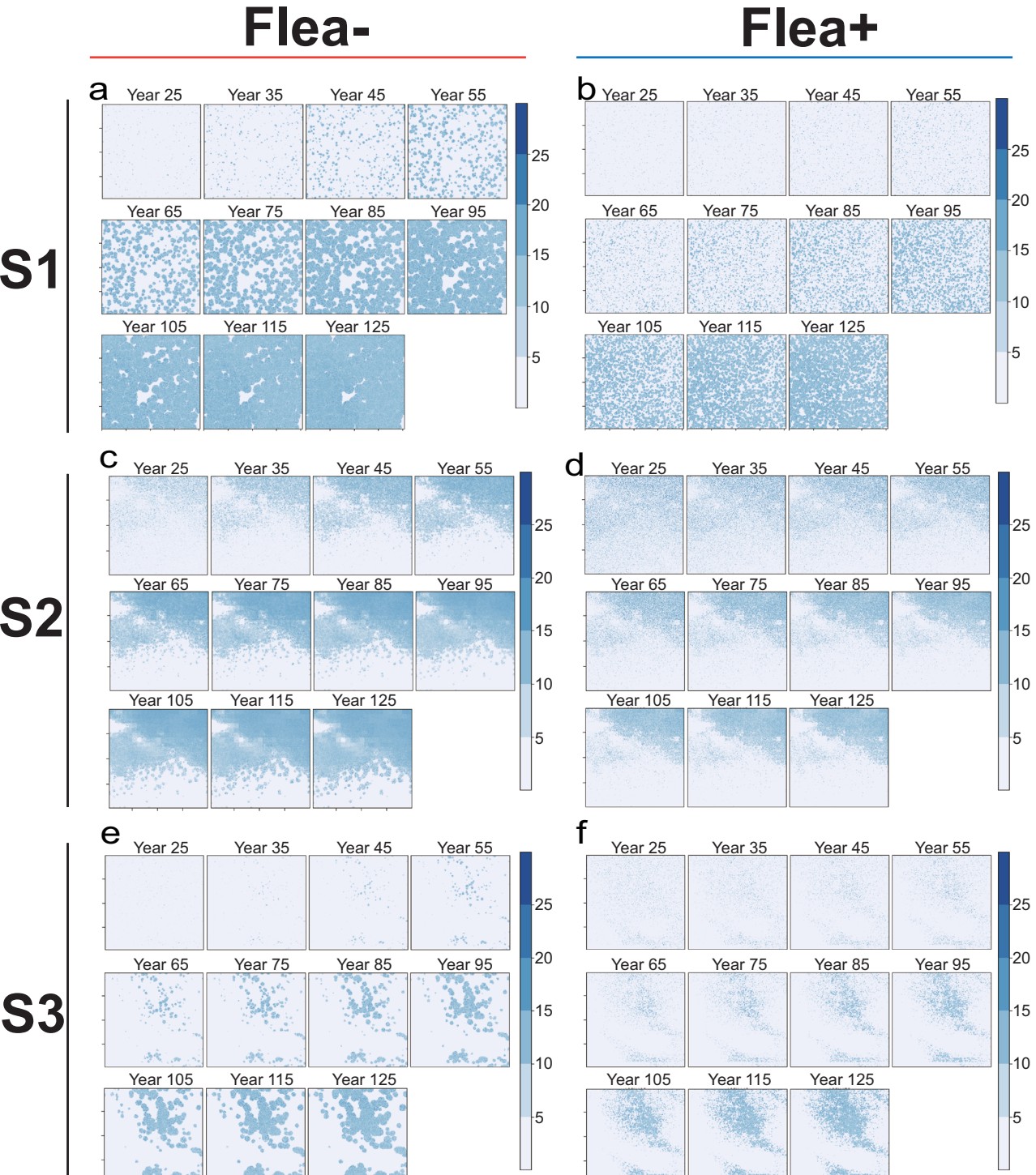

**Fig. 9 | Modeling range shifts in striped hamsters based on mechanistic models under dynamic climate conditions. a, b** Fitted plots showing the changes in the distribution range of striped hamsters over a 100-year period under dynamic climate conditions for S1 sites, comparing Flea− (**a**) and Flea+ (**b**) scenarios. **c, d** Fitted plots showing the changes in the distribution range of striped hamsters over a 100-year period under dynamic climate conditions for S2 sites, comparing Flea− (**c**) and Flea+ (**d**) scenarios. **e, f** Fitted plots showing the changes in the distribution range of striped hamsters over a 100-year period under dynamic climate conditions for S2 sites, comparing Flea− (**e**) and Flea+ (**f**) scenarios.

individually in cages under natural light conditions. To eliminate interference from wild-caught fleas, the hamsters were kept in cages for at least four weeks before being used in the infection experiment.

**Wild flea husbandry**
The primary flea species in W4 is the *Neopsylla bidentatiformis* based on morphological characteristics (Supplementary Fig. 10c), which is widely distributed across the Palearctic region, including Northeast Asia and Central Asia. This flea commonly infests various rodent and lagomorph species, including striped hamsters, *Meriones unguiculatus, Citellus dauricus*. We reared the fleas collected in glass containers lined with fine sand, pig blood powder, and yeast. These containers were kept at a constant temperature of 25 ± 1 °C and relative humidity of 85 ± 10%, with wild-caught striped hamsters providing the blood meals for the fleas.

## Indoor infection experiment with striped hamsters

The flea burden among the hamsters showed significant individual variation, influenced by factors such as sex, body weight, and the number of MHC alleles (Supplementary Fig. 10a). Striped hamsters were grouped by sex, weight, and field infection status. Given the nocturnal nature of striped hamsters, as in natura infection occurs when they are resting, and their solitary behavior, those in the Flea+ group were placed in flea-infested glass tanks at 7:00 A.M. each day, while those in the Flea− group were placed in flea-free tanks. At 7:00 P.M., all hamsters were returned to their respective cages. During the infection period, the hamsters were restrained in cages to prevent fighting and minimize the disruption of flea populations. After four weeks, behavioral tests were conducted. Afterward, the striped hamsters were sacrificed for tissue collection, including skin and blood, to analyze cytokine levels. We did not collect brain tissue due to restrictions of field conditions.

## Field enclosure experiment with striped hamsters

We conducted a field enclosure experiment in the Xilingol grassland of Inner Mongolia, China, where the dominant rodent species are mongolian gerbils, daurian ground squirrels, and striped hamsters. The goal was to assess the impact of flea bites on the exploratory behavior of striped hamsters in a natural environment.

We constructed six 30 m × 30 m enclosures using galvanized iron sheets, with each side formed by six 5 m-long sheets. The enclosures extended 70 cm above ground and 50 cm below, with small gaps (3 cm) between the sheets. Based on previous experience, we believed these measures would prevent external interference while allowing some movement of striped hamsters out of the enclosures. We hypothesize that the enclosures simulate a large open field, where striped hamsters subjected to flea bites will exhibit reduced exploratory behavior. As a result, these hamsters are expected to spend more time near the edges of the enclosures, increasing their likelihood of finding gaps and leaving the enclosure.

Before the experiment began, we cleared the enclosures of resident rodents and parasites and conducted a plant diversity survey. The experiment was conducted in three rounds, with all released striped hamsters tagged with electronic chips. In the first round, only striped hamsters were introduced into the enclosures, with no fleas present. In the second round, new striped hamsters were reintroduced, and some enclosures were subjected to flea infection at an intensity of 7 fleas per hamster. The fleas were introduced by placing the hamsters in a 5 L glass beaker with pre-separated fleas, allowing the fleas to attach before releasing the hamsters into the center of the enclosure. We employed the same reintroduction protocol for Flea− hamsters, with no flea infection. Additionally, we tested whether this movement was density-dependent. In the third round, all hamsters were reintroduced and infected with fleas. After 4 weeks of infection, trapping was conducted over four days. Each captured hamster was scanned for its chip, and its capture location was recorded to compare the rates of leaving the enclosures.

## MHC genotyping

The sequencing method for MHC genes followed the procedure described in our previous publication[51]. DNA was extracted from ear tissue using the Qiagen DNeasy Blood and Tissue Kit, and the exon 2 region of the MHC class II DRB gene was amplified. Each individual was assigned a unique barcode, and sequencing was conducted using Illumina's next-generation sequencing technology. The sequencing data were analyzed using Amplisas[52] to infer MHC allele sequences.

## Plant diversity survey

We assessed plant diversity within the enclosures by surveying plant cover, biomass, and density. In each enclosure, four 1 m × 1 m quadrats were randomly selected to measure the cover percentage, density (number of rooted plants per quadrat), and biomass (g/m²) of each plant species. Each 1 m × 1 m quadrat was further divided into 16 sub-quadrats (0.25 m × 0.25 m), and in one sub-quadrat, the above-ground parts of each plant species were clipped at ground level. The location of the clipped sub-quadrat varied in each sampling session to minimize the impact of clipping. The clipped plants were sorted by species, temporarily stored in paper envelopes, and then dried in an oven at 65 °C until a constant mass was reached. We calculated plant diversity using the Shannon–Wiener index and the Simpson index. Differences in diversity among enclosures were assessed using the Kruskal–Wallis test. A $p$-value of <0.05 was considered statistically significant.

## Open-field test

The open-field tests were conducted in a 50 × 50 cm² white acrylic arena, with recordings made using an overhead camera and analyzed with EthoVision XT 10 software (Noldus Information Technology, Leesburg, VA, USA). Prior to testing, the arena was disinfected with 75% alcohol and allowed to evaporate. The mice/striped hamster were then introduced into the arena and allowed to explore for 3 (mice)/5 (striped hamster) minutes while being tracked. We analyzed the total distance traveled, the number of entries into the central 25 × 25 cm² area, and the time spent there. The open field tests for mice were conducted in the afternoon (2:00 P.M.–7:00 P.M.), while the open field tests for striped hamsters were conducted in the evening (8:00 P.M.–11:00 P.M.), with illumination provided by 15 W red lights and recordings made using an infrared camera.

## Elevated plus maze (EPM)

The EPM test was conducted in a maze with 25 cm × 5 cm arms and a 5 cm × 5 cm central area. An overhead camera recorded the sessions, and the data were analyzed using EthoVision XT 10 software (Noldus Information Technology, Leesburg, VA, USA). Prior to testing, the maze was disinfected with 75% alcohol and allowed to evaporate. The mice were then introduced into the maze and allowed to explore for 3 min while being tracked. We analyzed the number of entries and the time spent in both open and closed arms. Mice that fell off or jumped from the maze during the test were excluded from the dataset.

## Autoradiography brain mapping

To identify the brain regions affected by flea bites, we obtained PET/CT images of Flea− and Flea+ mice. The mice were fasted and deprived of water for 12 h prior to the imaging procedure. A radioactive tracer, 18F-FDG (0.1 ml, 7.40–9.25 MBq, provided by Atomic Hi-Tech Co., Ltd., Beijing, China), was injected into the animals via the tail vein. After 40 min of free movement, the mice were placed in an induction chamber and anesthetized with isoflurane (3.0% isoflurane in 2.0 L/min airflow). They were then immediately transferred to the scanning bed, where the airflow was adjusted to 0.8–1.0 L/min with 1.0–1.5% iso-flurane, and static microPET/CT imaging was performed using the Inveon PET/CT system (Siemens, Berlin, GER). The PET scan duration was 10 min. CT scans were conducted in two bed positions under the "magnification low" mode, with the following specific parameters: projection: 180; binning: 4 × 4; transaxial field of view: 53.9 mm; axial scanning length: 134 mm; voltage: 80 kV; current: 500 μA. The mice remained anesthetized throughout the procedure, with continuous monitoring of body temperature and respiratory rate. For each animal, CT and PET images were co-registered in a single session using the same bed, ensuring that the animal was not moved between the two scans.

DICOM images produced with PET/CT were converted to NIfTI format using dcm2niix (https://github.com/rordenlab/dcm2niix). To reduce memory usage and ensure that normalization and registration algorithms focused on the relevant regions, an ad-hoc script was used to crop the images around the head. Following this, optimal values for the SPM4mouse (v 12) routines were set using the set_mouse_def

function from a custom package[17]. Each PET image was then co-registered to its corresponding CT image using the "Coregister (Estimate)" tool in the SPM GUI, a process that could be batched for efficiency. The origin of each PET image was adjusted, and the images were flipped as needed to match the orientation of the template, with the same adjustments applied to the corresponding CT images. Subsequently, a CT-based normalization script was executed. Finally, regions of interest (ROI) were extracted from Allen Brain Reference Atlases–Major divisions. Abbreviations used in the figures correspond to the following brain regions: Anterior Cingulate Left (ACL-L), Anterior Cingulate Right (ACL-R), Auditory Cortex Left (ACT-L), Auditory Cortex Right (ACT-R), Basal Ganglia Dorsal Left (BG-DL), BG-DR, Basal Ganglia Ventral Left (BG-VL), Basal Ganglia Ventral Right (BG-VR), Cerebellum (CB), Hindbrain (HB), Hippocampal Left (HIP-L), HIP-R, Hypothalamus Left (HY-L), Hypothalamus Right (HY-R), Somatomotor Left (SM-L), Somatomotor Right (SM-R), OLF-L, OLF-R, Orbital Left (ORB-L), Orbital Right (ORB-R), Somatosensory Left (SS-L), SS-R, Thalamus Left (TH-L), TH-R, Visual Cortex Left (VC-L), Visual Cortex Right (VC-R), MidBrain (MB). Significant correlations are indicated by asterisks.

## Transcriptome analysis

Following euthanasia with sodium pentobarbital, the mice were dissected. The hair on the back was removed, and the skin was sampled. The target brain region was carefully isolated on a cold operating table, and the tissue was immediately immersed in RNAlater (AM7024, Invitrogen, Carlsbad, CA) for preservation at 4 °C. After six hours, the samples were transferred to −80 °C for long-term storage. RNA was subsequently extracted using QIAzol Lysis Reagent (79306, Qiagen China, Shanghai, CN) according to the manufacturer's instructions.

For library construction, the Illumina Standard mRNA Prep was employed. mRNA was isolated using magnetic beads with Oligo (dT) that specifically bind to polyA tails. The isolated mRNA was fragmented into approximately 300 bp pieces using a fragmentation buffer. These fragments were then reverse transcribed into cDNA using random primers. The resulting double-stranded cDNA was end-repaired, adenylated at the 3′ ends, and ligated to Y-shaped adapters. The adapter-ligated cDNA was purified, size-selected, and amplified via PCR to create the final library. After quantification using Qubit 4.0, sequencing was performed on the Illumina NovaSeq Xplus platform.

Raw sequencing data underwent quality control using the fastp software, which removed adapter sequences, trimmed low-quality bases from the 3′ ends, and discarded reads with a high proportion of N bases or lengths below 20 bp. Cleaned data were then aligned to the reference genome using HiSat2 (v 2.2.1, https://daehwankimlab.github.io/hisat2/), and transcript quantification was performed with RSEM (v 1.3.3, https://deweylab.github.io/RSEM/). DEGs between samples were identified using DESeq2 (v 1.42.0, https://github.com/thelovelab/DESeq2)[53], with the criteria of FDR < 0.05 and |log2FC| ≥ 1. These DEGs were annotated using the GO database (v 2023.07)[54,55], and functional enrichment analysis was performed with Goatools using Fisher's exact test. Additionally, GSEA was conducted with the MSigDB database (v 2023.2, https://docs.gsea-msigdb.org/#MSigDB/Release_Notes/MSigDB_Latest/), employing the Signal2Noise ranking algorithm. To further understand the functional implications of these genes, cell type enrichment analysis was conducted using the pSI package (v 1.1) in R, with cell type markers from Zhang et al.[56].

## Neurotransmitter metabolomics analysis

The 46 neurotransmitter standards were accurately weighed and prepared as single standard master mixes with methanol or water. Measure an appropriate amount of each master batch to make a mixed standard, dilute it with water to a suitable concentration, and make a working standard solution. The isotope standards (Trp-D5, Glu-13C5) were weighed in appropriate amounts, and the single master solution

was prepared with methanol. The mixed isotope standards were made by measuring the appropriate amount of each master batch of the isotope standards and prepared with water to the concentration of 5000 ng/mL and 1000 ng/mL of the mixed isotope internal standard solutions.

We conducted a neurotransmitter analysis on 3 and 4 independent biological replicates of mice exposed to flea bites for four weeks, using untreated mice as controls. Each biological replicate was a pooled sample from three brain tissues. Weigh 50 mg of sample accurately in a 2 mL EP tube, add steel beads, add 20 μL of isotope internal standard (5000 ng/mL) and 480 μL of 80% methanol aqueous solution, put into a tissue grinder, grind at 60 Hz for 6 min, then stand at −20 °C for 30 min, centrifuge at 4 °C for 15 min at 14000 rcf, take 250 μL of supernatant and put into a freeze concentrated centrifuge. After spin-drying, 100 μL of water was added and vortexed to re-dissolve, and the supernatant was centrifuged at 4 °C for 5 min at 14000 rcf. The supernatant was taken into the injection vial and used as the sample for low concentration detection. Add 240 μL of water and vortex for 30 s. Take 50 μL of the diluted sample, add 50 μL of 1000 ng/mL isotope internal standard, vortex for 30 s and centrifuge at 4 °C for 14000 rcf for 2 min, and take the supernatant into the injection vial as the sample for high concentration.

Sample analysis was performed using LC-ESI-MS/MS (UHPLC-Qtrap) on a ExionLC AD system coupled with a QTRAP® 6500+ mass spectrometer (AB Sciex, USA) at Majorbio Bio-Pharm Technology Co. Ltd. (Shanghai, China) Briefly, samples were separated by an Waters HSS T3 (2.1 × 100 mm,1.8 μm) thermostated at 35 °C. Separation of the metabolites was achieved at 0.3 mL/min flow rate with a mobile phase as a gradient consisted of 0.1% Formic acid in water (solvent A) and 0.1% Formic acid and Acetonitrile in water(solvent B), 15 min of total chromatographic separation. The solvent gradient changed according to the following conditions:hold at 0% B, 0 and 1 min; from 0% to 5% B, 1–3 min; from 5% to 10% B, 3–5 min; from 10% to 15% B, from 5 to 6 min; hold at 15% B, 6 and 7 min; from 15% to 60% B, 7–10 min; from 60% to 100% B,10 and 11 min; hold at 100% B,11–12 min; From 100% to 0% B,12–12.01 min; hold at 0% B, until the end of separation. During the period of analysis, all these samples were stored at 4 °C.

QC (Quality control) is a mixture of samples or a moderate concentration of mixed standard solutions, mainly used to assess the stability of the analytical system, in the process of instrumental analysis, every 5 samples inserted a QC sample to examine the reproducibility of the entire analytical process. The RSD of the stability of each target should be less than 15%.

After the completion of the upload, the LC-MS raw data were imported into AB Sciex quantitative software OS using default parameters for automatic identification and integration of each ion fragment, with the aid of manual inspection. A linear regression standard curve was plotted using the ratio of the mass spectrometric peak area of the analyte to the peak area of the internal standard as the vertical coordinate and the concentration of the analyte as the horizontal coordinate. The sample concentration is calculated by substituting the ratio of the mass spectral peak area of the sample analyte to the internal standard peak area into the linear equation and calculating the concentration result.

Metabolomic analysis was conducted using MetaboAnalystR 6.0[57]. Metabolite concentrations were normalized before further analysis. We used unsupervised PCA and supervised PLS-DA to cluster the results and evaluate the reproducibility of flea bite effects. Differential metabolite analysis was performed using Fold Change (FC) Analysis with a threshold of 1.5 to detect as many differential metabolites as possible. Identified metabolites were annotated using the KEGG Compound database (v 2023.9, http://www.kegg.jp/kegg/compound/) and the KEGG Pathway database (v 2023.9, http://www.kegg.jp/kegg/pathway.html). Significantly enriched pathways were identified using a hypergeometric test with P-values calculated for the given list of metabolites.

## Immunofluorescence analysis

The mice were perfused through the cardiovascular system with PBS, followed by 4% paraformaldehyde. The brains were then extracted and fixed in 4% paraformaldehyde at 4 °C for one day. Following fixation, the brain tissues were dehydrated using a graded ethanol series, cleared with xylene, and then embedded in paraffin at 60 °C. Paraffin blocks were sectioned using a Leica microtome (RM 2016), and the sections were subsequently deparaffinized. Antigen retrieval was performed using 0.01 M Tris-EDTA, after which the sections were blocked with normal goat serum at room temperature for 30 min. The sections were incubated with the primary antibody overnight at 4 °C in a humid chamber. After three washes with TBST (3 min each), the sections were dried with absorbent paper and incubated with a diluted fluorescent secondary antibody at 37 °C for 1 h in a humid chamber. The sections were then washed four times with TBST (3 min each). To stain the nuclei, DAPI was applied, and the sections were incubated in the dark for 5 min. Excess DAPI was washed off with TBST (four washes, 5 min each). After drying the sections with absorbent paper, they were mounted with an anti-fade mounting medium and observed under a fluorescence microscope for image capture.

For immunofluorescence staining, the following primary antibodies were used with their respective dilutions: rabbitanti-IBA1 (1:500, 10904-1-AP, Proteintech), mouse anti-NeuN (1:100, ab104224, Abcam), mouse anti-TUNEL (1:100, A112-03, Vazyme), rabbit anti-PSD95 (1:100, ab238135, Abcam), rabbitanti-HOMER1 (1:100, A4302, Abbiotec), mouse anti-Synaptophysin (1:100, ab8049, Abcam), rabbit anti-GAD65/67 (1:100, ab183999, Abcam), rabbitanti-GABRG2 (1:100, 14104-1-AP, Proteintech), and rabbitanti-VGLUT1 (1:100, 55491-1-AP, Proteintech). The fluorescently conjugated secondary antibodies used were CY3-conjugated goat anti-mouse IgG (1:100, BA1031, BOSTER) and CY3-conjugated goat anti-rabbit IgG (1:100, BA1032, BOSTER).

## Flow cytometry

The PFC was carefully rinsed with PBS and then immersed in PBS containing 5% FBS. To achieve red blood cell lysis, the tissue was incubated with 10 ml of 0.25% Trypsin-EDTA at 37 °C for 10 min, with gentle mixing and pipetting to dissociate the cells. The cell suspension was filtered through a Falcon® 70 μm Cell Strainer, and any remaining tissue was gently ground on the strainer and rinsed with RPMI 1640 containing 5% FBS. The filtration was repeated twice, followed by centrifugation at $300 \times g$ for 5 min. The supernatant was discarded, and the pellet was re-suspended in sterile PBS to achieve a cell concentration of $2 \times 10^7$ cells/ml, then stored at 4 °C until further use.

For each sample, 100 μl of the cell suspension (approximately $2 \times 10^6$ cells) was transferred to the bottom of a flow cytometry tube. The FVS700 dye, previously dissolved in DMSO and diluted 1:1000 in PBS, was added to each sample tube at 1 ml, followed by incubation at room temperature for 15 min. After staining, the cells were washed with 2 ml PBS containing 1% FBS (referred to as PBS wash buffer), and centrifuged at $350 \times g$ for 5 min; the supernatant was discarded. The cells were then stained with 1 μl of CD45 FITC and 2 μl of CD11b PE-Cy7, mixed, and incubated at room temperature in the dark for 20 min.

Appropriate controls were included for each sample, including unstained controls, single-stained controls, and isotype controls, to adjust instrument settings and to determine non-specific binding. After incubation, the cells were washed again with 2 ml PBS wash buffer, centrifuged at $300 \times g$ for 5 min, and resuspended in 0.5 ml PBS wash buffer for analysis. Flow cytometry data were acquired using the Cytek NL-CLC3000 flow cytometer with SpectroFlo 1.0 software (https://cytekbio.com/pages/spectro-flo), and the resulting flow cytometry standard (FCS) data were exported in FCS 3.0 format[58].

## Western blot

A tissue sample approximately the size of a mung bean (25–30 mg) was placed in a 2 ml EP tube and mixed with 30 μl of RIPA lysis buffer containing protease inhibitors (1 mM PMSF and 1 mM sodium orthovanadate). The tissue was homogenized using magnetic beads. Protein concentration was quantified and standardized using a BCA Protein Assay Kit. The proteins were then denatured by boiling for 10 min, separated by PAGE gel electrophoresis, and transferred onto a PVDF membrane. The membrane was immediately blocked with blocking buffer. GAPDH was used as the loading control. The images of the same blot were captured under specific excitation wavelengths for quantification and visualization of the target proteins. We used Image-Pro Plus 6.0 software to perform grayscale analysis on each sample's band intensity to quantify protein concentration. The membrane was probed with the following primary antibodies: mouse anti-GAPDH (1:50000, 60004-4-Ig, Proteintech), mouse anti-IBA1 (1:1000, Ab283346, Abcam), rabbit anti-GAD65/67 (1:1000, ab183999, Abcam), and rabbit anti-Claudin1 (1:1500, AF0127, Affinity). The following secondary antibodies were used: goat anti-mouse (1:1000, SA00001-1, Proteintech), goat anti-rat (1:2000, A0192, Bioss), and goat anti-rabbit (1:1000, A0208, Bioss).

## qPCR

Total RNA was isolated from the PFC using the QIAzol Lysis Reagent (Qiagen) and cDNA was transcribed using the HiScript® II Q Select RT SuperMix for qPCR (VAZYME, Nanjing, CN). qPCR was performed with the AceQ Qpcr SYBR Green Master Mix (VAZYME) using the primers found in Supplementary Table 2.

## Transmission electron microscopy (TEM)

After euthanizing the mice, a 1 mm³ tissue block was rapidly dissected from the PFC and immediately fixed overnight at 4 °C in 2.5% glutaraldehyde. The tissue block was then washed three times with 0.1 M PBS, each wash lasting 10 min. Post-fixation was carried out in 1% osmium tetroxide solution for 1 and 2 h. Following post-fixation, the tissue was washed three times with 0.1 M PBS. The tissue block was dehydrated through a graded ethanol series (30%, 50%, 70%, 90%, and 100%) and acetone. After dehydration, the tissue was infiltrated with epoxy resin and baked at 60 °C for 48 h to cure the resin. The embedded tissue was sectioned into ultrathin slices approximately 70 nm thick using a Leica ultra-microtome. The sections were then mounted on copper grids and sequentially stained with 2% uranyl acetate and 0.4% lead citrate. The stained sections were observed under a Hitachi HT7800 transmission electron microscope at 80 kV to assess neuronal synaptic terminal loss.

## Cytokine analysis

Cytokine levels were measured in both tissue samples (skin, PFC) and serum. After weighing and finely mincing the tissue, it was homogenized in an equal volume of PBS using an ice-cold mortar. The homogenate was then centrifuged at 3000 rpm for 10 min, and the supernatant was collected for analysis. Blood samples were obtained from the mice by enucleation, left to stand for 2 h, and then centrifuged at 1000 rpm for 15 min to separate the serum. Immunoglobulins (IgE, IgG), inflammatory cytokines (TNFα, IL1β, IL6, IL10), oxidative stress markers (MDA, SOD, GP × 4), and serum corticosterone (Cort) were measured using ELISA kits from FineTest. Group comparisons were performed using a $t$-test, with significant differences indicated by an asterisk (* < 0.05).

## Hematoxylin and eosin (H&E) staining

After the mice were sacrificed, skin and brain tissues were immediately dissected and fixed in 4% paraformaldehyde. The tissues were then embedded in paraffin, sectioned, and stained with hematoxylin and eosin (H&E). Images were captured using a Leica FLEXACAM C1.

## Striped hamster occurrence data

In this study, data on striped hamsters were sourced from three areas: our team's field sampling records, literature reports, and the Global

Biodiversity Information Facility (GBIF, https://www.gbif.org). Field sampling records were collected using handheld GPS devices to obtain precise latitude and longitude coordinates. For distribution points reported in the literature, we recorded the coordinates when available; if exact coordinates were not provided, we retained the distribution data at the village or town level and randomly generated the coordinates using Google Maps (https://www.google.com.hk/maps/). We also used the rgbif package (v 3.8.1, https://github.com/ropensci/rgbif) to download distribution data with field coordinates from GBIF. After compiling data from these three sources, we cleaned the dataset by removing duplicates, points with zero coordinates, points located at sea, capital cities, and institutional coordinates. We retained only unique distribution points within a 5 km radius (Supplementary Table 1).

## Environmental variables

Species survival and geographic distribution are closely linked to climatic and environmental factors, particularly temperature and precipitation, which can either facilitate or hinder a species' life cycle. To evaluate the impact of these climatic factors on the distribution of striped hamster, we sourced climate data from the WorldClim database (http://worldclim.org/). WorldClim version 2.32 provides average monthly climate data, including minimum, mean, and maximum temperature, as well as precipitation for the period 1970–2000, with a spatial resolution of 5 arcminutes (-5 km²). We extracted environmental variables for each distribution point and conducted Pearson correlation analysis, retaining only one variable when the correlation coefficient exceeded 0.8. Additionally, we downloaded climate variables under four Shared Socioeconomic Pathways (SSP126, SSP245, SSP370, SSP585) for three future time periods (2041–2060, 2061–2080, 2081–2100) to predict the future habitat suitability of striped hamster.

## SDM model establishment

We used the Biomod2 (v4.3.1, https://github.com/biomodhub/biomod2) R package to construct species distribution models, leveraging its multiple algorithms and ability to create ensemble models. The ensemble model incorporated 11 algorithms: Artificial Neural Networks (ANN), Surface Range Envelope (SRE), Flexible Discriminant Analysis (FDA), General Linear Models (GLM), General Additive Models (GAM), General Boosted Models (GBM), Classification Tree Analysis (CTA), Multiple Adaptive Regression Splines (MARS), Random Forests (RF), and the Maximum Entropy model (MaxEnt). To enhance the sensitivity of model predictions, we used the SRE model to generate 500 pseudo-absence points. The dataset was split into two parts: 70% for model training and 30% for evaluation, with each algorithm running twice. Model performance was assessed using the Area Under the Curve (AUC) of the receiver operating characteristics (ROC) and True Skill Statistic (TSS). Algorithms with TSS < 0.75 and AUC < 0.9 were excluded. The ensemble model was constructed using the EMca method, selected for its highest accuracy. Using current environmental variables, we modeled the suitable habitat for striped hamster and compared the potential changes under future climate conditions. We used mean interpolation to calculate the habitat suitability index (HSI) for striped hamsters at 10-year intervals. Based on the current HSI and projected changes in future HSI, we selected three sites to serve as landscapes for model simulations. For each of these three sites, we conducted simulations under both static and dynamic climate conditions. A cell-based model was employed for these simulations, with the maximum field-sampled density of striped hamster translated into a carrying capacity of 8 individuals per cell.

## Demographic survey of striped hamsters

To gather demographic data on striped hamsters in the field, we conducted a two-year sampling study in the grasslands of Inner Mongolia. The detailed sampling methodology has been previously reported in our publications. Briefly, we selected nine sampling sites based on temperature and precipitation gradients. Sampling was performed three times each year during the striped hamster breeding season, specifically in May, July, and September. For each sampling session, we established three trap lines at each site, with each line spaced over 50 meters apart. Each trap line contained 100 traps, placed 5 meters apart, covering approximately 0.5 hectares per site. Traps were set overnight, and we checked them early the next morning to record the number of striped hamsters captured and to calculate the population density at each site. Captured hamsters were then brought to a field laboratory for weighing and dissection. For female hamsters, we counted the number of embryos or uterine scars to determine litter size.

## Demography and dispersal parameters

The striped hamster is a solitary species that disperses to new locations upon reaching adulthood. Reproduction in this species occurs annually, and their typical lifespan in the wild is about one year. We chose to use an only-female model because previous field studies revealed that nearly all adult females (>90%) were observed to be pregnant. This high pregnancy rate makes the female model particularly relevant for our research. Striped hamsters go through two main life stages: subadult and adult. Sexual maturity can be reached as early as 1.5 months. To avoid the common mistake in transition matrix modeling, where offspring born in year $t$ mistakenly advance to the next stage within the same year, we added an additional juvenile stage (stage 0). This ensures that juveniles develop to stage 1 in the year following their birth, allowing us to simulate post-natal dispersal accurately.

Previous research by Zhang et al. in Beijing found an average litter size of 6.1[59], consistent with our findings in Inner Mongolia (6.48 ± 1.97). We lacked specific data on the survival rate from stage 0 to stage 1, so we used the 68% (the female ratio) as survival rate reported in the literature. The adult mortality rate, not detailed in the literature, was conservatively set at 0.9. Given that striped hamsters primarily die during winter, we chose survival option 2 in our model. Although different site densities did not show significant reproductive differences ($P = 0.527$), we adjusted the model to account for increased mortality with rising density, reflecting resource competition pressures.

Dispersal in striped hamsters exhibits a stage-dependent pattern, predominantly occurring after they attain adulthood (beyond 2 months of age), during which they leave their natal areas to establish themselves in new habitats. We focused on the likelihood of leaving the natal area (departure), converting data from our enclosure experiments into departure rates for Flea– and Flea+ conditions. We assumed equal departure rates for both sexes, as our enclosure experiments showed no significant differences (Fig. 6j). The maximum transfer distance was set at 500 meters, shorter than previous research on small mammals[60], guaranteeing that striped hamster has the ability to dispersal to adjacent grids. In our model, successful establishment of female striped hamsters in a new habitat is contingent upon finding a male partner.

## RangeshiftR simulation

In this study, we initiated our simulation by introducing only adult individuals of striped hamsters into the selected sites. At the outset, 100 cells were chosen in Site 1 (S1), with 2 individuals placed in each cell. For Site 2 (S2), we selected 3000 cells, each receiving 8 individuals. Similarly, Site 3 (S3) included 500 cells, also with 8 individuals per cell. These specific parameters were chosen to ensure the visualization of the simulation results. We then ran each model 100 times, simulating a 100-year period with. Throughout this period, we monitored the density and distribution area of striped hamster within each

cell, aiming to capture the dynamics of population spread and habitat occupation over time.

### Statistical methods

Behavioral test results of mice were analyzed using a one-tailed *t*-test. A two-tailed *t*-test was applied to compare data between Flea− and Flea+ groups. The behavioral test results of striped hamsters were analyzed using a Generalized Linear Model (GLM), controlling for individual body weight. Additionally, appropriate distributions were chosen for each indicator. We employed a GLM with a Poisson distribution to analyze the effects of various ecological factors on flea burden. Additionally, to examine the factors influencing the escape of black-striped hamsters from the enclosures, we used a generalized linear mixed model with a Poisson distribution, incorporating the enclosures as a random variable.

### Reporting summary

Further information on research design is available in the Nature Portfolio Reporting Summary linked to this article.

## Data availability

The raw RNA-seq data generated in this study have been deposited in Dryad under accession code [https://doi.org/10.5061/dryad.x3ffbg7vh]. The raw neurotransmitter data are available at Dryad under accession code [https://doi.org/10.5061/dryad.7d7wm384n]. Source data are provided with this paper.

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

## Acknowledgements

This research was funded by the Major Program of National Natural Science Foundation of China, grant number 32090023 (Q.L.).

## Author contributions

L.L., Q.L., and P.L. designed the study; P.L., X.X., and D.R. worked on the methodology; P.L., G.L., W.L., N.Z., D.L., M.C., J.W., X.L., C.Z., Q.L., and L.L. performed investigation; P.L. and L.P. performed worked on data analysis and visualization; Q.L. and L.L. supervised this work; P.L. wrote the original manuscript; P.L., Q.L., and L.L. reviewed and edited the writing.

## Competing interests

The authors declare no competing interests.
