## [Transparent Peer Review file · Nature Communications]

Ectoparasites Enhance Survival by Suppressing Host Exploration and Limiting Dispersal

Corresponding Author: Professor Liang Lu

Version 0:

Reviewer comments:

Reviewer #1

(Remarks to the Author)

The authors describe an interesting article in which behavior caused by an ectoparasitic infection is correlated to biochemical and consequently behavioral changes in the host animal. The authors apply behavioral tests, metabolomics workflows and transcriptomics work in their investigations. I wonder that the authors did not endeavor to perform a discovery proteomics analysis of the test animals to determine ectoparasitic impacts at the proteomic level. It would have been a nice addition to their work, making it more of a 'systems biology' investigation. Indeed, what is observed at the transcriptomic level is not always well-correlated with observations at the proteomic level. The manuscript is experiment-diverse and data-rich. The authors need to bring it all together to support their conclusions. The paper presently reads like a laundry-list of different experiments. I encourage the authors to draw connections and connections throughout the work they have presented.

(01) The authors perform experiments with ectoparasitic fleas, hamsters and mice, though their work reads like they are drawing conclusions on ectoparasites in general. Why do they believe such broad conclusions can be drawn? The term 'ectoparasite' should be defined.

(02) It was not immediately clear in the introduction if the authors based their experiments on mice or hamsters or both. Further reading revealed that it was both but this needs to be made clear early on in the manuscript.

(03) The authors mention the "One Health" framework a couple of times. If this is important to their work - and I gather that it is - they should define it and make frequent connections to it. I feel that this is missing.

(04) Line 36: Replace "Evidences" with "Evidence".

(05) Line 40/41: "For ectoparasites, host dispersal can introduce greater stress, as they are more vulnerable to environmental pressure". Who is "they"? The ectoparasite or the host or both?

(06) Line 48: The authors describe using the flea-mouse system in their work. They also used the hamster model. The confusion should be clarified.

(07) Line 56: The authors state that "After one week, all *C. felis* had died, while approximately 50% of the *X. cheopis* fleas survived..." Can the authors elaborate on why?

(08) Line 60: The authors state "After infecting for four weeks, ...". It is assumed that the mice were flea-free at the beginning of the experiment. What was the survival rate in 4 weeks? Was behavior recorded in the Flea+ group before infection? If so, how did it compare with the behavior from the Flea- group?

(09) The authors use phrases like 'significant increase/decrease' throughout the manuscript. These are vague statements. What does 'significant' in terms of the data they describe? One example is on Lines 72-73. Another example is on Line 160. There are many other instances.

(10) In line 78, the authors state that brain regions undergoing changes in glucose uptake (in a flea infestation of mice), are associated with anxiety. A given brain region is responsible for many functions and traits. Why did the authors draw this

conclusion? If the authors are describing conclusions based on their data (or literature data), this ought to be elaborated.

(11) Lines 79-80: "We concluded that flea bites lead to changes in multiple brain regions, including those involved in emotion regulation." This is a very broad conclusion. The authors should elaborate exactly how, early in their work, they were able to conclude this as all their subsequent work is tied to this conclusion.

(12) The authors describe some great information they could obtain from DEGs in transcriptomics analysis. They also describe some important neurotransmitter metabolomics work. What is missing is some connecting proteomics experiments to really make this a true systems biology investigation. The authors use their data to draw sweeping conclusions on migration, behavior, survival and effects from climate change. The proteomics data is a noticeable missing piece.

(13) Line 112: The authors describe "neurotransmitter metabolomics analysis". What extraction methods did they use to ensure maximum coverage of neurotransmitters? What neurotransmitters did they miss? Neurotransmitter concentrations in the mouse brain vary by region and can be difficult to measure accurately, due in no small part to their very low abundance and the amount of sample that can be used is small.

(14) Line 128: The authors state that "We infer that flea bites primarily impact the PFC, leading to reduced exploratory behavior in mice." This should be stated towards the end of the "Microglia Activation" section, after all the data evidence is presented.

(15) Line 193: "...noticeable damage..." needs elaboration.

(16) Line 196: "WB analysis showed a decrease in Claudin1 protein..." How are the authors quantifying this decrease and by how much is the protein concentration decreased?

(17) Section: Reduced exploratory behavior in Flea+ striped hamster. The authors describe field-capturing striped hamsters. So are these wild hamsters who already are flea-infested? The authors then proceed to discuss their assessments "after 4 weeks of infection." The hamsters were already infected at capture, though. Some clarification is needed here.

(18) Lines 214-216. The authors state "Additionally, heightened immune responses, increased inflammation, and oxidative stress were detected in both the skin and serum of the infected hamsters." This is based on transcriptomic data. The proteomic data was not explored and there likely is some non-correlation between the two.

(19) Line 219. The authors describe a 'field closure experiment.' So were flea+ hamsters put in one enclosure and flea-hamsters in another? This is confusing because the authors mention that these are wild hamsters that already had a flea burden. Clarification is needed.

(20) Line 232. Can the authors postulate on why flea bites reduce host exploratory behavior? If this is explained in the "Discussion" (and I think it is), this conclusion should not be drawn before the presentation and discussion of the evidence.

(21) Line 235: "Reduced Occupancy in Flea+ host." This section needs to be clarified. The link between the experiments described and climactic changes is not clear.

(22) Lines 279-281: "Present study demonstrates that reduced exploratory behavior is linked to microglial activation and abnormal differentiation of GABAergic neurons in the PFC." This conclusion seems a bit stretched.

(23) Lines 2296-297: "...interconnectedness of ecological and public health systems under the One Health framework." This concept has not been adequately elaborated in this work.

(24) Line 328. Were 150 new fleas added weekly?

(25) Line 340. So was it assumed that at the end of the 4-week period, all wild-caught fleas were dead?

(26) Line 344. Is the "study area" referring to Xilinhot, Inner Mongolia, China?

(27) Including common names of the organisms would be helpful.

(28) Line 499. Change "detection" to "analysis".

(29) Line 499. UHPLC-QTRAP - Specific name of instrument and vendor name required.

(30) The MS method needs to be described in the "Neurotransmitter Metabolomics Analysis" section (line 492) of the Methods.

(31) Line 506: More details on developing the quantitative method are required.

(32) Line 516: The authors describe using a threshold of 1.5. What was the rationale behind this?

(33) Line 524: PBS acronym needs to be spelled out at first mention and the acronym placed in parenthesis. Same for TBST

(line 532), DAPI (line 536), FBS (line 553), EDTA (line 554), RMPI (line 556), FVS700 (line 562), DMSO (line 563), RIPA (line 579), PMSF (line 580), PVDF (line 583), PAGE (line 583), GAPDH (line 584), cDNA (line 596) and others.

(34) Line 578: "...approximately the size of a mung bean..." It is best to give approximate size measurements in mathematical units.

(35) The figures within figures are too small. Each figure composite is too densely-packed. The authors should consider sending some of the 'figures within figures' to Supplementary Information.

(XX) Line 614: Cytokine Analysis. Was this analysis performed on both tissue and blood? Please clarify.

(34)

Reviewer #2

(Remarks to the Author)

In the manuscript entitled "Ectoparasites Enhance Survival by Limiting Host Dispersal through Microglial Activation," the authors investigate the impact of flea bites on mice and striped hamsters in a multiscale framework, from behavioral changes in terms of dispersion to gene expression in both skin and brain tissues. Their results suggest an impact of flea bites on the dispersal behavior of the host that is mediated by an alteration of their neurobiology, notably through microglial activation in the PFC. They further tested the impact of flea bites on current and predicted host distribution and demonstrated that flea bites tend to reduce the host's geographical range.

This study addresses a very interesting topic: host manipulation by ectoparasites, which is far less documented than manipulation by endoparasites. The data they produce to document such alterations of host behavior and neurobiology induced by flea bites is very impressive in quantity and diversity. The range of methods used is also astonishing, going from behavioral experiments in controlled setups to field experiments, transcriptomics, metabolomics, and species distribution models. Such a diversity and complexity of approaches is truly original and allows for a very in-depth investigation of the impact of an ectoparasite on its host. I would like to congratulate the authors on such work done.

Below are my major comments, followed by some line-by-line, more minor comments.

While the quantity and diversity of approaches add great value to the manuscript, it is also, to me, the main drawback of the study. It is often difficult to follow how the different experiments and datasets used are articulated with one another. To help with this, hypotheses could be stated at the end of the introductory paragraph and could be more clearly articulated between the different sections to provide a better sense of progression regarding the main questions throughout the manuscript. A schematic summarizing how the authors envision the way flea bites can alter host dispersal could also be added, which would show how each aspect is tested.

Still linked to the fact that the impact of flea bites on the host is investigated at different levels (from behavioral or transcriptomic responses in brain and skin tissue, for instance), I noticed that the authors often claim causal links between flea bites, brain changes, and dispersal behavior. While they provide strong evidence of associations between these, no causal link has been established. However, the authors tend to argue so. More nuance would be appreciated. For example, in line 128: There is evidence that flea bites are associated with decreased dispersal behavior and that flea bites are associated with changes in brain gene expression; however, no test was done to establish that these brain alterations cause changes in dispersal behavior. See line 165, for instance, too. Some nuance could be brought.

The authors focus on what they refer to as exploratory behavior as the main altered phenotype caused by flea bites. I am curious about how they distinguish between a reduction in exploratory behavior and reduced locomotion. In most of their setups, it seems that the two could overlap. I raise this question because previous studies have shown that parasites, by depleting host resources, can decrease their overall locomotion. Have you investigated the expression of molecular functions related to metabolism (for instance, in the skin transcriptomic data) that would be enriched, for example, in downregulated genes?

I have some difficulties with the enclosure experiment conducted on striped hamsters. First, I don't understand why the non-capture rate is a good proxy for the non-dispersal rate. If the hamster escapes the enclosure, then one could argue that it has dispersed. Experimental setups with enclosures linked by dispersal corridors seem more appropriate as they would allow a direct measure of dispersal. This issue also affects the future host range simulations, as the dispersal rate used was the one calculated from the enclosure experiment. Finally, in the Extended Data Fig. 8 enclosure experiment schematics, it seems that both the number of hosts and their sex ratio vary between rounds. However, rounds also vary regarding whether hamsters are infected or not. Considering the sample size suggested in the schematics, I doubt this would be enough to properly distinguish between sex, population size, and infection status effects on dispersal. Wouldn't it have been more straightforward to simply compare infected versus non-infected populations in terms of dispersion?

I had some issues regarding figure readability. The text is often too small to read, making it hard to confirm what is claimed. On this same note, I think more metrics could be provided, especially regarding the transcriptomics, such as count data, log fold changes, and corrected p-values. The same goes for the results of the enrichment tests. Plots are a good visual option, but more statistical metrics should be made available to ensure the results' trustworthiness.

Finally, the discussion is rather short and superficial, considering the numerous aspects of host alterations caused by flea bites. Again, I would like to see clear links made between the different dimensions of the paper. See my comment above about a general schematic.

Minor Comments:

L. 60:

Clarifying the method used to measure dispersal behavior would improve overall clarity (e.g., "We assess dispersal behavior by testing the time spent in an open field experimental setup and in an elevated maze...").

L. 89:

Please provide the number of DEGs. Also, log fold changes and corrected p-values should be provided in the supplementary material. The same applies to enrichment tests.

L. 68-125:

I agree with the authors that the mentioned brain gene expression differences could be linked to emotion regulation and, as such, to anxiety. However, with this analysis, no causal link can be evidenced. I suggest emphasizing more on altered brain expression (since this is the main result) rather than on the functional consequences that cannot be verified. More clarity on the hypotheses tested could help.

L. 104-106:

I don't fully understand how the GSEA suggests that the PFC is more impacted than the HC and TH. Typically, this analysis investigates impacted functions but does not indicate the extent of alteration. DEG analysis might provide more insight into this.

L. 142-143:

In which tissue was the expression of these neurons investigated? This is not very clear.

L. 156:

Please provide a reference. Also, does this imply that flea bites do not directly increase anxiety, as previously stated? A different mechanism seems to be proposed here.

L. 187-189:

Please elaborate on the rationale behind this statement.

L. 197-199:

Explain in more detail why you support this explanation, even though the PCR does not confirm it.

L. 275:

Should it be "exploratory behavior" instead of "explanation behavior"?

L. 211:

Please mention if you detected an effect of infection on the time spent in the central area of the open field.

L. 229:

If I understand correctly, this is true only for the second round. The way results are presented tends to blur the interpretation related to the stated hypothesis. The reader would want to know if there was a difference between infected and non-infected hamsters regarding the non-recapture rate. Also, results could be presented as a time series rather than as three separate outputs, with the overall dynamics over time tested.

L. 233:

The results indicate that the non-capture rate increases for infected hamsters (L. 229), which seems contradictory to the statement that "flea bites reduce host exploratory behavior, thereby decreasing the likelihood of the host leaving its natal habitat." If this were the case, I would expect the non-capture rate to increase for non-infected hamsters and remain the same for infected ones.

L. 241:

Please clarify that W4 is a site.

L. 279-281:

Again, this statement is somehow confusing, as this is not what is tested. The effect of flea bites is tested. Please revise the phrasing, as it is misleading.

L. 187:

Please elaborate on the rationale here.

L. 356:

Why infecting hamsters during the daytime if they are nocturnal?

L. 375-377:

This hypothesis is not very intuitive. Why not measuring the time spent near the edge directly, as escaping shows exploratory tendencies? Alternatively, use a different setup to measure dispersal directly.

L. 384:

Were non-infected hamsters also placed in a 5L glass beaker to control for potential behavior changes, particularly increased stress levels?

L. 417:

Please specify mice and hamsters.

L. 456:

Please provide access to the code for the custom package. Also, throughout the manuscript, provide software versions and references.

L. 186:

Please provide the reference and the update date of the database used.

L. 669:

Please provide the total number of sites used for the model.

L. 721-723:

You mentioned that striped hamsters disperse upon reaching maturity (L. 700). In your simulation, this specific dispersal event is considered. However, you deduced the dispersal rate from field experiments conducted on adult striped hamsters. How can you ensure that the dispersal observed in your field experiment is comparable to dispersal occurring from the juvenile to adult stage?

L. 749:

Six enclosures might be too few to estimate enclosure-associated variance as done in a GLMM.

L. 872:

Please clarify that the PCA is performed on gene expression data.

L. 874:

It should be "significantly differentially expressed genes" rather than "different genes."

Extended Data Fig. 6 and 8:

The figure is difficult to read. The panel letters are disproportionately large compared to the figure content.

Extended Data Fig. 8:

I don't fully understand the experimental setup from the schematic. Since the number of hamsters and the sex ratio vary between enclosures, how can you be sure that differences in dispersal are due to the treatment? Please clarify the hypothesis tested and how the setup addresses it.

Data Availability:

I noticed that the raw fastq reads from transcriptomic and neurotransmitter datasets are available. However, other data, such as occurrences and environmental data used for the SDM, should also be made available. Additionally, count data from the transcriptomic analysis, along with log fold changes and corrected p-values, should be provided.

Reviewer #3

(Remarks to the Author)

Overall comments:

In this manuscript, Liu, et al investigated the effect of ectoparasitic infection on host dispersal and the underlying machinery of immune interactions. They found ectoparasites can alter metabolic activity in specific brain regions of mice and decreased their exploratory behavior. In addition, systemic inflammation and increased microglia activation were observed in mice with ectoparasitic infection.

Their work is interesting and provides new insights into the mechanism of ectoparasites influencing host dispersal and the underlying immune interactions. However, some key experiments showed unsolid or controversial data, making their conclusion not convincing. In addition, quite a few small mistakes and mislabeling severely dampen the quality of this manuscript. I will discuss some of this below:

Major points:

1) Authors don't provide any PET-CT images in the main figures or extended figures. The quantification shown in Fig.1C-D is not convincing without any images. It's strongly recommended to add the image from the Flea- and Flea+ group with the labeling of brain regions (e.g. PMID: 36451231; 28654383). In addition, most of the brain region showed reduced 18F-FDG uptake in mice from Flea+ group (Fig. 1c), reflecting the Loss of neuronal activity. However, recent studies demonstrate

anxiety is associated with increased neuronal excitability, which is controversial with authors' data of 18F-FDG uptake.
2) Since authors found significant enrichment in Tyrosine and Tryptophan metabolism pathways in the PFC which are crucial for synthesizing dopamine, norepinephrine, and serotonin, it would strengthen the support to their conclusion if the concentration of these neurotransmitters can be added to main figures.

3) In Fig. 1g, it seems authors failed to characterize the microglia and infiltrating macrophages. The increased inflammation in brain may also cause the infiltration of macrophages (PMID: 29133437; 31325960). In addition, the huge increase in the transcriptional level of CD11b and CD68 from Flea+ group can't be reflected by the FACS data (Extended Fig. 6b). A better FACS analysis should be repeated in the revision.

4) The signal shown in the TUNEL staining of Extended Fig. 6g is so low and needs to be replaced with pictures with higher quality.

5) It has been reported that the activation of microglia resulted in increased GABAergic neurons. (PMID: 17459525; 37806513). Therefore, the reduced GABAergic neurons observed in Flea+ group may not be induced by increased inflammation and microglia activation.

6) To determine whether the flea bite-induced systemic inflammation affected exploratory behavior of mice through microglia activation, a depletion of microglia or treatments with the activation inhibitor need to be utilized in Flea+ group.

7) The study in the exploratory behavior in striped hamster has no supporting mechanistic data. Although authors did a huge amount of work in mouse model, whether the same mechanism can be shared in hamster is questionable. Authors should either use mouse model in natural settings, or provide evidence of the similarity of brain metabolism and inflammation between mice and hamsters.

Minor points:

1) The resolution of some Extended figures is so low to identify the labeling of graphs.

2) The abstract part should not include any references.

3) The word "survival" in the Title was misspelled.

4) The "Introduction" title is missing.

5) There are multiple mistakes in Figure numbers throughout the manuscript. For instance, on line 123, it says "In the HC, enrichment of the Histidine metabolism pathway (Extended Data Fig. 5j)". However, the related figure is Extended Data Fig. 5k. In the figure legend of extended Fig. 1e, it says "Spearman correlation of 18F-FDG uptake between different brain regions in the Flea- group (n=6)". However, the labeling in this figure is "Flea+".

Version 1:

Reviewer comments:

Reviewer #1

(Remarks to the Author)

I feel that the authors have adequately addressed my comments on the manuscript.

Reviewer #2

(Remarks to the Author)

Overall, the authors have carefully addressed the previous reviewer comments, and the manuscript is now much clearer and more consistent. I have only minor comments that could further improve clarity:

In the abstract, often you have space inserted before period, please remove them.

L. 67 Perhaps clarify by stating 'using two rodent models' so that it is clearer later.

L. 127-128 it is not the differentially expressed genes that are enriched in biological functions but some functions that are enriched with differentially expressed genes.

L. 259 To clarify, consider adding "On the contrary, a hamster that explores more will end up in the center of the enclosure and is more likely to be recaptured." This information could also be included in the Methods section
I appreciated the detailed explanation in the reviewer's response document, but for next readers I think adding that here will help to understand the rationale.

L. 261: The difference of treatment between each round needs to be explained before you start describing the results.

Discussion, I am not convinced at all by the additions made to the discussion about other host-parasite systems where the parasite manipulates its host. These examples do not significantly enhance the perceived novelty of the work. From literature there is abundant evidence of parasitic manipulation, which does not need to be enumerated here. What is more striking is that most examples involve endoparasites, which, in my view, should be more strongly emphasized. Revealing mechanistically how a parasite that is not inside the host can still directly alter its behavior (and not through resources reallocation) is quite new which is why to me this should be central here.

L. 426 Please include an explanation for infecting during the daytime (since in natura infection occurs when hamsters are resting) for the benefit of future readers.

In the reference list, double check that any species name is consistently italicized.

Fig. 2: Panels C and E are not readable.

Fig. 3: The axis labels are too small and should be enlarged for readability.

Fig. 7: Panels a and e are not readable.

Fig. 9: The overall figure is too small and difficult to read.

Reviewer #3

(Remarks to the Author)

Generally, Authors have done a good job for the revision. Most of questions have been answered properly. However, there are still a few problems to be addressed before the final acceptance.

1. The resolution of main figures has been improved in the revised version but needs further elevation. In contrast, the resolution of supplemental figures is good for publication.
2. The gating strategy in Supplementary Fig 14 and Fig. 3h may have some problems since there should be such huge amount of CD45+CD11b- cells (the population below the microglia cluster) in Flea- group (as shown in PMID: 29133437; 31325960). In addition, during the systemic inflammation, the amount of macrophages should be highly increased, which was not reflected by the ratio in Fig. 3h.
3. Authors explanation for my comment 5-7 ("This study is primarily exploratory, focusing on the behavioral strategy of "ectoparasite manipulation of host dispersal," while the molecular mechanisms serve as supporting evidence for our conclusions") is acceptable but also weakens the scientific significance of this study due to unclear mechanism. The abstract needs further editing because no solid evidence to support the conclusion that "ectoparasites can alter metabolic activity in specific brain regions of mice, particularly by excessively activating microglia in the prefrontal cortex. This activation damages the synaptic structures of neurons, disrupting the normal differentiation of GABAergic neurons" without an experiment deleting or inhibiting microglia.

Version 2:

Reviewer comments:

Reviewer #3

(Remarks to the Author)

Authors addressed all the issues mentioned in the previous review and the manuscript can be accepted in its current version.

Submission ID: **NCOMMS-24-65632A**
point-by-point response to reviewers

REVIEWER COMMENTS

Reviewer #1 (Remarks to the Author):

The authors describe an interesting article in which behavior caused by an ectoparasitic infection is correlated to biochemical and consequently behavioral changes in the host animal. The authors apply behavioral tests, metabolomics workflows and transcriptomics work in their investigations. I wonder that the authors did not endeavor to perform a discovery proteomics analysis of the test animals to determine ectoparasitic impacts at the proteomic level. It would have been a nice addition to their work, making it more of a 'systems biology' investigation. Indeed, what is observed at the transcriptomic level is not always well-correlated with observations at the proteomic level. The manuscript is experiment-diverse and data-rich. The authors need to bring it all together to support their conclusions. The paper presently reads like a laundry-list of different experiments. I encourage the authors to draw connections and connections throughout the work they have presented.

Comment 1

The authors perform experiments with ectoparasitic fleas, hamsters and mice, though their work reads like they are drawing conclusions on ectoparasites in general. Why do they believe such broad conclusions can be drawn? The term 'ectoparasite' should be defined.

Response:

We sincerely appreciate the reviewers' thorough evaluation of our study and their valuable suggestions. Our study aims to address a broader question regarding ectoparasites through experiments involving fleas, laboratory mice, and wild striped hamsters. This approach is somewhat analogous to using animal models to investigate disease pathogenesis. Ectoparasites are organisms that live on the surface or skin of a host, primarily including blood-feeding arthropods such as fleas, mites, and ticks. They depend on their hosts for survival, feeding on blood or tissues, and consequently influencing host health and behavior.

First, ectoparasites mainly belong to the phylum Arthropoda, including insects (fleas) and arachnids (mites, ticks). They use specialized mouthparts, either piercing-sucking or cutting, to penetrate the host's skin and extract nutrients from blood or tissue fluids. Over prolonged infestations, hosts develop immune responses such as

inflammation, allergic reactions, and cellular immune activation. Furthermore, ectoparasite saliva contains various immunomodulatory molecules that suppress host immune responses, prolonging feeding duration. In our previous research [PMID: 38084399] on the mechanisms shaping MHC gene diversity in striped hamsters across 10 sites in Inner Mongolia, we found that two key ectoparasites—fleas and gamasid mites—contribute to MHC allele diversity through heterozygote advantage. Additionally, we observed a significant negative correlation between MHC allele diversity and parasite richness (Figure 1, Table 1). These findings suggest that different ectoparasites may exert similar selective pressures on their hosts.

Figure 1. Testing predictions of heterozygosity advantage for the four parasite indices. (A) Flea load; (B) Gamasid mite; (C) Bartonella infection; (D) Parasite richness. For the purpose of plotting, the residual parasite indices for each hamster was calculated by using a mixed-effect generalized linear model. Each dot represents a fish. The data supported heterozygosity advantage in Flea load, gamasid mite load, and parasite richness.

Table 1. Coefficient estimates and significance of parameters in best-fit models predicting parasitism in striped hamster.

Response	coeffocoents	estimate	s.e.	z-value	Pr(> z)
Flea	(intercept)	2.509	0.779	3.223	0.001**
	season(Spring)	0.413	0.291	1.417	0.156
	season(Summer)	-1.009	0.268	-3.766	<0.001***
	sex(Female)	-0.595	0.203	-2.937	0.003**
	Carcass Weight	0.102	0.022	4.551	<0.001***
	Hs_exp	-0.698	0.360	-1.936	0.053
	st_num	-1.379	0.230	-5.985	<0.001***
Gamasid Mite ^a	(intercept)	0.709	0.403	1.760	0.078
	season(Spring)	-0.184	0.318	-0.577	0.564
	season(Summer)	-0.245	0.212	-1.157	0.247
	sex(Female)	-0.561	0.184	-3.051	0.002**
	Hs_exp	-0.345	0.338	-1.020	0.308
	MHC	-0.040	0.019	-2.052	0.040*
Parasite Richness	(intercept)	1.162	0.290	4.010	<0.001***
	season(Spring)	-0.038	0.207	-0.183	0.855
	season(Summer)	-0.193	0.140	-1.378	0.168
	sex(Female)	-0.545	0.122	-4.458	<0.001***
	Hs_exp	-0.211	0.226	-0.933	0.351
	MHC	-0.028	0.013	-2.218	0.027*

^aResponse variable for Gamasid Mite Load was square-root transformed

* $p < 0.05$, ** $p < 0.01$, * $p < 0.001$.

Note: season(Spring) means the difference between spring and autumn; season(Summer) means the difference between summer and autumn; sex(Female) means the difference between female and male.

Second, ectoparasites typically undergo multiple developmental stages, including egg, larva, pupa, and adult. Many ectoparasite larvae or eggs must develop in the host's nest, burrow, or a stable environment. For example, flea larvae primarily feed on organic debris and feces in the host's nest, while ticks often experience prolonged molting and developmental stages in the external environment. As a result, host habitats (e.g., the temperature and humidity of nests and burrows) play a crucial role in maintaining ectoparasite populations.

Lastly, compared to endoparasites that reside within hosts, ectoparasites are directly exposed to the external environment, making them more susceptible to temperature, humidity, and climate fluctuations. For instance, changes in ambient temperature can influence flea and tick survival and reproduction rates. Additionally, because ectoparasites move between the host and its environment, their populations are

constrained by the host's range of activity. Host migration, nesting behaviors, or habitat disturbances may significantly impact ectoparasite survival and dispersal. Consequently, ectoparasites tend to be more sensitive to environmental changes than endoparasites and may exhibit pronounced population fluctuations under global climate change or ecological shifts.

Although our experimental subjects are fleas and mice/hamsters, our study is designed based on the general biological characteristics of ectoparasites. Our findings are grounded in fundamental host-parasite interactions, such as parasite manipulation of host behavior, host immune regulation, and the relationship between parasite diversity and host adaptation. These mechanisms are present across many ectoparasite-host systems, suggesting that our findings have a degree of generalizability. However, we acknowledge that specific ectoparasite systems may exhibit unique traits, and future studies should explore the applicability of our conclusions to other parasite-host systems.

In response to the reviewers' suggestions, we will refine the definition of "ectoparasites" in our manuscript [L47]. Additionally, we will enhance our discussion of parasite manipulation strategies to emphasize the broader relevance of our findings.

Comment 2

It was not immediately clear in the introduction if the authors based their experiments on mice or hamsters or both. Further reading revealed that it was both but this needs to be made clear early on in the manuscript.

Response:

We appreciate the reviewers' careful evaluation of the manuscript's structure and agree that the choice of experimental animals should be more clearly explained in the introduction. To address this, we will explicitly state at the final paragraph of the introduction that our study involves both mice and hamsters [L66]. We will also outline that mice are used to investigate mechanistic aspects in a controlled laboratory environment, while hamsters are used in field experiments.

Comment 3

The authors mention the "One Health" framework a couple of times. If this is important to their work - and I gather that it is - they should define it and make frequent connections to it. I feel that this is missing.

Response:

Thank you for your suggestion. In the revised manuscript, we provided a brief definition of "One Health" in the introduction and further emphasize its significance in our study [L61].

Comment 4

Line 36: Replace "Evidences" with "Evidence".

Response:

Thank you very much for the reminder. We have made revisions accordingly [L41].

Comment 5

Line 40/41: "For ectoparasites, host dispersal can introduce greater stress, as they are more vulnerable to environmental pressure". Who is "they"? The ectoparasite or the host or both?

Response:

Thank you for pointing this out. To clarify, the term "they" refers to the ectoparasites, as they are more directly exposed to environmental pressures, as we noted in our response to Comment 1. The updated sentence will read:

"For ectoparasites, host dispersal can introduce greater stress, as ectoparasites are more vulnerable to environmental pressures."[L49]

Comment 6

Line 48: The authors describe using the flea-mouse system in their work. They also used the hamster model. The confusion should be clarified.

Response:

Thank you for your suggestion. We agree that the term "flea-mouse system" may cause confusion, and a more accurate description should be "**flea-rodent parasitic system**". In the laboratory, we use mice to test hypotheses and verify related mechanisms, providing mechanistic evidence. In the field, we use the flea-striped hamster system, aiming to expand the study to a broader ecological context and make the conclusions more universally applicable. We revised the description as above [L54].

Comment 7

Line 56: The authors state that "After one week, all *C. felis* had died, while approximately 50% of the *X. cheopis* fleas survived..." Can the authors elaborate on why?

Response:

Thank you for your question. The differences in mortality rates are related to the composition of the flea-host system. *Boris's* research suggests that the greater the phylogenetic distance between host and parasite, the higher the stress imposed on the host [PMID: 24661039]. This, in turn, may trigger a stronger defensive response, leading to increased parasite mortality. The high mortality rate of *C. felis* may be due to its strong host specificity, as it primarily parasitizes specific hosts like cats. In contrast, *X. cheopis* has lower host specificity, allowing it to survive more easily across different hosts, resulting in a higher survival rate.

Comment 8

Line 60: The authors state "After infecting for four weeks, ...". It is assumed that the mice were flea-free at the beginning of the experiment. What was the survival rate in 4 weeks? Was behavior recorded in the Flea+ group before infection? If so, how did it compare with the behavior from the Flea- group?

Response:

Thank you for your question. At the beginning of the experiment, all mice were flea-free, and we used 3-week-old SPF mice purchased directly from the supplier. Due to the continuous, chronic stress induced by flea bites, mice did not show any noticeable signs of mortality.

We did not conduct behavioral tests prior to infection because mouse behavior can vary with age during their growth. Therefore, pre-infection behavioral tests would not provide meaningful data for this experiment. However, both the Flea- and Flea+ groups were in identical condition before infection, ensuring that any behavioral changes observed were comparable between the Flea+ and Flea- groups. Our primary focus was on comparing the behavioral changes of age-matched mice under infected and uninfected conditions. In an unpublished study, we conducted an eight-week experiment with behavioral tests in the open field every two weeks. Our findings showed that in the second week post-infection, there was no significant difference between the Flea+ and Flea- groups in terms of time spent in the central area or the number of entries into it (Figure 2). We firmly believe that, following random assignment before the start of the infection experiment, there were no behavioral differences between the mice.

REDACTION

Comment 9

The authors use phrases like 'significant increase/decrease' throughout the manuscript. These are vague statements. What does 'significant' in terms of the data they describe? One example is on Lines 72-73. Another example is on Line 160. There

are many other instances.

Response:

Thank you for your suggestion. In our manuscript, we use terms like "significant increase/decrease" based on statistical tests, typically considering results significant when $P < 0.05$. In fact, this type of expression is commonly found in research articles in this field [PMID: 32703941, 29343269]. To ensure greater clarity and precision, we will refer to the article (PMID: 35165440), similar to this manuscript, to guide our revisions, and appropriately reduce the use of such expressions in accordance with the reviewer's recommendation [such as L149, L168, 193, 196].

Comment 10

In line 78, the authors state that brain regions undergoing changes in glucose uptake (in a flea infestation of mice), are associated with anxiety. A given brain region is responsible for many functions and traits. Why did the authors draw this conclusion? If the authors are describing conclusions based on their data (or literature data), this ought to be elaborated.

Response:

Thank you for your question. First, we would like to clarify that this manuscript was initially submitted to *Nature*, but the editor recommended transferring it to *Nature Communications*. Therefore, many aspects of the manuscript—such as structure, word count, number of references, and figures—were initially aligned with *Nature's* requirements. Since there are space limitations, we need to integrate the discussion within the results section, which is also a common practice in *Nature Communications* articles.

In line 78, we mentioned that certain brain regions in flea-infected mice exhibit changes in glucose uptake and that these regions are associated with anxiety-related behaviors. This conclusion is primarily based on existing literature on the functions of these brain regions, which we have cited accordingly. Specific areas, such as the prefrontal cortex, hippocampus, hypothalamus, thalamus, and basal ganglia, have been extensively studied and linked to anxiety behaviors. Our data indicate a correlation between glucose uptake changes in these regions and the anxiety-like behaviors observed in mice.

We acknowledge that our original description may have been unclear to readers. To improve clarity, we will revise the manuscript to explicitly state that this conclusion is primarily drawn from literature data. The revised sentence will read as follows:

“Notably, many of these regions have also been linked to anxiety in previous studies.” [L109]

Comment 11

Lines 79-80: "We concluded that flea bites lead to changes in multiple brain regions, including those involved in emotion regulation." This is a very broad conclusion. The authors should elaborate exactly how, early in their work, they were able to conclude this as all their subsequent work is tied to this conclusion.

Response:

Thank you for your valuable feedback. In lines 79–80, we stated: *"We concluded that flea bites lead to changes in multiple brain regions, including those involved in emotion regulation"*[L110]. This conclusion was drawn with reference to a *Nature* article investigating the effects of a gut microbiota-derived metabolite, 4EPS, on the mouse brain. In that study, researchers assessed glucose uptake in the mouse brain using 2DG and found that 4EPS was associated with increased glucose uptake in specific subregions of the hypothalamus (anterior area, lateral, and paraventricular nucleus), amygdala (anterior, basolateral, central, and cortical), and the bed nucleus of the stria terminalis (BNST), as well as the paraventricular nucleus of the thalamus (PVT). These regions are known to mediate responses to innate and learned fear stimuli and anxiety-related behaviors. The authors ultimately concluded: *"We conclude that gut exposure to 4EP in mice results in altered functional connectivity and activity in multiple brain regions, including several associated with the limbic system."*

We acknowledge that our conclusion may be broad and requires additional supporting details. Our hypothesis, based on a literature review and the characteristics of ectoparasites, suggests that ectoparasites enhance their survival fitness by reducing host dispersal from their birthplace. This effect may manifest through decreased exploratory behavior and increased anxiety in the host. To test this hypothesis, we conducted behavioral experiments (OFT and EPM). Since changes in mammalian behavior are primarily driven by neurophysiological mechanisms, our next step was to explore the neural basis of flea bite-induced anxiety.

Currently, no studies have examined ectoparasite-mediated changes in the mammalian brain. Given the complexity of mammalian neural circuits, which involve numerous brain regions and diverse functions, we first sought to identify the specific brain regions affected by flea bites. Using PET-CT, we evaluated metabolic activity changes across different brain regions in mice. By comparing flea-infected (Flea+) and

uninfected (Flea-) groups, we identified brain regions exhibiting significant alterations in glucose uptake. While these changes do not necessarily account for all anxiety-related behaviors, literature comparisons suggest that many of the identified regions are linked to emotional regulation, including anxiety. This led us to conclude: *"We concluded that flea bites lead to changes in multiple brain regions, including those involved in emotion regulation."* Furthermore, to further elucidate the molecular mechanisms underlying these neurophysiological changes, we specifically selected three brain regions that are more directly associated with emotion regulation for further investigation.

Nevertheless, to improve the scientific rigor and clarity of this conclusion, we have cited additional references to enhance the credibility of our conclusions in the revised manuscript [110].

Comment 12

The authors describe some great information they could obtain from DEGs in transcriptomics analysis. They also describe some important neurotransmitter metabolomics work. What is missing is some connecting proteomics experiments to really make this a true systems biology investigation. The authors use their data to draw sweeping conclusions on migration, behavior, survival and effects from climate change. The proteomics data is a noticeable missing piece.

Response:

We sincerely appreciate your valuable feedback. We recognize the importance of proteomics data in providing a more comprehensive understanding of the underlying mechanisms and agree that it would further enhance the systems biology framework of this study. However, due to experimental design constraints and sample limitations, proteomic analysis was not included in this study.

Nevertheless, our transcriptomic and neurotransmitter metabolomic data provide strong evidence supporting our conclusions. Transcriptomic analysis revealed key DEGs associated with neuroinflammation, oxidative stress, and neuronal differentiation, while metabolomic analysis further confirmed alterations in neurotransmitter metabolic pathways, highlighting the potential physiological impact of these molecular changes. Although the absence of proteomics data prevents this study from being a true systems biology investigation, we believe that transcriptomics and neurotransmitter metabolomics have provided crucial insights into the molecular mechanisms underlying our findings. To validate key pathways at the protein level, we conducted

additional analyses, including western blotting and immunofluorescence, which demonstrated microglial activation (IBA1), reduced mature neurons (NeuN+), increased neuronal apoptosis (TUNEL), synaptic damage (PSD95), and, notably, abnormal differentiation of GABAergic neurons (GAD65/67). Furthermore, these findings were cross-validated using flow cytometry and transmission electron microscopy. After reviewing two relevant Nature articles [PMID: 35165440, 27337340], we are confident that our current dataset adequately supports the conclusions of this study.

Additionally, we fully acknowledge the critical role of proteomics in bridging transcriptomic changes with functional protein expression. Therefore, in future studies, we plan to incorporate proteomic analysis to further refine the systems biology framework. We greatly appreciate the reviewer's constructive suggestion, which will help guide the next steps in our research.

Comment 13

Line 112: The authors describe "neurotransmitter metabolomics analysis". What extraction methods did they use to ensure maximum coverage of neurotransmitters? What neurotransmitters did they miss? Neurotransmitter concentrations in the mouse brain vary by region and can be difficult to measure accurately, due in no small part to their very low abundance and the amount of sample that can be used is small.

Response:

We thank the reviewer for their thoughtful comments and apologize for any confusion caused to the readers. We have revised the "neurotransmitter metabolomics analysis" section to provide additional details, including the extraction methods, quality control analysis, and data processing, which were of particular concern to the reviewer. For this work, we employed the methodology from Majorbio Bio-Pharm Technology Co. Ltd. (Shanghai, China), which analyzed 46 types of neurotransmitters. We have updated Source Data with a table displaying the neurotransmitter concentrations in various brain regions. As you correctly pointed out, some neurotransmitter concentrations were too low to be detected.

Comment 14

Line 128: The authors state that "We infer that flea bites primarily impact the PFC, leading to reduced exploratory behavior in mice." This should be stated towards the end of the "Microglia Activation" section, after all the data evidence is presented.

Response:

Thank you for your thoughtful comment. We respectfully disagree with the suggestion to move the statement "We infer that flea bites primarily impact the PFC, leading to reduced exploratory behavior in mice" to the end of the "Microglia Activation" section. Before making this inference, we conducted transcriptomic and neurotransmitter metabolomic analyses on the PFC, HC, TH. The transcriptomic data showed that the PFC exhibited the highest number of differentially expressed genes (DEGs) and was enriched in the most functional pathways. Compared to the HC and TH, the PFC upregulated genes were significantly associated with immune functions, while the downregulated genes were linked to neuronal differentiation.

In the neurotransmitter metabolomics analysis, we also detected the most differential neurotransmitter metabolites in the PFC—14 metabolites, significantly more than the 6 identified in the HC and 3 in the TH. Additionally, KEGG pathway enrichment analysis in the PFC highlighted pathways related to Tyrosine and Tryptophan metabolism, which are critical for the synthesis of dopamine, norepinephrine, and serotonin. These neurotransmitters are involved in regulating mood, cognition, and pain perception, and abnormalities in these pathways may affect neural excitability and signal transmission.

Based on these data, we believe there is strong evidence to infer that flea bites primarily impact the PFC. We also performed cell-type enrichment analysis on the transcriptomic data, which revealed that microglial cells and epithelial cells were the only cell types significantly upregulated in the PFC. We feel it is more logical to present this inference at the beginning of the "Microglia Activation" section, as this section focuses specifically on the PFC and its associated microglial activation at the cellular, protein, and mRNA levels. There is no direct comparison of the three brain regions in this section. Additionally, to avoid over-interpretation, we have made the following revisions: "*We infer that flea bites primarily impact the PFC, which may be linked to reduced exploratory behavior in mice.*"

We hope this explanation clarifies our reasoning for the placement of this inference. Thank you again for your helpful feedback.

Comment 15

(15) Line 193: "...noticeable damage..." needs elaboration.

Response:

Thank you for pointing this. We elaborated this description as follows [L227]:

“Further ELISA analysis of the PFC reveals elevated levels of inflammatory and oxidative markers (Error! Reference source not found.d). Consistently, HE staining shows noticeable cellular damage in the PFC of the Flea+ group, with a small number of infiltrating mononuclear cells at the injury site and extensive nuclear pyknosis and condensation leading to deep staining in the parenchyma. Additionally, the Flea+ group exhibits a larger area of damage (Error! Reference source not found.e).”

Comment 16

Line 196: "WB analysis showed a decrease in Claudin1 protein..." How are the authors quantifying this decrease and by how much is the protein concentration decreased?

Response:

Thank you for your question. In our Western Blot (WB) analysis, we quantified Claudin-1 protein expression changes by measuring the grayscale intensity of the protein bands. We used IPP6.0 software to perform grayscale analysis on each sample's Claudin-1 band and normalized the results to the internal control protein (GAPDH) to ensure data accuracy and comparability.

Regarding the specific reduction in protein levels, our results show that Claudin-1 expression in the Flea+ group decreased by approximately 50% compared to the Flea-group.

In response to the reviewer's comments, we have added this reduction percentage to the results section as follows:

"WB analysis showed a decrease (~50%) in Claudin-1 protein expression in the Flea+ group (Fig. 2f-g)."

Additionally, we have updated the "Western Blot" methods section to include a description of our use of IPP software for quantifying grayscale values from the blot images [L696].

Comment 17

Section: Reduced exploratory behavior in Flea+ striped hamster. The authors describe field-capturing striped hamsters. So are these wild hamsters who already are flea-infested? The authors then proceed to discuss their assessments "after 4 weeks of infection." The hamsters were already infected at capture, though. Some clarification is needed here.

Response:

Thank you for your valuable comments. We believe that conducting a control experiment using wild-caught striped hamsters under natural conditions to verify the impact of flea bites on host exploratory behavior is a key step in generalizing our hypothesis that "ectoparasites reduce host dispersal to increase survival." The striped hamsters we captured from the wild exhibit variable infection statuses, as mentioned in the text, which could be influenced by factors such as weight, sex, and MHC alleles (immune function). Therefore, we cannot guarantee that the striped hamsters used in the indoor infection experiment are consistent in these aspects.

To minimize the impact of confounding factors on the control experiment, we took the following measures: First, upon capturing the striped hamsters, we immediately removed any fleas they were carrying and recorded their flea burdens. Second, after flea removal, the hamsters were housed individually in the lab for at least 4 weeks to eliminate any residual effects of wild infections on the individuals. Finally, when conducting the indoor infection experiment, we re-grouped the hamsters based on their field infection status, weight, and other factors to ensure that the only difference between the Flea- and Flea+ was the presence or absence of flea infestation.

We believe that we have made the greatest effort to ensure the rigor of the wild-caught striped hamster control infection experiment. However, due to some issues in phrasing, it may have caused some confusion for readers. Therefore, in the revised version, we will make the following modifications to ensure clearer and more accurate expression. Specifically, we have moved and revised the sentence:

"The flea burden among the hamsters showed significant individual variation, influenced by factors such as capture method, body weight, and the number of MHC alleles (Extended Data Fig. 8b-c)."

from the "**Reduced Exploratory Behavior in Flea+ Striped Hamster**" section in the RESULTS to the "**Indoor Infection Experiment with Striped Hamsters**" section in the METHODS.

Comment 18

Lines 214-216. The authors state "Additionally, heightened immune responses, increased inflammation, and oxidative stress were detected in both the skin and serum of the infected hamsters." This is based on transcriptomic data. The proteomic data was not explored and there likely is some non-correlation between the two.

Response:

Thank you for your valuable comments. Regarding the increased immune response, inflammation, and oxidative stress that you mentioned, we indeed drew these conclusions based on protein-level data (from ELISA results), although they are not directly derived from proteomics data. Our experiments with mice showed that flea bites on the skin caused inflammation and oxidative damage (based on skin transcriptomics and ELISA data), which then led to systemic inflammation (evidenced by serum ELISA data), ultimately affecting neurophysiological health.

To further investigate, we used ectoparasites (from *Xenopsylla cheopis* to *Ctenocephalides felis*) and hosts (from mice to striped hamsters) established in the wild environment, and replicated the phenomena observed in the *Xenopsylla cheopis*-mice parasitic system. We confirmed these findings with additional ELISA results from the skin and serum of striped hamsters.

Of course, we recognize that there might be some discrepancies between transcriptomics and proteomics data. Therefore, we emphasize it as a potential direction for data integration and validation in future research. We also plan to incorporate proteomics data into our future studies to enhance our understanding and validation of these biological processes.

Comment 19

Line 219. The authors describe a 'field closure experiment.' So were flea+ hamsters put in one enclosure and flea- hamsters in another? This is confusing because the authors mention that these are wild hamsters that already had a flea burden. Clarification is needed.

Response:

Thank you for your valuable comments. As we mentioned in our response to Comment 17, when wild striped hamsters are captured, we immediately remove all fleas from each individual to eliminate potential effects caused by differences in flea burden. The hamsters are then housed in a flea-free laboratory environment for more than four weeks to further eliminate any residual influences. Therefore, by the time we conduct the enclosure experiment, we consider the hamsters to be in a standardized condition. However, to further minimize potential differences between groups, we still take body weight and field infection status into account when grouping. The term "Flea+ hamster" refers to those striped hamsters that were subjected to flea infection treatment during the enclosure experiment, meaning they were re-infected with fleas.

Comment 20

Line 232. Can the authors postulate on why flea bites reduce host exploratory behavior? If this is explained in the "Discussion" (and I think it is), this conclusion should not be drawn before the presentation and discussion of the evidence.

Response:

Thank you for the reviewer's suggestion. Based on the literature and the characteristics of ectoparasites, we proposed a hypothesis in the introduction, which we validated through behavioral experiments in the flea-mice laboratory infection system. We expanded this hypothesis using the wild flea-striped hamster parasitic system, where both indoor infection experiments and semi-natural enclosure experiments with striped hamsters replicated this phenomenon. We believe that this evidence is sufficient to support our conclusion. Furthermore, we discussed in the first paragraph of DISCUSSION section the different parasitic manipulation strategies, to strengthen our support for this conclusion.

Comment 21

Line 235: "Reduced Occupancy in Flea+ host." This section needs to be clarified. The link between the experiments described and climactic changes is not clear.

Response:

Thank you for your valuable comments. We believe the reviewer's concern is about whether climate change, such as temperature shifts, could influence the phenotype of "ectoparasites reducing exploratory behavior" and how this phenotype is linked to climate change without compromising the integrity of the manuscript. We will address this concern as follows:

1. Phenotypic Effects of Climate Change:

In the previously described experiments, we did not consider the impact of climate change on the observed phenotype. This decision was made for two main reasons. First, the phenomenon of ectoparasites manipulating host behavior has been largely unexplored. As an exploratory study, we aimed to take a step-by-step approach to uncover this phenomenon. Second, we acknowledge that climate change could influence the dispersal parameters obtained from our experiments, just as it may affect host reproductive output and mortality rates. However, these factors are beyond the primary focus of our study. Instead, our findings provide a foundation for future research to investigate how climate change might influence this hypothesis.

2. Integration of Climate Change in the Range Modeling:

It is essential to introduce the concept of climate change in the "Reduced Occupancy in Flea+ hosts" section because projecting the suitable habitat of striped hamster under climate change serves as an input parameter for the RangeShiftR model. The central theme that connects our study is "**ectoparasites reducing host exploration,**" but our ultimate objective is to answer whether "**ectoparasites limit host distribution to avoid environmental heterogeneity**". After establishing how ectoparasites affect host exploration, it is necessary to scale up our analysis to a broader ecological context. Conducting manipulative experiments at this scale is unrealistic, making the RangeShiftR model an essential tool. The RangeShiftR model is an individual-based mechanistic model that simulates host growth, dispersal, mating, reproduction, and mortality across different habitats to estimate species distribution and density. This model requires two types of input data:

- **Habitat quality data**, derived from climatic variables, elevation, and historical occurrence records to predict the species' suitable habitat.
- **Species parameters**, including demographic data and dispersal parameters.

To ensure robustness, we analyzed two climate scenarios: static climate and dynamic climate (accounting for climate change). Although both scenarios yielded consistent conclusions, incorporating climate change further extends our hypothesis and strengthens the reliability and credibility of our findings.

We acknowledge that certain descriptions in the manuscript may have led to confusion. Therefore, in the revised version, we provided additional background information and clarified the connection between ectoparasite-induced host behavioral changes and climate-driven range shifts to enhance clarity [L276].

Comment 22

Lines 279-281: "Present study demonstrates that reduced exploratory behavior is linked to microglial activation and abnormal differentiation of GABAergic neurons in the PFC." This conclusion seems a bit stretched.

Response:

Thank you for the reviewer's feedback. We understand the concerns regarding this conclusion. Although some studies suggest that microglial activation and damage to GABAergic neurons can lead to reduced exploratory behavior in mice, our data do not support a direct causal relationship, and we recognize that this is a major issue in our

study. In the revised version, we presented this result more cautiously to avoid over-speculation. In the results section, we replaced some of the inferential statements with more correlational descriptions. Additionally, we will further explore the potential mechanisms underlying these findings in the "DISCUSSION" section [L335].

Comment 23

Lines 296-297: "...interconnectedness of ecological and public health systems under the One Health framework." This concept has not been adequately elaborated in this work.

Response:

Thank you for the reviewer's feedback. Our response is similar to the one provided for Comment 3.

Comment 24

Line 328. Were 150 new fleas added weekly?

Response:

Thank you for the clarification. Yes, by introducing 150 new fleas at the start of each week and cleaning the bedding at the end of the week, which removes any surviving fleas, we ensure that the infection pressure remains constant and chronic throughout the experiment. This approach allows us to maintain a consistent level of flea infestation and ensures the mice experience ongoing stress from the flea bites.

Comment 25

Line 340. So was it assumed that at the end of the 4-week period, all wild-caught fleas were dead?

Response:

Thank you for the reviewer's question. In fact, all the fleas have died.

First, upon capturing the wild striped hamsters and bringing them to the lab, we anesthetized them and used a comb to remove any ectoparasites from their fur. This is the most commonly used method for cleaning ectoparasites from rodents. In addition, we housed the hamsters individually in the cages for 4 weeks to ensure that any fleas that might have been left behind in the previous step were completely eliminated. Since the average lifespan of adult fleas after feeding is about 30 days, this time period allowed for the complete clearance of any remaining fleas.

Comment 26

Line 344. Is the "study area" referring to Xilinhot, Inner Mongolia, China?

Response:

Thank you for the reviewer's comment. The "study area" refers to the site where we captured the striped hamsters, specifically the W4 site (West Ujimqin Banner), which is located near Xilinhot, both of which are part of the Xilin Gol League in Inner Mongolia, China. We have mentioned this in the "Wild Striped Hamsters Husbandry" section. We understand that this phrasing may confuse readers, so we will revise the description as follows:

“The primary flea species in W4 is the Neopsylla bidentatiformis based on morphological characteristics”

Comment 27

Including common names of the organisms would be helpful.

Response:

Thank you for your suggestion. We understand that including common names for species, such as "striped hamster" for *Cricetulus barabensis*, would be helpful for readers' understanding. However, as far as we know, the flea species mentioned in this study only have scientific names and do not have commonly used names.

Comment 28

Line 499. Change "detection" to "analysis".

Response:

Revised accordingly.

Comment 29

Line 499. UHPLC-QTRAP - Specific name of instrument and vendor name required.

Response:

Thank you for your reminder. The changes we made in the manuscript are as follows:

“Sample analysis was performed using LC-ESI-MS/MS (UHPLC-Qtrap) on a ExionLC AD system coupled with a QTRAP® 6500+ mass spectrometer (AB Sciex, USA)”

Comment 30

The MS method needs to be described in the "Neurotransmitter Metabolomics Analysis" section (line 492) of the Methods.

Response:

Thank you for the reviewer's suggestion. We have incorporated the MS method into the revised manuscript [L572].

Comment 31

Line 506: More details on developing the quantitative method are required.

Response:

Thank you for the reviewer's suggestion. We have incorporated the quantitative method into the revised manuscript [L583].

Comment 32

Line 516: The authors describe using a threshold of 1.5. What was the rationale behind this?

Response:

Thank you for your question. The choice of a 1.5 threshold was primarily based on common standards found in the literature(PMID: 38657519, 37659294).

Comment 33

Line 524: PBS acronym needs to be spelled out at first mention and the acronym placed in parenthesis. Same for TBST (line 532), DAPI (line 536), FBS (line 553), EDTA (line 554), RMPI (line 556), FVS700 (line 562), DMSO (line 563), RIPA (line 579), PMSF (line 580), PVDF (line 583), PAGE (line 583), GAPDH (line 584), cDNA (line 596) and others.

Response:

Thank you very much for your valuable feedback. We truly appreciate your attention to detail regarding the use of acronyms in our manuscript. We have carefully reviewed the Nature Communications author guidelines regarding acronym usage. While we note that some commonly used abbreviations in the field (such as PBS, DAPI, FBS, etc.) may not require full spelling out at first mention (PMID: 40000641, 40011426). Thank you again for bringing this to our attention.

Comment 34

Line 578: "...approximately the size of a mung bean..." It is best to give approximate size measurements in mathematical units.

Response:

Thank you for your reminder. The changes we made in the manuscript are as follows:

"A tissue sample approximately the size of a mung bean (25 ~30 mg) was placed in a 2 ml EP tube and mixed with 30 μ l of RIPA lysis buffer containing protease inhibitors (1 mM PMSF and 1 mM sodium orthovanadate)."

Comment 35

The figures within figures are too small. Each figure composite is too densely-packed. The authors should consider sending some of the 'figures within figures' to Supplementary Information.

Response:

Thank you. We have revised the figures according your suggestions.

Comment 36

Line 614: Cytokine Analysis. Was this analysis performed on both tissue and blood? Please clarify.

Response:

Thank you for your insightful question. The cytokine analysis was indeed performed on both tissue and blood samples. We assessed cytokine levels in the tissue (specifically from the skin and brain-PFC) and blood (serum) to evaluate systemic inflammation as well as localized immune responses. We will make this clarification in the manuscript to ensure the readers fully understand the scope of our cytokine analysis.

In the revised manuscript, we will modify the section to read as follows:

"Cytokine levels were measured in both tissue samples (skin, PFC) and serum."

Reviewer #2 (Remarks to the Author):

In the manuscript entitled "Ectoparasites Enhance Survival by Limiting Host Dispersal through Microglial Activation," the authors investigate the impact of flea bites on mice and striped hamsters in a multiscale framework, from behavioral changes in terms of dispersion to gene expression in both skin and brain tissues. Their results suggest an impact of flea bites on the dispersal behavior of the host that is mediated by an alteration of their neurobiology, notably through microglial activation in the PFC. They further tested the impact of flea bites on current and predicted host distribution and demonstrated that flea bites tend to reduce the host's geographical range.

This study addresses a very interesting topic: host manipulation by ectoparasites, which is far less documented than manipulation by endoparasites. The data they produce to document such alterations of host behavior and neurobiology induced by flea bites is very impressive in quantity and diversity. The range of methods used is also astonishing, going from behavioral experiments in controlled setups to field experiments, transcriptomics, metabolomics, and species distribution models. Such a diversity and complexity of approaches is truly original and allows for a very in-depth investigation of the impact of an ectoparasite on its host. I would like to congratulate the authors on such work done.

Below are my major comments, followed by some line-by-line, more minor comments.

Comment 1

While the quantity and diversity of approaches add great value to the manuscript, it is also, to me, the main drawback of the study. It is often difficult to follow how the different experiments and datasets used are articulated with one another. To help with this, hypotheses could be stated at the end of the introductory paragraph and could be more clearly articulated between the different sections to provide a better sense of progression regarding the main questions throughout the manuscript. A schematic summarizing how the authors envision the way flea bites can alter host dispersal could also be added, which would show how each aspect is tested.

Response:

Thank you for the reviewer's valuable suggestions and recognition. We agree that

the logical flow of the experimental design in the manuscript can be further improved.

In the revised version, we will add a clear research hypothesis in the INTRODUCTION to help readers better understand the research objectives. Additionally, following the reviewer's suggestion and the guidelines of *Nature Communications*, we have revised the final paragraph of the INTRODUCTION to summarize our key experimental findings [L66]. Furthermore, we adjusted the structure of the manuscript, incorporating transitional statements where necessary to enhance the logical connection between the experiments and make the paper more cohesive. To further clarify the experimental design, we included a schematic diagram to visually represent how flea bites affect host dispersal, clearly indicating how each experiment validates the related hypothesis-Figure1. This provided a more intuitive understanding of the experimental progression and the underlying logic. We appreciate the reviewer's constructive feedback, and we believe these changes will improve the clarity and coherence of our manuscript.

Comment 2

Still linked to the fact that the impact of flea bites on the host is investigated at different levels (from behavioral or transcriptomic responses in brain and skin tissue, for instance), I noticed that the authors often claim causal links between flea bites, brain changes, and dispersal behavior. While they provide strong evidence of associations between these, no causal link has been established. However, the authors tend to argue so. More nuance would be appreciated. For example, in line 128: There is evidence that flea bites are associated with decreased dispersal behavior and that flea bites are associated with changes in brain gene expression; however, no test was done to establish that these brain alterations cause changes in dispersal behavior. See line 165, for instance, too. Some nuance could be brought.

Response:

Thank you for your careful review and valuable feedback. We agree that the causal relationship between flea bites, brain changes, and dispersal behavior may have been overemphasized in the manuscript. We will revise the relevant descriptions to clarify that the observed relationships represent associations rather than causal links. Additionally, we will provide more detailed explanations of these associations in the DISCUSSION section. Specifically, we will clarify that while significant associations were found between flea bites, changes in dispersal behavior, and brain gene expression, our current study did not directly test whether brain alterations cause behavioral

changes. We will further discuss this limitation and suggest that future studies could use causal inference methods (e.g., experimental manipulation) to investigate these potential causal relationships.

Comment 3

The authors focus on what they refer to as exploratory behavior as the main altered phenotype caused by flea bites. I am curious about how they distinguish between a reduction in exploratory behavior and reduced locomotion. In most of their setups, it seems that the two could overlap. I raise this question because previous studies have shown that parasites, by depleting host resources, can decrease their overall locomotion. Have you investigated the expression of molecular functions related to metabolism (for instance, in the skin transcriptomic data) that would be enriched, for example, in downregulated genes?

Response:

Thank you for the reviewer's thoughtful comments and concerns. We fully understand the reviewer's point regarding the systemic negative effects that flea bites can have on the host, leading to a sickness state. This could, in some cases, result in a simultaneous decrease in both exploratory behavior and overall physical activity.

The Open Field Test (OFT) is one of the most well-established behavioral assays and works by assessing the balance between a mouse's curiosity to explore the central area and its instinct to seek safety by staying near the walls, which serves as a measure of anxiety. In our study, we used the time spent in the central area and the frequency of entries into the central zone as metrics to measure exploratory behavior, while the total distance traveled in the open field was used as an indicator of overall activity. Importantly, there were no significant differences in this activity measure between groups, suggesting that flea bites primarily affect exploratory behavior rather than overall locomotion.

We are also aware of research suggesting that certain parasites reduce the physical capacity of their hosts, such as affecting the muscle tissue in birds' hearts, chest, and wings. This reduction in movement can lead birds to stop more frequently during migration, increasing the chance of parasite/pathogen contact with other hosts and enhancing transmission. This suggests that different parasites may adopt different strategies to enhance their survival by manipulating their hosts in ways that ultimately benefit the parasite's propagation.

In our study, the gene sets downregulated in the skin transcriptomic data did not

show significant enrichment in metabolic functions. While it's possible that exploring other tissues, such as heart or thigh muscle tissue, could reveal changes related to metabolism, we believe that the primary focus of this study remains on the impact of ectoparasites on the host's exploratory behavior.

Nevertheless, we appreciate the reviewer's suggestion, which has provided us with a valuable new perspective. We plan to explore the broader effects of ectoparasites on host physical activity in future studies.

Comment 4

I have some difficulties with the enclosure experiment conducted on striped hamsters. First, I don't understand why the non-capture rate is a good proxy for the non-dispersal rate. If the hamster escapes the enclosure, then one could argue that it has dispersed. Experimental setups with enclosures linked by dispersal corridors seem more appropriate as they would allow a direct measure of dispersal. This issue also affects the future host range simulations, as the dispersal rate used was the one calculated from the enclosure experiment. Finally, in the Extended Data Fig. 8 enclosure experiment schematics, it seems that both the number of hosts and their sex ratio vary between rounds. However, rounds also vary regarding whether hamsters are infected or not. Considering the sample size suggested in the schematics, I doubt this would be enough to properly distinguish between sex, population size, and infection status effects on dispersal. Wouldn't it have been more straightforward to simply compare infected versus non-infected populations in terms of dispersion?

Response:

Thank you for the detailed feedback. We understand the reviewer's concerns regarding whether the capture rate accurately represents the dispersal rate, and we are happy to further clarify our experimental design.

As mentioned in our response to **Comment 3**, the primary focus of this study is on the impact of ectoparasites on host exploratory behavior. The most authoritative and intuitive measure of exploratory behavior is the time spent in the center of the open field test (OFT). We conducted open field tests in both indoor mouse and black-tailed hamster infection experiments, and verified that flea bites reduce host exploratory behavior. In order to extrapolate our hypothesis, an important step was to conduct controlled experiments under natural conditions to test whether flea bites would still reduce host exploration. Thus, we moved the open field test setup outdoors by constructing enclosures (30m x 30m) on the natural grassland habitat of black-tailed

hamsters. The goal of these enclosure experiments was to assess the differences in the time spent in the central area of the enclosure between Flea+ and Flea- groups of black-tailed hamsters.

We hypothesized that, since rodents tend to avoid open spaces and prefer the edges of enclosures, hamsters that spend more time at the enclosure's edge (indicating higher anxiety and lower exploratory behavior) would be more likely to find and escape through gaps in the enclosure, while those spending more time in the central area (indicating lower anxiety and higher exploratory behavior) would be more likely to stay within the enclosure. Therefore, we used the capture rate to assess differences in exploratory behavior.

We acknowledge that this is an original method based on extrapolating laboratory-based tests, but we believe it is theoretically feasible. To test the feasibility of this hypothesis, we first conducted a round of experiments with only black-tailed hamsters and no fleas. After a 4-week period, we captured 29 out of the 30 released hamsters (one escaped). This result demonstrated that in the absence of flea infection, most hamsters stayed within the enclosure, supporting our hypothesis that non-infected hosts exhibit higher exploratory behavior.

We then conducted a second round of experiments, with enclosures A1 and B1 designated as the Flea- group, and the other enclosures as the Flea+ group. While we recognize that random enclosure selection would improve the reliability of our conclusions, we were concerned that hamster movement between enclosures could introduce fleas into the Flea- enclosures, so we chose enclosures on the edges to minimize this risk. The results were as expected: the capture rate was higher in the Flea- enclosures, and lower in the Flea+ enclosures, confirming that flea bites reduce host exploratory behavior.

To further validate our hypothesis, we conducted a third round of experiments, where all enclosures were treated with fleas to rule out potential quality issues in the A1 and B1 enclosures (Flea- group) from the second round. The results showed that the capture rates in all six enclosures were reduced, with no significant difference between enclosures. This indicates that even under natural conditions, flea infestation reduces host exploratory behavior.

Regarding the reviewer's suggestion that "experimental setups with enclosures linked by dispersal corridors seem more appropriate as they would allow a direct measure of dispersal," we understand that the reviewer wants to capture a more direct

measure of hamster dispersal. This method is commonly used for studying dispersal in insects or organisms like ciliates, but we believe it is not suitable for our study. Please allow me to explain based on the mechanism of the RangeshiftR model. When constructing the RangeshiftR model, dispersal is divided into three stages/parameters: (1) the likelihood of leaving the birthplace, which reflects exploratory behavior, (2) migration to new habitats, which is reflected in movement ability, and (3) colonization in new habitats. We believe that using the suggested method would combine the first two parameters, which could reduce the model's precision. Therefore, we continue to use capture rates as an evaluation metric. In the RangeshiftR model, the Flea- and Flea+ groups differ only in the proportion of individuals leaving the birthplace, which directly impacts the host distribution range and density.

We appreciate the reviewer's valuable suggestion and will continue to focus on the various negative impacts of flea bites on hosts (such as reduced mobility, reproductive performance, and increased mortality). We believe that these additional negative impacts will further enhance the differences between Flea- and Flea+ groups in our models (such as in host distribution range and density).

Finally, regarding the reviewer's comment on "the number of hosts and their sex ratio varying between rounds in the Extended Data Fig. 8 enclosure experiment schematics," we agree that a larger sample size and more consistent host factors would improve the reliability of our conclusions. However, enclosures are relatively enclosed spaces, so the number of hamsters that can be housed is limited, and we must balance density effects. For sample size, we calculated the maximum density of hamsters in a single enclosure based on field sampling, and given that black-tailed hamsters are solitary animals with strong territorial instincts, the appropriate density helps avoid excessive competition between individuals.

Regarding the issue of varying numbers and sex ratios between rounds, this is mainly due to practical limitations in the field. The grassland in Xilin Gol, Inner Mongolia, where we conducted our experiments, has long winters, and field experiments can only be conducted between June and October. During this time, we capture wild black-tailed hamsters and fleas to establish experimental animal populations, then conduct the enclosure control experiments. Additionally, black-tailed hamsters are difficult to breed in captivity, so most of our samples come from wild populations. Therefore, we must rely on the most reliable data we can gather within a limited timeframe. Despite these constraints, we believe our analysis remains valid. We

conducted independent analyses for each round, especially for the critical second round, and performed statistical analyses based on both enclosures and treatments. Furthermore, we performed individual-based GLMM analysis for all the data across the three rounds, as suggested by the reviewer “Wouldn’t it have been more straightforward to simply compare infected versus non-infected populations in terms of dispersion?” Our results indeed show that Flea- groups had a higher capture rate.

Comment 5

I had some issues regarding figure readability. The text is often too small to read, making it hard to confirm what is claimed. On this same note, I think more metrics could be provided, especially regarding the transcriptomics, such as count data, log fold changes, and corrected p-values. The same goes for the results of the enrichment tests. Plots are a good visual option, but more statistical metrics should be made available to ensure the results' trustworthiness.

Response:

Thank you for your feedback, and apologies for the inconvenience caused. We have updated the quality of the figures to ensure that the text is clearer and easier to read, allowing for better presentation of our results. Additionally, as per your suggestion, we have updated the supplementary material with transcriptomics data, including count data, log fold changes, and corrected p-values, and have provided more statistical metrics for the enrichment tests. We used p-values rather than corrected p-values to screen for differentially expressed genes because the corrected p-value approach resulted in too few significant genes, which would limit the biological interpretation of our data. This approach has also been adopted in other studies under similar circumstances: .

- PMID: 39106567

Differential expression analysis of mRNA sequencing: The DESeq2 (R package, v1.32.0) was used to evaluate differentially expressed genes, including differential gene expression mRNA (DEGs). To identify as many transcriptome changes caused by COS as possible, the standard for identifying DEGs was $pvalue < 0.05$.

We believe these updates will further enhance the trustworthiness and biological relevance of our results. Thank you again for your valuable suggestions.

Comment 6

Finally, the discussion is rather short and superficial, considering the numerous

aspects of host alterations caused by flea bites. Again, I would like to see clear links made between the different dimensions of the paper. See my comment above about a general schematic.

Response:

We sincerely appreciate the reviewer's suggestion regarding the discussion section. Due to space limitations of *Nature* as we mentioned above, our initial discussion was relatively brief and did not explore certain aspects in depth. In the revised manuscript, we have expanded and refined the DISCUSSION section, providing a more detailed analysis of parasite manipulation strategies and the molecular mechanisms through which parasites influence host behavior. Additionally, we have integrated findings from previous studies to systematically elucidate the biological mechanisms underlying parasite manipulation. We hope that the updated discussion will provide greater clarity and meet the reviewer's expectations. Thank you for your valuable feedback!

Minor Comments:

Comment 7

L. 60: Clarifying the method used to measure dispersal behavior would improve overall clarity (e.g., "We assess dispersal behavior by testing the time spent in an open field experimental setup and in an elevated maze...").

Response:

Thanks for your suggestions. We have revised the manuscript according to your request.

Comment 8

L. 89: Please provide the number of DEGs. Also, log fold changes and corrected p-values should be provided in the supplementary material. The same applies to enrichment tests.

Response:

We have added the number of DEGs in the manuscript [L123] and updated the supplementary material with the statistical results of the transcriptomic data, including log fold changes and p-values. Additionally, the results of the enrichment analyses have also been included in the supplementary material. Thank you for your suggestion!

Comment 9

L. 68-125: I agree with the authors that the mentioned brain gene expression

differences could be linked to emotion regulation and, as such, to anxiety. However, with this analysis, no causal link can be evidenced. I suggest emphasizing more on altered brain expression (since this is the main result) rather than on the functional consequences that cannot be verified. More clarity on the hypotheses tested could help.

Response:

Thank you for your suggestion. We agree that our current analysis does not provide direct causal evidence linking gene expression changes to anxiety. Therefore, in the revised manuscript, we will place more emphasis on the observed alterations in brain gene expression rather than speculating on their functional consequences. In the results section, we replaced some of the inferential statements with more correlational descriptions, as mentioned at **Comment 2**.

Comment 10

L. 104-106: I don't fully understand how the GSEA suggests that the PFC is more impacted than the HC and TH. Typically, this analysis investigates impacted functions but does not indicate the extent of alteration. DEG analysis might provide more insight into this.

Response:

Thank you for your suggestion. We apologize for the confusion caused by our oversimplified description. We also agree with your point that GSEA typically investigates impacted functions but does not indicate the extent of alterations. However, in our study, we aimed to explain the greater impact on the PFC by the number of enriched functions, similar to the DEGs analysis.

We also apologize for the mistake in the figure legend, which should be "(Extended Data Fig. 2g, 3g, 4g)." In the updated version, we performed GSEA analysis on three brain regions and selected the significantly different functions related to neural functions. We found that these functions were significantly suppressed in the Flea+ group. The specific statistical results will be provided in the supplementary material. By counting the number of functional pathways, we observed that the PFC had significantly more functions (35) compared to TH (14) and HC (0). Therefore, like the DEGs analysis, we believe GSEA analysis similarly suggests that the PFC may be the most affected brain region by flea bites [L137].

Once again, we appreciate your valuable feedback and will refine these details in the revised version.

Comment 11

L. 142-143: In which tissue was the expression of these neurons investigated? This is not very clear.

Response:

Thank you for pointing this out. The expression of these neurons was investigated in the prefrontal cortex (PFC). We have revised the manuscript to clarify this detail and specify the tissue in question [L163]. We appreciate your careful review.

Comment 12

L. 156: Please provide a reference. Also, does this imply that flea bites do not directly increase anxiety, as previously stated? A different mechanism seems to be proposed here.

Response:

Thank you for your suggestion. We have added the relevant reference [L191]. Additionally, this is the neuro-molecular mechanism we are exploring regarding flea-induced anxiety in mice. Flea bites may induce anxiety through the activation of microglia in PFC and damage to GABAergic neurons. The damage to GABAergic neurons is likely mediated by excessive activation of microglia, leading to the pruning of neuronal axons.

Comment 13

L. 187-189: Please elaborate on the rationale behind this statement.

Response:

Thank you for your valuable comment. The observation that HE staining did not reveal significant skin damage suggests that flea bites induce only mild inflammation in mice. This aligns with the transcriptomic findings, where the downregulated gene set did not show significant functional enrichment.

In cases of severe inflammation or tissue damage, one would typically expect strong changes in gene expression, particularly enrichment of pathways related to immune responses, wound healing, or cell death. However, the lack of such enrichment in the downregulated gene set indicates that the inflammatory response to flea bites is relatively mild. We believe this correlation between histological and transcriptomic findings supports the conclusion that flea bites cause mild inflammation rather than severe pathological changes. We hope this clarification addresses your concerns.

Comment 14

L. 197-199: Explain in more detail why you support this explanation, even though the PCR does not confirm it.

Response:

Thank you for your insightful comment. In our study, we observed a decrease in Claudin-1 protein levels in the PFC based on Western blot (WB) analysis, while qPCR results showed an increase in Claudin-1 mRNA expression. We acknowledge this apparent discrepancy and would like to clarify the possible underlying mechanisms.

Protein expression levels are influenced not only by transcriptional regulation but also by post-transcriptional mechanisms, including mRNA stability, translational efficiency, and protein degradation. It is possible that despite the upregulation of Claudin-1 mRNA, post-translational modifications or increased protein degradation led to a net reduction in Claudin-1 protein levels. Similar discrepancies between mRNA and protein levels have been reported in various biological contexts due to differences in regulatory processes.

Given that functional outcomes are more directly linked to protein levels, we believe that the WB results provide a more relevant measure of Claudin-1's role in blood-brain barrier integrity.

We appreciate the reviewer's attention to this detail, and we have revised the discussion section to clarify this point.

Comment 15

L. 275: Should it be "exploratory behavior" instead of "explanation behavior"?

Response:

Thank you to the reviewer for pointing out this spelling error. We have corrected it in the manuscript.

Comment 16

L. 211: Please mention if you detected an effect of infection on the time spent in the central area of the open field.

Response:

Thank you for the reviewer's comment. We did mention that "female striped hamsters spent significantly less time in the central area," but what we intended to express was that the Flea+ group of female striped hamsters spent less time in the central area, whereas there was no difference in the time spent in the central area

between the Flea- and Flea+ groups of male striped hamsters. We apologize for the misunderstanding caused by our wording.

To ensure clearer expression, we have revised the manuscript to explicitly highlight the effect of infection on the time spent in the central area, specifying the differences between the female and male hamsters. The modified sentence reads: *“Compared to the Flea- group, female striped hamsters in the Flea+ group spent less time in the central area; however, there was no difference in the time spent in the central area between the Flea- and Flea+ groups of male striped hamsters.”* [L247]

Comment 17

L. 229: If I understand correctly, this is true only for the second round. The way results are presented tends to blur the interpretation related to the stated hypothesis. The reader would want to know if there was a difference between infected and non-infected hamsters regarding the non-recapture rate. Also, results could be presented as a time series rather than as three separate outputs, with the overall dynamics over time tested.

Response:

Thank you for the reviewer’s feedback. We agree with the statement that "this is true only for the second round," but as we pointed out in our comment, the data from the first and third rounds actually enhance the reliability of the results from the second round. Additionally, we have shown the difference between infected and non-infected hamsters regarding the non-recapture rate in Extended Data Fig. 8i.

We also share the reviewer’s concern that presenting the results using the non-recapture rate as an indicator might blur the hypothesis related to changes in exploratory behavior. However, we chose this expression because we believe it is important to present the raw results in the middle of the paragraph, while at the same time, we provided explanations and summaries regarding the relationship between non-recapture rate and exploratory behavior at both the beginning and end of the paragraph.

In fact, the results shown in Extended Data Fig. 8j are based on the GLMM model test of the overall dynamics across the three rounds at the individual level. We agree with the reviewer that presenting the integrated analysis of all three rounds of experiments will provide a more comprehensive view of the results.

Comment 18

L. 233: The results indicate that the non-capture rate increases for infected hamsters (L. 229), which seems contradictory to the statement that "flea bites reduce

host exploratory behavior, thereby decreasing the likelihood of the host leaving its natal habitat." If this were the case, I would expect the non-capture rate to increase for non-infected hamsters and remain the same for infected ones.

Response:

Thank you for the reviewer's question. We have provided a detailed response in comment 4. In brief, the enclosure experiment was conducted in a larger open field. Based on our observations, infected striped hamsters showed reduced exploratory behavior, increased activity in the peripheral areas, and a higher likelihood of leaving the enclosure, which led to a higher non-recapture rate.

Comment 19

L. 241: Please clarify that W4 is a site.

Response:

Thank you for the reviewer's suggestion. We have made the following modification in the manuscript:

"Over three years and nine sampling events, the highest number of striped hamsters was captured in W4 sample site," [L281]

Comment 20

L. 279-281: Again, this statement is somehow confusing, as this is not what is tested. The effect of flea bites is tested. Please revise the phrasing, as it is misleading.

Response:

Thank you for your valuable feedback. We have revised the manuscript to clarify the phrasing and ensure it accurately reflects the focus of the study. The revised statement is as follows [L344]:

"Present study demonstrates that reduced exploratory behavior might link to microglial activation and abnormal differentiation of GABAergic neurons in the PFC."

We hope this revision more clearly conveys the intent and findings of our research. Thank you again for your careful review.

Comment 21

L. 187: Please elaborate on the rationale here.

Response:

Thank you for the reviewer's feedback. This comment is the same as **comment 13**,

and we have already addressed it.

Comment 22

L. 356: Why infecting hamsters during the daytime if they are nocturnal?

Response:

Thank you for your question. In fact, we modeled the experimental design based on the natural diurnal behavior of striped hamsters. These hamsters are nocturnal, spending the daytime in their burrows, where fleas are evidently more abundant. Therefore, we infected the striped hamsters during the daytime and returned them to their individual cages at night, allowing them to move freely.

Comment 23

L. 375-377: This hypothesis is not very intuitive. Why not measuring the time spent near the edge directly, as escaping shows exploratory tendencies? Alternatively, use a different setup to measure dispersal directly.

Response:

Thank you for your comment. As we explained in comment 4, the enclosure is 30m x 30m, which is quite large. Additionally, there is no suitable equipment available to monitor the nocturnal activity of striped hamsters. Furthermore, the focus of this study is on how flea infection affects the exploratory behavior of the host, rather than their migratory abilities (i.e., leaving their birthplace and migrating to a new habitat).

We appreciate your feedback and hope this clarifies our approach.

Comment 24

L. 384: Were non-infected hamsters also placed in a 5L glass beaker to control for potential behavior changes, particularly increased stress levels?

Response:

Thank you for your comment. Yes, we performed the same release procedure for the non-infected hamsters, except they were not infected with fleas. We clarified it in the revised manuscript [L457].

Comment 25

L. 417: Please specify mice and hamsters.

Response:

Thank you for the reviewer's suggestion. We have made the following

modification in the manuscript:

“The mice/striped hamster were then introduced into the arena and allowed to explore for 3 (mice) / 5 (striped hamster) minutes while being tracked.

Comment 26

L. 456: Please provide access to the code for the custom package. Also, throughout the manuscript, provide software versions and references.

Response:

Thank you for your suggestion. We have updated the manuscript to include the software versions and references for all software used to ensure clarity.

Comment 27

L. 186: Please provide the reference and the update date of the database used.

Response:

Thank you for your suggestion. We did not find any reference to the database in line 186. We believe you are referring to the description of the GO database in line 486. In the revised manuscript, we have updated the relevant information and provided the reference and update date for the database.

Comment 28

L. 669: Please provide the total number of sites used for the model.

Response:

Thank you for your suggestion. We used 562 occurrence records of *Cricetulus barabensis* to build the species distribution model, ensuring its reliability. This information has been updated in the manuscript.

Comment 29

L. 721-723: You mentioned that striped hamsters disperse upon reaching maturity (L. 700). In your simulation, this specific dispersal event is considered. However, you deduced the dispersal rate from field experiments conducted on adult striped hamsters. How can you ensure that the dispersal observed in your field experiment is comparable to dispersal occurring from the juvenile to adult stage?

Response;

Thank you for your comment. We apologize for any confusion caused by our wording. What we intended to express was the dispersal of striped hamsters after they

reach adulthood (after 2 months of age), rather than the transition period from juvenile to adult (which is difficult to define). Additionally, the dispersal parameters we provided in our model only reflect the differences during the adult stage, specifically derived from the enclosure experiment. We have revised the original text to avoid any further confusion for the readers, as shown below [L833]:

“Dispersal in striped hamsters exhibits a stage-dependent pattern, predominantly occurring after they attain adulthood (beyond 2 months of age), during which they leave their natal areas to establish themselves in new habitats.”

Comment 30

L. 749: Six enclosures might be too few to estimate enclosure-associated variance as done in a GLMM.

Response:

We sincerely thank the reviewer for their insightful comments. We fully understand your concerns and agree that a larger number of enclosures, such as 9-12, could potentially provide more robust results. However, to the best of our knowledge, when constructing GLMM models, ensuring at least 5 effect values is sufficient to maintain the model's efficacy (PMID: 29844961). While we acknowledge the benefits of additional enclosures, our current design meets the minimum requirements for reliable statistical analysis. We appreciate the reviewer's suggestion and will consider expanding the number of enclosures in future studies to further enhance the reliability and generalizability of our findings.

Comment 31

L. 872: Please clarify that the PCA is performed on gene expression data.

Response:

Thank you for the reviewer's suggestion. We have made the following modification in the manuscript:

“Principal component analysis of all transcriptome data showing tight clustering of brain regions, with each region color-coded according to the legend.” [L1055]

Comment 32

L. 874: It should be "significantly differentially expressed genes" rather than "different genes."

Response:

Thank you. We revised accordingly.

Comment 33

Extended Data Fig. 6 and 8: The figure is difficult to read. The panel letters are disproportionately large compared to the figure content.

Response:

We sincerely thank the reviewer for pointing out this issue. We have carefully revised all Figures to improve readability.

Comment 34

Extended Data Fig. 8: I don't fully understand the experimental setup from the schematic. Since the number of hamsters and the sex ratio vary between enclosures, how can you be sure that differences in dispersal are due to the treatment? Please clarify the hypothesis tested and how the setup addresses it.

Response:

We sincerely thank the reviewer for their valuable feedback. A detailed response to the comments, including the specific revisions made to address the concerns, can be found in **Comment 4**.

Comment 35

Data Availability: I noticed that the raw fastq reads from transcriptomic and neurotransmitter datasets are available. However, other data, such as occurrences and environmental data used for the SDM, should also be made available. Additionally, count data from the transcriptomic analysis, along with log fold changes and corrected p-values, should be provided.

Response:

We sincerely thank the reviewer for their attention to the issue of data availability. In response to this comment, we have updated the supplementary materials to include most of the requested data, with the exception of the environmental variables. The environmental data were obtained from publicly available databases, and we have included the relevant access URLs in the Methods section to ensure transparency and facilitate access for readers [L762]. We believe these updates address the reviewer's concerns and enhance the reproducibility of our study. Please refer to the updated supplementary materials and the Methods section for further details.

Reviewer #3 (Remarks to the Author):

Overall comments:

In this manuscript, Liu, et al investigated the effect of ectoparasitic infection on host dispersal and the underlying machinery of immune interactions. They found ectoparasites can alter metabolic activity in specific brain regions of mice and decreased their exploratory behavior. In addition, systemic inflammation and increased microglia activation were observed in mice with ectoparasitic infection.

Their work is interesting and provides new insights into the mechanism of ectoparasites influencing host dispersal and the underlying immune interactions. However, some key experiments showed unsolid or controversial data, making their conclusion not convincing. In addition, quite a few small mistakes and mislabeling severely dampen the quality of this manuscript. I will discuss some of this below:

Major points:

Comment 1

Authors don't provide any PET-CT images in the main figures or extended figures. The quantification shown in Fig.1C-D is not convincing without any images. It's strongly recommended to add the image from the Flea- and Flea+ group with the labeling of brain regions (e.g. PMID: 36451231; 28654383). In addition, most of the brain region showed reduced 18F-FDG uptake in mice from Flea+ group (Fig. 1c), reflecting the Loss of neuronal activity. However, recent studies demonstrate anxiety is associated with increased neuronal excitability, which is controversial with authors' data of 18F-FDG uptake.

Response:

We would like to thank the reviewer for their valuable comments. In response to your suggestions, we have made the following modifications and additions:

We have added representative PET-CT images [Figure 2d, f, g, h] for the Flea- and Flea+ groups, with relevant brain regions annotated to provide a more intuitive comparison. Additionally, we have referenced relevant literature to optimize image quality and analysis methods, thereby improving the readability and reliability of the data.

We agree with the reviewer's point that anxiety is associated with increased neuronal excitability, which is consistent with our findings. Specifically, we observed a reduction in GABAergic neurons, which, as inhibitory neurons, lose their ability to reset other

excited neurons, leading to increased neuronal excitability. However, this is not mutually exclusive or contradictory to our observed 18F-FDG uptake data. Firstly, for PET-CT imaging of anxiety, there is no highly specific marker; glucose is typically used to assess the impact of treatment on energy metabolism and glucose uptake. However, brain tissue damage does not necessarily result in reduced glucose uptake, as damage can occur in different stages. For example, compensatory mechanisms post-injury may actually increase glucose uptake to maintain functional intensity. Additionally, as a major excitatory neuron, we observed a significant decrease in synaptic terminal markers of glutamatergic neurons in the PFC of the Flea+ group. This may result in lower overall glucose uptake despite some glutamatergic neurons being in an excited state. Moreover, the increase in excitability we refer to is more about the relative imbalance between excitatory and inhibitory neuronal functions, rather than an absolute quantitative difference between groups. Thirdly, a specific brain region contains various cell types, such as excitatory neurons, inhibitory neurons, and glial cells, each with potentially different metabolic patterns. The excitation of some neurons may have little impact on the overall glucose uptake of the brain region.

In summary, we emphasize that the purpose of our PET-CT is to identify the brain regions most significantly affected by flea bites, whether through increased or decreased glucose uptake, to guide further mechanistic investigations. Given the lack of suitable markers for anxiety, as well as potential regional effects, compensatory mechanisms, and complex cellular composition, we believe it may be inappropriate to directly correlate reduced 18F-FDG uptake with increased neuronal excitability. Nevertheless, we appreciate the reviewer's question and will focus on elucidating the specific mechanisms underlying this phenomenon in future research.

Comment 2

Since authors found significant enrichment in Tyrosine and Tryptophan metabolism pathways in the PFC which are crucial for synthesizing dopamine, norepinephrine, and serotonin, it would strengthen the support to their conclusion if the concentration of these neurotransmitters can be added to main figures.

Response:

Thank you for your valuable suggestion. We have emphasized the changes in amino acid levels in the manuscript: "*Notably, we observed a decrease in tryptophan and serotonin levels in the Flea+ group.*" Additionally, following your recommendation, we have incorporated the concentrations of these neurotransmitters into the main

figures.

Comment 3

In Fig. 1g, it seems authors failed to characterize the microglia and infiltrating macrophages. The increased inflammation in brain may also cause the infiltration of macrophages (PMID: 29133437; 31325960). In addition, the huge increase in the transcriptional level of CD11b and CD68 from Flea+ group can't be reflected by the FACS data (Extended Fig. 6b). A better FACS analysis should be repeated in the revision.

Response:

Thank you very much for your insightful comments and suggestions. We sincerely appreciate your careful review of our manuscript and your constructive feedback. We have carefully read the recommended references, and agree that under normal conditions, microglia are the only resident macrophages in the brain. However, when the blood-brain barrier is disrupted, circulating macrophages can infiltrate the brain, which may have led to our initial inability to distinguish between microglia and infiltrating macrophages in our results.

In response to your suggestion, we have repeated the FACS analysis using an improved gating strategy based on the methods described in the recommended literature. The updated analysis now allows us to better differentiate between microglia and infiltrating macrophages. The detailed FACS analysis strategy and results are provided in Supplementary Fig 14..

We believe that your suggestions have significantly improved the quality of our study, and we are grateful for your guidance. We hope that the revised analysis addresses your concerns adequately.

Comment 4

The signal shown in the TUNEL staining of Extended Fig. 6g is so low and needs to be replaced with pictures with higher quality.

Response:

Thank you for your suggestion. The TUNEL staining in Extended Fig. 6g was performed as a double-labeling with NeuN, and the TUNEL signal itself is inherently low, which is expected given the nature of apoptotic cells in this context. However, we

acknowledge the need for improved image quality, and we have replaced the original images with higher-resolution versions to enhance clarity. We appreciate your feedback, which has helped us improve the presentation of our data.

Comment 5

It has been reported that the activation of microglia resulted in increased GABAergic neurons. (PMID: 17459525; 37806513). Therefore, the reduced GABAergic neurons observed in Flea+ group may not be induced by increased inflammation and microglia activation.

Response:

Thank you for your valuable comments. We have carefully read the two references you provided and analyzed their findings in detail.

In PMID: 37806513, it was reported that neuronal stimulation activates microglia to produce increased levels of neurotrophic factors (NGF, NT-4/5, TGF β 1, GDNF, FGF2, IL-3, and IL-10), enhancing the survival and maturation of catecholaminergic, GABAergic, and cholinergic neurons.

In contrast, PMID: 17459525 demonstrated that LPS-induced microglial activation enhances GABAergic inhibitory synapse formation in the hippocampal CA1 region, leading to cognitive impairment, which can be reversed by blocking GABA receptors or depleting microglia.

We believe these results are consistent with the well-established roles of microglia in the central nervous system (CNS), as microglia are the primary immune cells in the CNS and perform the following key functions:

(1) Immune defense: As part of the innate immune system, microglia recognize and clear pathogens, apoptotic cells, and damaged tissues, while mediating neuroinflammatory responses via the release of inflammatory cytokines (e.g., IL-1 β , TNF- α).

(2) Regulation of synaptic plasticity: Microglia modulate synaptic remodeling by engulfing synapses and releasing cytokines, thereby influencing neural circuit formation and function (corresponding to PMID: 17459525).

(3) Neurotrophic support: Microglia secrete neurotrophic factors (e.g., NGF, BDNF, GDNF) to promote neuronal survival, growth, and synapse formation (corresponding to PMID: 37806513).

(4) Maintenance of neural homeostasis: Microglia continuously monitor neuronal

activity, maintaining CNS homeostasis and rapidly responding to stress or injury.

(5 Involvement in neurodegenerative diseases: In diseases such as Alzheimer's and Parkinson's, excessive microglial activation can lead to chronic inflammation, exacerbating neuronal damage and cognitive dysfunction.

Therefore, we believe that microglia rapidly become activated in response to CNS damage. Depending on the nature of the stimulus, microglia can take a number of activation states, which correspond to altered microglia morphology, gene expression, and function. Furthermore, we have observed cases where excessive microglial activation leads to a reduction in GABAergic neurons, for example: Activated microglia suppress GABA(CeA) neuronal activity by engulfing their dendritic spines (PMID: 38200023); In traumatic brain injury (TBI), GABAergic neuron loss and microglial activation are critical factors in its pathophysiology (PMID: 28701948).

We have further clarified this point in the discussion section and appreciate your suggestions, which have helped us refine our manuscript.

Comment 6

To determine whether the flea bite-induced systemic inflammation affected exploratory behavior of mice through microglia activation, a depletion of microglia or treatments with the activation inhibitor need to be utilized in Flea+ group.

Response:

Thank you for your valuable suggestion. We completely agree and acknowledge that one of the major limitations of our study is the potential overinterpretation of correlation as causation. This study is primarily exploratory, focusing on the behavioral strategy of "ectoparasite manipulation of host dispersal," while the molecular mechanisms serve as supporting evidence for our conclusions. Given the study's focus and complexity, we plan to conduct further experiments in future research to establish causal relationships. To avoid confusion for readers, we have revised the manuscript to use terms like "associated with" rather than implying direct causation.

Comment 7

The study in the exploratory behavior in striped hamster has no supporting mechanistic data. Although authors did a huge amount of work in mouse model, whether the same mechanism can be shared in hamster is questionable. Authors should either use mouse model in natural settings, or provide evidence of the similarity of brain metabolism and inflammation between mice and hamsters.

Response:

Thank you for your insightful comments. We acknowledge your concerns regarding the use of a single species to elucidate the entire mechanism. However, due to practical and technical constraints, implementing such an approach is challenging.

First, striped hamsters are not suitable for laboratory infection experiments to explore molecular mechanisms, for the following reasons:

(1) At the mechanistic level, using mice is a feasible approach. From reference genomes to ELISA kits and various protein markers, extensive resources and mature experimental tools are available. However, such resources are entirely lacking for striped hamsters.

(2) The flea population (*X. cheopis*) we used for laboratory infection experiments in mice has been maintained and adapted to mice over time. If we attempted to conduct an infection experiment in striped hamsters, we anticipate a similar outcome to the *C. felis*-mouse system, where fleas quickly die and fail to maintain infection pressure.

(3) Standardized experimental striped hamsters can only be obtained from Shanxi Medical University, but our previous attempts have shown suboptimal results. Striped hamsters exhibit breeding difficulties in laboratory conditions, have limited population availability, and lack standardized experimental protocols. Additionally, the challenges in acquiring laboratory animals due to the pandemic in recent years have further complicated this process.

Second, conducting field experiments with mice is also problematic:

Mice (*Mus musculus*) are domesticated from house mice, which are commensal species primarily inhabiting human settlements. In grassland environments, the density of house mice is extremely low, making it infeasible to establish sufficient experimental populations for both laboratory infections and enclosure experiments. Additionally, the prairie environment is unsuitable for mice, limiting the ecological relevance and applicability of our findings.

Third, we attempted brain dissections in striped hamsters but faced logistical constraints:

As mentioned in the manuscript, our research site was under pandemic-related lockdown restrictions, and we lacked the necessary dissection tools and sample preservation equipment. This was an unfortunate but unavoidable limitation.

Considering these constraints, we believe our experimental design represents the most practical and scientifically rigorous approach given the available conditions.

Despite these limitations, we maintain that extrapolating from laboratory-infected mice to indoor-infected striped hamsters and ultimately to field experiments is valid for the following reasons:

(1) As we stated in our response to Reviewer #1 (Comment 1), ectoparasites are expected to employ similar manipulation strategies across different hosts, reducing host dispersal to enhance parasite survival.

(2) As we mentioned in our response to your Comment 6, the central theme of our study is the hypothesis that "ectoparasites reduce host exploratory behavior." This behavioral phenotype has been consistently observed across laboratory mouse infection experiments, indoor striped hamster infections, and enclosure experiments. The mechanistic insights gained from mice further strengthen the evidence supporting our hypothesis.

(3) The primary goal of this study is to derive dispersal parameters through controlled experiments and ultimately validate our hypothesis on a macro scale by modeling the impact of flea bites on host distribution. Our findings suggest that flea-induced reductions in host dispersal enhance parasite survival.

Minor points:

Comment 8

The resolution of some Extended figures is so low to identify the labeling of graphs.

Response:

Thank you for your comment. We have now replaced the low-resolution images with high-resolution versions to ensure clarity and readability of the graph labels. Please let us know if you have any further concerns.

Comment 9

The abstract part should not include any references.

Response:

Thank you for your comment. We have now removed all citations from the abstract to comply with the journal's formatting guidelines.

Comment 10

The word "survival" in the Title was misspelled.

Response:

Thank you for pointing this. We have revised it in the title.

Comment 11

The “Introduction” title is missing.

Response:

Revised accordingly.

Comment 12

There are multiple mistakes in Figure numbers throughout the manuscript. For instance, on line 123, it says “In the HC, enrichment of the Histidine metabolism pathway (Extended Data Fig. 5j)”. However, the related figure is Extended Data Fig. 5k. In the figure legend of extended Fig. 1e, it says “Spearman correlation of 18F-FDG uptake between different brain regions in the Flea- group (n=6)”. However, the labeling in this figure is “Flea+”.

Response:

Thank you for your careful review. We appreciate you pointing out these inconsistencies in figure numbers and labels. We have carefully checked and corrected all figure numbers and labeling errors throughout the manuscript to ensure accuracy and clarity.

Submission ID: **NCOMMS-24-65632B**
point-by-point response to reviewers

Reviewer #1 (Remarks to the Author):

I feel that the authors have adequately addressed my comments on the manuscript.

Response: We sincerely appreciate your time and effort in reviewing our manuscript. We are grateful for your constructive feedback, which has helped us improve the quality and clarity of our work. We are pleased to hear that our revisions have adequately addressed your concerns. Thank you again for your insightful comments and for supporting the improvement of our manuscript.

Reviewer #2 (Remarks to the Author):

Overall, the authors have carefully addressed the previous reviewer comments, and the manuscript is now much clearer and more consistent. I have only minor comments that could further improve clarity:

Response: Thank you for your positive feedback. We appreciate your acknowledgment of our revisions and your additional suggestions for improving clarity. We have carefully addressed the minor comments and incorporated the necessary changes to further enhance the manuscript.

In the abstract, often you have space inserted before period, please remove them.

Response: Thank you for your careful review. We have thoroughly checked the abstract and removed all unnecessary spaces to ensure proper formatting.

L. 67 Perhaps clarify by stating 'using two rodent models' so that it is clearer later.

Response: Thank you for your suggestion. We have revised the text to explicitly state "using two rodent models" to enhance clarity and ensure consistency throughout the manuscript.

L. 127-128 it is not the differentially expressed genes that are enriched in biological functions but some functions that are enriched with differentially expressed genes.

Response: Thank you for your insightful comment. We have revised the wording as follows:

"Some functions that are enriched with upregulated genes in all three regions include hemoglobin complexes, oxygen transport, and detoxification."

L. 259 To clarify, consider adding "On the contrary, a hamster that explores more will end up in the center of the enclosure and is more likely to be recaptured." This information could also be included in the Methods section. I appreciated the detailed explanation in the reviewer's response document, but for next readers I think adding that here will help to understand the rationale.

Response: Thank you for your thoughtful suggestion. We have incorporated this sentence to clarify our reasoning at L259.

L. 261: The difference of treatment between each round needs to be explained before you start describing the results.

Response: Thank you for your valuable feedback. We have added a description of the treatments for each round of the experiment before presenting the results, as follows:

“We conducted three rounds of enclosure infection experiments with striped hamsters: no fleas were introduced in any enclosures-R1, fleas were introduced in some enclosures-R2, and fleas were introduced in all enclosures-R3 (Figure 6e, Supplementary Fig. 10g).”

Discussion, I am not convinced at all by the additions made to the discussion about other host-parasite systems where the parasite manipulates its host. These examples do not significantly enhance the perceived novelty of the work. From literature there is abundant evidence of parasitic manipulation, which does not need to be enumerated here. What is more striking is that most examples involve endoparasites, which, in my view, should be more strongly emphasized. Revealing mechanistically how a parasite that is not inside the host can still directly alter its behavior (and not through resources reallocation) is quite new which is why to me this should be central here.

Response: Thank you for your insightful feedback. In our discussion, we aimed to address two key aspects of parasite manipulation: (1) the behavioral alteration observed in our study—namely, the reduction in exploratory behavior following fleabites, and (2) the neuroimmune molecular mechanisms underlying this behavioral change. We understand your suggestion that the discussion should focus more on the mechanistic aspect, and we greatly appreciate this perspective.

Our rationale for structuring the discussion in this manner is as follows: Although we provided multiple examples of parasite manipulation, these likely represent the majority, if not all, of the documented cases in the literature. During our literature review, we found that studies on parasite-induced behavioral manipulation remain relatively scarce. Moreover, discussions with other researchers and experts revealed that many are either unaware of or skeptical about the phenomenon of parasite manipulation. Therefore, we believe that this field is still relatively niche.

As you correctly pointed out, the vast majority—likely over 99%—of reported cases of parasite-induced host manipulation involve endoparasites, particularly those residing in the host's brain. To our knowledge, there are almost no studies demonstrating that an ectoparasite can manipulate the behavior of a mammalian host with a complex brain. Given this, we consider the observed reduction in exploratory behavior following flea bites to be a particularly novel finding. However, recognizing that many researchers may be unfamiliar with this field, we initially chose to present a broader context by discussing

other examples of parasite-induced manipulation. This was intended both to enhance readers' understanding of the topic and to highlight the uniqueness of our study by comparison.

Furthermore, as we mentioned in our previous response to Reviewer 3's comments, we consider the observed phenotype—namely, the reduction in exploratory behavior following flea infestation—to be a central theme that connects various aspects of our study. This includes the mechanistic investigation of reduced exploratory behavior in mice, the results from both the indoor infection experiment and the fenced field experiment in striped hamsters, as well as the *rangeshiftR* model simulations.

Therefore, we believe that this study should be framed from a broader perspective, focusing on the impact of flea infestation on host population dynamics. While our research does explore neuroimmune molecular mechanisms, these findings primarily serve to reinforce our conclusions rather than being the central focus of the study. Additionally, we fully acknowledge that we have not yet conducted microglial depletion or inhibition experiments, which would provide more direct evidence for the mechanistic pathway. This remains an important direction for our future work.

That said, we fully agree with your statement that "*Revealing mechanistically how a parasite that is not inside the host can still directly alter its behavior (and not through resource reallocation) is quite new.*" Indeed, the mechanistic investigation is a critical and novel aspect of our study. However, we believe that elucidating these mechanisms relies on first establishing the behavioral phenotype—namely, that flea bites reduce host exploratory behavior. Our approach to investigating the underlying neuroimmune mechanisms was largely informed by research on microbial infections affecting the host brain. In fact, to our knowledge, the only well-documented case of a parasite manipulating host behavior to enhance its own fitness at the mechanistic level comes from studies on *Toxoplasma gondii*. We have incorporated comparisons with these limited studies in the second part of our discussion. Admittedly, discussing the molecular mechanisms is challenging due to the scarcity of related research.

Nonetheless, we appreciate your concerns and will revise the manuscript accordingly. Specifically, we will reduce the proportion of discussion dedicated to parasite manipulation strategies, either by condensing or summarizing this section. Additionally, we will further emphasize the novelty of the molecular mechanisms underlying "flea bite-induced reduction in exploratory behavior" and their contribution to advancing this field.

Thank you again for your valuable suggestions, which have helped us refine our discussion.

L. 426 Please include an explanation for infecting during the daytime (since *in natura* infection occurs when hamsters are resting) for the benefit of future readers.

Response: Thank you for your suggestion. We have clarified that, “*Given the nocturnal nature of striped hamsters, as in natura infection occurs when they are resting, and their solitary behavior*”. This explanation has been incorporated to ensure that future readers understand the rationale behind our experimental timing and design.

In the reference list, double check that any species name is consistently italicized.

Response: Thank you for your valuable feedback. We have carefully reviewed the reference list and made the necessary adjustments as per your request.

Fig. 2: Panels C and E are not readable.

Fig. 3: The axis labels are too small and should be enlarged for readability.

Fig. 7: Panels a and e are not readable.

Fig. 9: The overall figure is too small and difficult to read.

Response: Thank you for your valuable feedback regarding the readability of the figures. We have made the necessary adjustments as per your suggestions, including rearranging the figure layout, modifying the figure size, and enlarging the axis labels to improve clarity and readability. Additionally, we would like to state that the size of the revised manuscript for this round exceeds 30M, so we have uploaded it in PDF format, which may affect the resolution and readability. We hope these modifications meet your expectations. Thank you again for your review and suggestions!

Reviewer #3 (Remarks to the Author):

Generally, Authors have done a good job for the revision. Most of questions have been answered properly. However, there are still a few problems to be addressed before the final acceptance.

Response: Thank you for your thoughtful review and for acknowledging our efforts in the revision. We appreciate your constructive feedback and have carefully addressed the remaining issues to ensure the manuscript meets the highest standards. Below, we provide detailed responses to each of your comments.

1. The resolution of main figures has been improved in the revised version but needs further elevation. In contrast, the resolution of supplemental figures is good for publication.

Response: Thank you for your feedback. We have further enhanced the resolution of the main figures to ensure they meet publication standards while maintaining clarity and readability. Additionally, we would like to state that the size of the revised manuscript for this round exceeds 30M, so we have uploaded it in PDF format, which may affect the resolution.

2. The gating strategy in Supplementary Fig 14 and Fig. 3h may have some problems since there should be such huge amount of CD45⁺CD11b⁻ cells (the population below the microglia cluster) in Flea⁻ group (as shown in PMID: 29133437; 31325960). In addition, during the systemic inflammation, the amount of macrophages should be highly increased, which was not reflected by the ratio in Fig. 3h.

Response: Thank you for your insightful comments. We appreciate the opportunity to clarify and improve our manuscript based on your valuable feedback.

Regarding your first concern—*“there should be such a huge amount of CD45⁺CD11b⁻ cells (the population below the microglia cluster) in the Flea⁻ group”*—we believe this may have resulted from a misunderstanding due to the way our data were presented. However, we did observe a higher number of CD45⁺CD11b⁻ cells in the Flea⁻ group. There are two main reasons that might have led to this confusion:

1. In **Figure 3h**, we did not explicitly label the flow cytometry plots to distinguish between the Flea⁻ and Flea⁺ groups, which may have obscured the difference.
2. In **Supplementary Figure 14**, the gating strategy we presented was specifically from the Flea⁻ group, which may have inadvertently contributed to the confusion.

We sincerely appreciate your keen observation and have now updated the figures and figure legends to better illustrate this distinction, hoping this revision will address your concerns.

Regarding your second concern—*“during systemic inflammation, the amount of macrophages should be highly increased, which was not reflected by the ratio in Figure 3h”*—we acknowledge that several factors may contribute to this observation:

1. **The time point of our analysis may correspond to an early or specific stage of inflammation**, where macrophage infiltration has not yet reached its peak. As we indicated in our **first-round response**, unpublished behavioral data suggest that exploratory differences in flea-infected mice firstly become apparent by the fourth week, whereas no significant differences were observed at the second week. This suggests that host responses to flea infestation may follow a gradual rather than an acute trajectory.
2. **Differences in experimental models** may account for the discrepancy with the studies you referenced. In those studies, TMEV (Theiler’s murine encephalomyelitis virus) infection was used, where the virus directly infects the brain, likely triggering an acute immune response and rapid inflammation. In contrast, flea infestation induces a chronic immune response through repeated skin bites and blood feeding, which may engage different inflammatory pathways and kinetics.
3. **Although partial blood-brain barrier (BBB) disruption occurs, it may not be sufficient to allow substantial infiltration of peripheral macrophages** into the central nervous system (CNS).
4. **Microglia are the primary immune cells in neuroinflammation**, and their activation may modulate or suppress the infiltration of peripheral macrophages.
5. **The cardiac perfusion procedure performed prior to tissue collection (Line 644) may have removed a significant portion of peripheral macrophages**, thus reducing their detected presence in our samples.

We greatly appreciate your insightful suggestions. While we may not be able to provide a definitive answer within the scope of this study, your questions have sparked important considerations for our future research. In particular, they highlight the need to further investigate the role of the immune system in the context of parasite-induced host manipulation.

Thank you again for your thoughtful review. We hope our clarifications and updates address your concerns, and we remain open to any further suggestions you may have.

3. Authors explanation for my comment 5-7 ("This study is primarily exploratory, focusing on the behavioral strategy of "ectoparasite manipulation of host dispersal," while the molecular mechanisms serve as supporting evidence for our conclusions") is acceptable but also weakens the scientific significance of this study due to unclear mechanism. The abstract needs further editing because no solid evidence to support the conclusion that "ectoparasites can alter metabolic activity in specific brain regions of mice, particularly by excessively activating microglia in the prefrontal cortex. This activation damages the synaptic structures of neurons, disrupting the normal differentiation of GABAergic neurons" without an experiment deleting or inhibiting microglia.

Response: Thank you for your insightful comment. We acknowledge that without a direct experiment inhibiting or deleting microglia, the mechanistic evidence remains correlative. To address this concern, we have revised the abstract to adopt a more cautious phrasing regarding microglial activation and its effects. Specifically, we have changed:

- *"ectoparasites can alter metabolic activity in specific brain regions of mice, particularly by excessively activating microglia in the prefrontal cortex"*
to *"ectoparasites can influence metabolic activity in specific brain regions of mice, with evidence suggesting a potential role of microglial activation in the prefrontal cortex."*
- *"This activation damages the synaptic structures of neurons, disrupting the normal differentiation of GABAergic neurons."*
to *"This activation may contribute to synaptic alterations and changes in neuronal differentiation, particularly affecting GABAergic neurons."*

These revisions ensure that our conclusions remain well-supported by the available data while avoiding overstatement. We appreciate your suggestion, as it has helped us refine the scientific clarity and rigor of our manuscript.

Submission ID: **NCOMMS-24-65632C**
point-by-point response to reviewers

Reviewer #3 (Remarks to the Author):

Authors addressed all the issues mentioned in the previous review and the manuscript can be accepted in its current version.

Response: We sincerely thank the reviewer for their time and positive evaluation. We are glad that the revisions have addressed all concerns, and we appreciate the recommendation to accept the manuscript in its current form.